# A click-based electrocorticographic brain-computer interface enables long-term high-performance switch scan spelling
Daniel N. Candrea[1] ✉, Samyak Shah[2], Shiyu Luo [1], Miguel Angrick [2], Qinwan Rabbani [3], Christopher Coogan [2], Griffin W. Milsap [4], Kevin C. Nathan[2], Brock A. Wester[4], William S. Anderson [5], Kathryn R. Rosenblatt [6], Alpa Uchil[2], Lora Clawson[2], Nicholas J. Maragakis[2], Mariska J. Vansteensel[7], Francesco V. Tenore [4], Nicolas F. Ramsey [7], Matthew S. Fifer [4] & Nathan E. Crone [2]

## Abstract

**Background** Brain-computer interfaces (BCIs) can restore communication for movement- and/or speech-impaired individuals by enabling neural control of computer typing applications. Single command click detectors provide a basic yet highly functional capability.

**Methods** We sought to test the performance and long-term stability of click decoding using a chronically implanted high density electrocorticographic (ECoG) BCI with coverage of the sensorimotor cortex in a human clinical trial participant (ClinicalTrials.gov, NCT03567213) with amyotrophic lateral sclerosis. We trained the participant's click detector using a small amount of training data (<44 min across 4 days) collected up to 21 days prior to BCI use, and then tested it over a period of 90 days without any retraining or updating.

**Results** Using a click detector to navigate a switch scanning speller interface, the study participant can maintain a median spelling rate of 10.2 characters per min. Though a transient reduction in signal power modulation can interrupt usage of a fixed model, a new click detector can achieve comparable performance despite being trained with even less data (<15 min, within 1 day).

**Conclusions** These results demonstrate that a click detector can be trained with a small ECoG dataset while retaining robust performance for extended periods, providing functional text-based communication to BCI users.

## Plain Language Summary

Amyotrophic lateral sclerosis (ALS) is a progressive disease of the nervous system that causes muscle weakness and leads to paralysis. People living with ALS therefore struggle to communicate with family and caregivers. We investigated whether the brain signals of a participant with ALS could be used to control a spelling application. Specifically, when the participant attempted a grasping movement, a computer method detected increased brain signals from electrodes implanted on the surface of his brain, and thereby generated a mouse-click. The participant clicked on letters or words from a spelling application to type sentences. Our method was trained using 44 min' worth of brain signals and performed reliably for three months without any retraining. This approach can potentially be used to restore communication to other severely paralyzed individuals over an extended time period and after only a short training period.

Brain-computer interfaces (BCIs) can allow individuals with a variety of motor impairments to control assistive devices using their neural signals[1–10]. These capabilities are derived from single neuron activity or population activity recorded by implantable microelectrode arrays (MEAs) or macro-electrodes (typically consisting of electrocorticographic (ECoG) arrays on the cortical surface)[11], respectively, as well as by non-invasive recording modalities such as electroencephalography (EEG). Although sophisticated capabilities of MEA-based BCIs have been reported, signal attrition[12–14] may affect long-term performance while day-to-day signal instability may require frequent decoder recalibration[15]. Nevertheless, there have been promising advances in continual online recalibration (in the background and on a per-trial basis) after correcting text outputs using a language

[1]Department of Biomedical Engineering, Johns Hopkins University School of Medicine, Baltimore, MD, USA. [2]Department of Neurology, Johns Hopkins University School of Medicine, Baltimore, MD, USA. [3]Department of Electrical and Computer Engineering, Johns Hopkins University, Baltimore, MD, USA. [4]Research and Exploratory Development Department, Johns Hopkins University Applied Physics Laboratory, Laurel, MD, USA. [5]Department of Neurosurgery, Johns Hopkins University School of Medicine, Baltimore, MD, USA. [6]Department of Anesthesiology and Critical Care Medicine, Johns Hopkins University School of Medicine, Baltimore, MD, USA. [7]Department of Neurology and Neurosurgery, UMC Utrecht Brain Center, Utrecht, The Netherlands. ✉e-mail: dcandre3@jh.edu

model[16]. On the other hand, EEG-based BCIs can be effective for single-command decoding, which has been used in a variety of paradigms[17]. However, the frequent application and maintenance of the external EEG sensors by a caregiver or technician would still be necessary. Alternatively, ECoG-based BCIs may offer robust long-term and accessible functionality without frequent model retraining due to the cortical stability at the population level[1,18]. However, the utility of ECoG for chronically (>30 days) implanted BCIs has only been tested in a few participants[1,3,19].

Recent studies have demonstrated ECoG-based BCI control for participants with amyotrophic lateral sclerosis (ALS) by detecting a brain click[1,3,19], an event-related change in spectral signals due to a distinct action, such as attempting a hand movement. In a recent clinical trial, participants with ALS (or primary lateral sclerosis) were implanted with an endovascular stent-electrode array for detecting such brain clicks[3,19]. These brain clicks were generated by attempted foot movements and were used to select a particular icon or letter on a computer screen after navigating to it via eye-tracking (ET)[19]. As a result, participants achieved high spelling rates and required 1–12 sessions of training with their brain click BCI before long-term use. Despite these impressive results, when testing the BCI-only system (without ET), an accuracy of 97.4% was achieved but with a detection latency of 2.5 s, while a detection latency of 0.9 s corresponded with a reduced accuracy of ~82%[3]. Thus, the spelling performance of the BCI system alone remains unclear.

A switch scanning paradigm is an augmentation and alternative communication (AAC) method of communication that allows users to navigate to and select icons or letters by timing their clicks to when a desired row or column is highlighted[1,20–26]. However, the user is not required to control a cursor using ET, which can be tiring[27–29] and can become ineffective as eye movements deteriorate in ALS[30–33]. In an earlier clinical trial of chronic ECoG BCI, a participant with ALS attempted hand movements to generate brain clicks, in turn controlling a switch scanning speller application. These brain clicks were detected from a single pair of electrodes on the surface of the hand area of the contralateral motor cortex[1]. Though the participant used these brain clicks to communicate in her daily life for more than 3 years[18], several months of data collection were necessary for parameter optimization. Click accuracy, comprised of correctly detected and withheld clicks, was reported between 87–91% with a 1 s latency and a maximum scan rate of 0.5 per s.

The previous studies described above showed that click detectors can be used with a variety of BCI applications and can contribute substantially to a user's repertoire of communication modalities. Despite these promising results, the potential performance limits of such click detectors have remained relatively underexplored. In particular, chronic high-performance use without model retraining is a critical factor for enabling independent home-use, as BCI users should have round-the-clock access to a functioning click detector that requires minimal caregiver involvement. By leveraging the stability of ECoG signals, we are able to train a model on a limited dataset and test it for a period of three months without retraining or daily model adaptation. Specifically, we demonstrate a switch scanning BCI with a substantially improved spelling rate compared to prior switch scanning BCI work[1].

## Methods
### Clinical trial
This study was performed as part of the CortiCom clinical trial (ClinicalTrials.gov Identifier: NCT03567213), a phase I early feasibility study of the safety and preliminary efficacy of an implantable ECoG BCI. The current recruitment status is "Recruiting" and no more than five participants are planned to be enrolled and implanted. Due to the exploratory nature of this study and the limited number of participants, the primary outcomes of the trial were stated in general terms and were designed to gather preliminary data on: (1) the safety of the implanted device, (2) the recording viability of the implanted device, and (3) BCI functionality enabled by the implanted device using a variety of strategies. No methods or statistical analysis plans were predefined for assessing these outcomes given their exploratory nature

and the limited number of participants. Nevertheless, the results reported for each participant are to be evaluated to the highest statistical rigor in keeping with comparable studies, which were also limited to individual participants[1,2,5,7,8,34]. Results related to the first two primary outcome variables, though necessarily provisional as they are drawn from only one participant, are reported in the Results and Supplementary Note 1, respectively. Results related to BCI functionality, also necessarily provisional and exploratory (Supplementary Note 2), are addressed within the subsequent methodology and results, which nevertheless employed rigorous analyses and statistics. The secondary outcomes of the CortiCom trial are reported here only partially, as click detection is only one of a variety of BCI control strategies explored by the trial; specifically, the success rate and latency are reported in terms of click detection accuracy and time from attempted movement onset to click. The study protocol can be found as an additional supplemental file. Clinical trial inclusion and exclusion criteria are available in Supplementary Method 1. The study protocol was reviewed and approved by Johns Hopkins University Institutional Review Board and by the US Food and Drug Administration (FDA) under an investigational device exemption (IDE).

### Participant
All results reported here were based on data from the first participant in the CortiCom trial. The participant gave written consent to participation in the trial after being informed of the nature of the research and implant related risks and was enrolled on July 5th, 2022. Additionally, the participant gave written consent for publication of clinical data and results relevant to the study. The patient later gave written consent for use of his audio and video recordings in the publication of study results. The experimental team was scheduled to meet with the participant three times each week for training data collection or BCI use. Experimental planning occurred weekly in a laboratory setting at Johns Hopkins Hospital and was informed by task-specific progress. To date this participant has had no serious or device-related adverse events, and thus the primary outcome of the CortiCom trial has been successful. The participant has consented to continue the study. At this time the device has been implanted for >2 years and continues to be used for research purposes.

The participant was a right-handed man who was 61 years old at the time of implant in July 2022 and diagnosed with ALS roughly 8 years prior. Due to bulbar dysfunction, the participant had severe dysphagia and progressive dysarthria. This was accompanied by progressive dyspnea. The participant could still produce overt speech, but slowly and with limited intelligibility. He did not, however, heavily rely on assistive communication devices (Supplementary Note 3). He had experienced progressive weakness in his upper limbs such that he was incapable of performing activities of daily living without assistance. He could partially close his fingers in an attempted grasp gesture, but he had insufficient strength to hold a cup with one hand. His lower limbs had good strength and allowed him to ambulate, albeit with intermittent imbalance due to impaired arm swing. The ALSFRS-R[35] measure was used to evaluate the participant's capacity to perform activities of daily living, and he received a score of 26 out of a total of 48 points (Supplementary Note 4).

The participant was screened with cognitive testing prior to his enrollment in the study, and no evidence for dementia was found. During monthly safety assessments, the participant underwent a brief cognitive testing battery. This has not revealed any substantial decline in cognitive function since study enrollment.

### Neural implant
The CortiCom study device was composed of two 8×8 subdural ECoG grids manufactured by PMT Corporation (Chanhassen, MN), which were connected to a percutaneous 128-channel Neuroport pedestal manufactured by Blackrock Neurotech Corporation (Salt Lake City, UT). Final assembly and sterilization of the study device was performed by Blackrock Neurotech. Both subdural grids consisted of soft silastic sheets embedded with platinum-iridium disc electrodes (0.76 mm thickness, 2 mm diameter

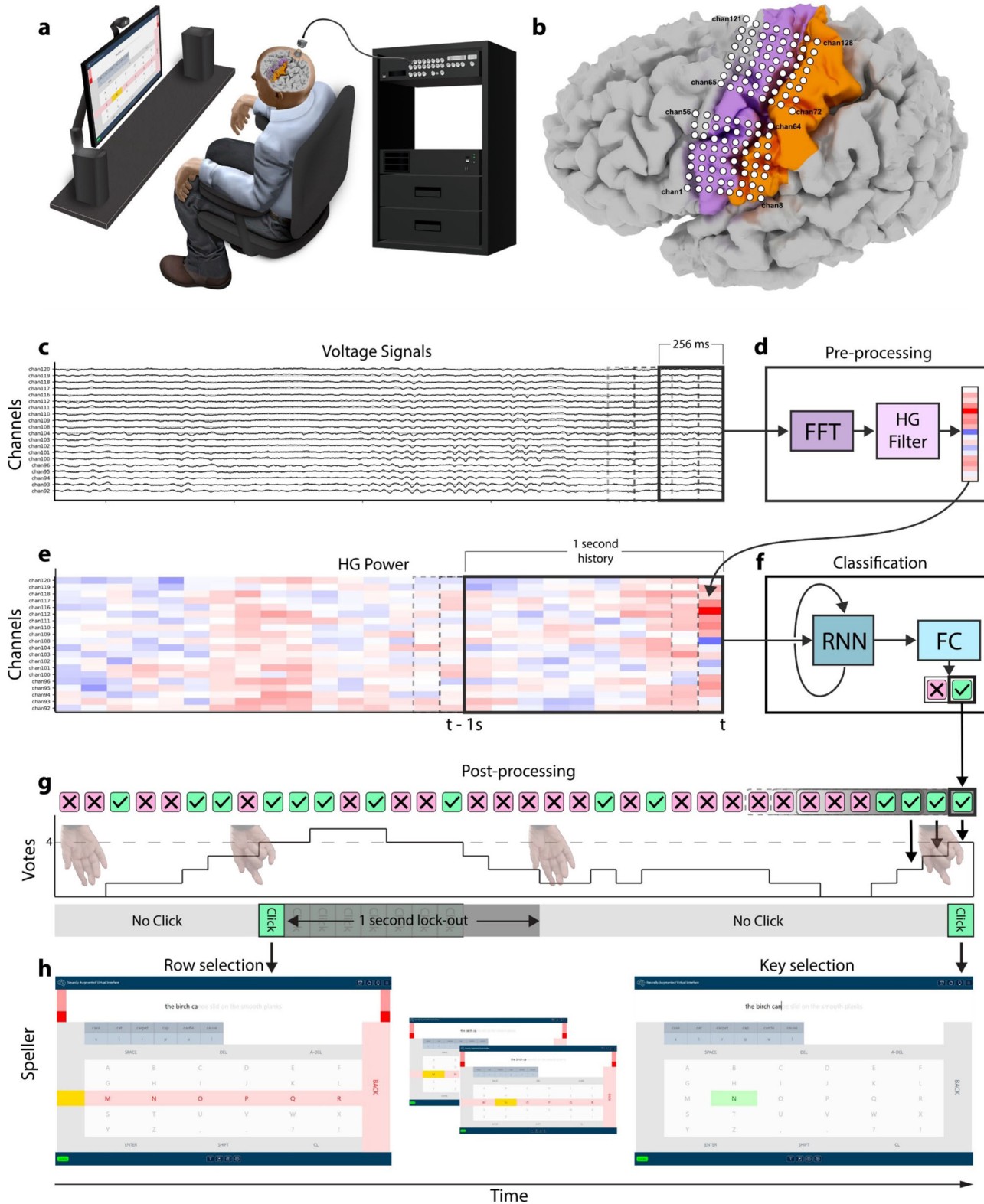

exposed surface) with 4 mm center-to-center spacing and a total surface area of 12.11 cm² (36.6 mm × 33.1 mm). The device included two subdural reference wires, the tips of which were not insulated to match the recording surface area of the ECoG electrodes. Due to the small diameter of the wires (0.07 mm), it was not possible to localize them on a post-surgical CT scan. During all recordings with the study device, the Neuroport pedestal was

coupled to a small (24.9 mm × 17.7 mm × 17.9 mm) external device (Neuroplex-E; Blackrock NeurotechCorp.) for signal amplification, digitization, and digital transmission via a micro-HDMI cable to the Neuroport Biopotential System (Blackrock Neurotech Corp.) (Fig. 1a). During all recordings, signals were referenced to the same reference wire and no other referencing was performed.

**Fig. 1 | Online click detection pipeline. a** The participant was seated upright with his forearms on the armrests of a chair facing a computer monitor where the switch scanning speller application was displayed. **b** Position of both 64-electrode grids overlayed on the left cortical surface of a virtual reconstruction of the participant's brain. The dorsal and ventral grids primarily covered cortical upper limb and face regions, respectively. The electrodes are numbered in increasing order from left to right and from bottom to top. Magenta: pre-central gyrus; Orange: post-central gyrus. **c** ECoG voltage signals were streamed in 100 ms packets to update a 256 ms running buffer for online spectral pre-processing. A sample of signals from 20 channels is shown. **d** A Fast Fourier Transform filter was used to compute the spectral power of the 256 ms buffer, from which the high gamma (HG, 110-170 Hz) log-power was placed into a 1 s running buffer (10 feature vectors). **e** The running

buffer was then used as time history for the recurrent neural network (RNN). **f** An RNN-FC (RNN-fully connected) network then predicted rest or grasp every 100 ms depending on the higher output probability. **g** Each classification result was stored as a vote in a 7-vote running buffer such that the number of grasp votes had to surpass a predetermined voting threshold (4-vote threshold shown) to initiate a click. A lock-out period of 1 s immediately followed every detected click to prohibit multiple clicks from occurring during the same attempted movement. Transparent images of the hand are shown to indicate the attempted grasp before a click and a relaxed configuration otherwise. **h** Once a click was detected, the switch scanning speller application selected the highlighted row or element within that row. Two clicks were necessary to type a letter or autocomplete a word. The example sentence shown is "the birch canoe slid on the smooth planks."

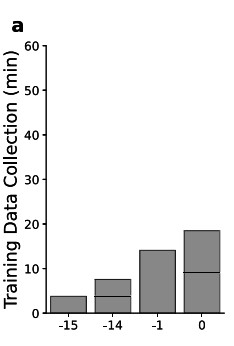
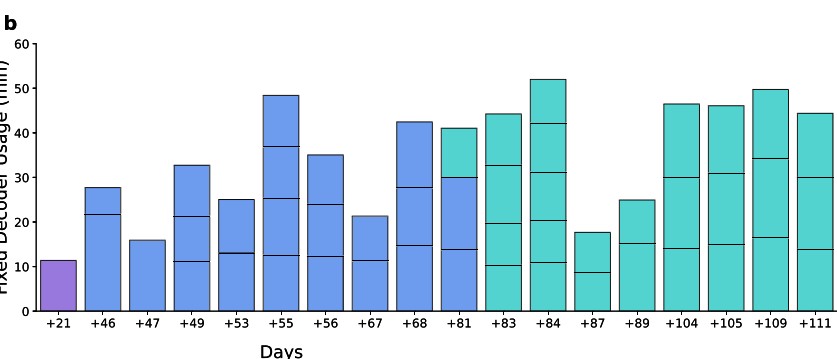

**Fig. 2 | Long-term use of a fixed click detector. a** Training data was collected during 4 sessions that occurred within a period of 16 days. For each day, each sub-bar represents a separate block of training data collection (6 training blocks total). **b** Using the fixed detector, one block of switch scanning with the communication board was performed +21 days post-training data collection (purple). From Day +46 to Day +81, the fixed detector was used for switch scan spelling with a 7-vote

threshold (blue). From Day +81 to Day +111, the fixed detector was used for switch scan spelling with a 4-vote threshold (teal). For each day, each sub-bar represents a separate spelling block of 3-4 sentences. The horizontal axis spanning both (**a**) and (**b**) represents the number of days relative to the last day of training data collection (Day 0).

The two electrode grids of the study device were surgically implanted subdurally, over sensorimotor cortex representations for speech and upper extremity movements in the left hemisphere. Implantation was performed via craniotomy under monitored anesthesia care with local anesthesia and sedation tailored to intraoperative task participation. There were no surgical complications or surgically related adverse events. The locations of targeted cortical representations were estimated prior to implantation using anatomical landmarks from a pre-operative structural MRI, functional MRI (sequential attempted finger tapping, tongue movement, and humming), intraoperative somatosensory evoked potentials, and intraoperative high gamma responses to vibrotactile stimulation of the individual fingers. The locations of the subdural grids with respect to surface gyral anatomy were confirmed after implantation by co-registering a post-operative high-resolution CT with a pre-operative high-resolution MRI using FreeSurfer[36] (Fig. 1b).

### Testing and calibration

At the beginning of each session, a 60 s calibration period was recorded, during which the participant was instructed to sit still and quiet with his eyes open and visually fixated on a computer monitor. For each channel, we then computed the mean and standard deviation of the spectral-temporal log-powers for each frequency bin. These statistics of resting baseline cortical activity were subsequently used for normalization of power estimates during model training and BCI operation.

### Training task

Training data was collected across four sessions (six training blocks in total) spanning 16 days (Fig. 2a). We defined Day 0 as the last session of training data collection. For each block, the participant was instructed to attempt a brief grasp with his right hand (i.e., contralateral to the implanted arrays) in

response to visual cues (Supplementary Fig. 1). Due to the participant's severe upper extremity impairments, his attempted movements primarily involved flexion of the middle and ring fingers. After each attempt, the participant released his grasp and passively allowed his hand to return to its resting position hanging from the wrist at the end of his chair's armrest.

Each trial of the training task consisted of a single 100 ms Go stimulus (referred to as "Go cue" hereafter) prompting the participant to attempt a grasp, followed by an interstimulus interval (ISI), during which the participant remained still and fixated his gaze on a crosshair in the center of the monitor. Previous experiments using longer cues had resulted in more variable response latencies and durations. The length of each ISI was randomly chosen to vary uniformly between a lower and upper bound (Supplementary Table 1) to reduce anticipatory behavior. The experimental parameters across all training sessions are shown in Supplementary Table 1. In total, almost 44 min of data (480 trials) were collected for model training.

### Data collection

All data collection and testing were performed in the laboratory. Neural signals were recorded by the Neuroport system at a sampling rate of 1 kHz. BCI2000[37] was used to present stimuli during training blocks and to store the data from training and online BCI use with the click detector for offline analysis[37]. Video of the participant's right hand (which was overtly attempting grasp movements) and the monitor displaying the spelling application was recorded at 30 frames per second (FPS) during all spelling sessions except the last two (at 60 FPS). A 150 ms synchronization audio cue was played at the beginning of each spelling block (see *Online Switch scanning*) so that the audio recorded by the Neuroport biopotential system's analog input could be used offline to synchronize the video frames with the neural data. Click detection time stamps were recorded by BCI2000 and were synchronized with the neural data. A pose estimation algorithm[38] was

applied offline to the video of the participant's hand to infer the horizontal and vertical positions of 21 hand and finger landmarks within each video frame. The horizontal coordinates of the metacarpal-phalangeal (MCP) joint landmarks for the first and fifth digits were used to normalize horizontal positions of all landmarks, while the MCP and fingertip coordinates of the same digits were used to normalize vertical positions.

### Feature extraction and label assignment

**Feature extraction.** For each of the 128 recording channels, we used a Fast Fourier Transform (FFT) filter to compute the spectral power of 256 ms windows shifted by 100 ms increments. The spectral power in each frequency bin was log-transformed and normalized to the corresponding calibration statistics. We summed the spectral power in the frequency band between 110 and 170 Hz to compute the high-gamma (HG) power. 110 Hz was chosen as the lower bound of this HG frequency band because post-movement low frequency activity sometimes extended to and slightly past 100 Hz in several channels (Supplementary Fig. 2). For each 100 ms increment, this resulted in a 128-channel feature vector that was used in subsequent model training. We chose this frequency band due to the rapid timescale of HG modulation during attempted grasping and chose to exclude features from lower frequency bands due to event-related synchronization (ERS) occurring immediately after event-related desynchronization (ERD) (Supplementary Fig. 2). This pattern of low frequency ERD quickly followed by ERS occurred on a longer timescale than the HG activity and would have made it difficult to clearly define the onset and offset times of the trial-averaged neural activity for assigning rest and grasp labels for model training (see *Label Assignment*). Similarly, some channels contained low-frequency ERD prior to cue, which was likely due to anticipatory activity (Supplementary Fig. 2). This could have caused greater variance in the feature space of samples labeled as rest.

**Label assignment.** We assigned rest and grasp labels to each sample in the training dataset by the following steps. First, for each channel we concatenated segments of HG power across all trials, where each trial segment ranged from -1 s to 2.5 s relative to the beginning of the visual Go cue (Supplementary Fig. 3, Cue-aligned). To account for the inter-trial variability of the participant's reaction delay to the visual Go cue, we temporally re-aligned the HG power across all trial segments using a shift warping model[39] (Supplementary Fig. 3, Re-aligned). This model was trained on only a subset of highly modulated channels (determined qualitatively; Supplementary Fig. 3 caption) to decrease the potential influence of artificial patterns from low-modulation channels when re-aligning trial segments. Note that for each trial, the resulting temporal re-alignment was applied similarly across all 128 channels. This re-alignment resulted in generally increased HG power correlations between trials (Supplementary Figs. 4, 5). We then computed the trial-averaged HG power traces using the re-aligned trial segments of only these highly modulated channels and visually determined the onset and offset of modulation relative to the beginning of the visual Go cue (Supplementary Fig. 6). This onset and offset time were estimated to be 0.3 s and 1.1 s, respectively, relative to onset of the Go cue. We consequently assigned grasp labels to ECoG feature vectors falling between and including $0.3\,s + t_{shift}$ and $1.1\,s + t_{shift}$ relative to the Go cue for each trial. Note that it was necessary to include the term $t_{shift}$ to the bounds where grasp labels were assigned to account for the shift that was applied to each trial. Rest labels were applied to feature vectors at all other time points. We adopted this labeling strategy because it relied only on the visual inspection of neural signals, simulating the lack of ground truth for attempted movements that would be expected for BCI users with Locked-in Syndrome (LIS).

### Model architecture and training

We used a recurrent neural network (RNN) for classifying rest vs. grasp. As this study is part of a larger clinical trial, we aimed to build the model training pipeline such that in the future we could train more complex models for tasks in which the temporal domain would substantially contribute to decoder performance. Additionally, we aimed to allow the participant to use a high-performing BCI as soon as possible, and to this end we anticipated that a non-linear classifier would achieve higher performance than a linear model due to the advantage of recognizing temporal patterns in neural activity.

**Model architecture.** We designed an RNN in a many-to-one configuration to learn the temporal dynamics in HG power over sequences of 1 s with each sequence associated with only the label at the leading edge of the sequence. Each 128-channel HG power vector was input into a long short-term memory (LSTM) layer with 25 hidden units for modeling sequential dependencies. From here, 2 consecutive fully-connected (FC) layers with 10 and 2 hidden units, respectively, determined probabilities of the rest or grasp class (Supplementary Fig. 7). The former utilized an eLU activation function while the latter employed softmax to output normalized probability values. In total, the architecture consisted of 17,932 trainable parameters.

**Equal class sizes for training.** Since 800 ms of data per-trial were labeled as grasp (see *Label assignment*), while the remainder of the time in the trial ($t_{remainder} = t_{min\,ISI} + t_{Go} - 800\,ms \geq 2,300\,ms$) was labeled as rest, the rest class was overrepresented and therefore randomly downsampled such that the classification model would be trained on a balanced dataset of rest and attempted grasping sequences. Note that $t_{remainder}$ is at least 2,300 ms because the minimum ISI was 3 s, while the duration of the visual cue Go was 0.1 s.

**Model training.** We determined the classification model's hyperparameters by evaluating its offline accuracy using 10-fold cross-validation with data collected for training (see *Cross-validation*). For each cross-validated model, we limited training to 75 epochs during which classification accuracy of the validation fold plateaued. We used categorical cross-entropy for computing the error between true and predicted labels of each 45-sample batch and updated the weights using adaptive moment optimization (Adam optimizer)[40]. To prevent overfitting on the training data, we used a 30% dropout of weights in the LSTM and FC layers. All weights were initialized according to a He Normal distribution[41]. The model was implemented in Python 3.8 using Keras with a TensorFlow backend (v2.8.0).

**Cross-validation.** We partitioned the training data into 10 folds such that each fold contained an equal number of rest and grasp samples of HG power feature vectors (rest samples were randomly downsampled to match the number of grasp samples). To minimize data leakage of time dependent data into the validation fold, all samples within a fold were contiguous and each sample belonged to only one fold. Each fold was used once for validation and a corresponding cross-validated model was trained on the remaining 9 folds.

### Online pipeline

**Pipeline structure.** We used ezmsg, a Python-based messaging architecture (https://github.com/iscoe/ezmsg)[42], to create a directed acyclic graph of processing units, in which all pre-processing, classification, and post-processing steps were partitioned.

**Online pre-processing.** Neural data was streamed in intervals of 100 ms via a ZeroMQ connection from BCI2000 to the online pipeline, which was hosted on a separate machine dedicated to online inference. Incoming data updated a running 256 ms buffer, from which a 128-channel feature vector of HG power was then computed as described above (Fig. 1c, d). This feature vector was stored in a running buffer of 10 feature vectors (Fig. 1e), which represented 1 s of feature history for the LSTM input (Fig. 1f).

**Fig. 3 | Switch scanning applications.** The participant was instructed to select an experimenter-cued graphical button (**a**) or to spell the sentence prompt (pale gray text) (**b**) by timing his clicks to the appropriate highlighted row or column during the switch scanning cycle. For a detailed description of (**a**) and (**b**), refer to Supplementary Figs. 8 and 9, respectively.

**Classification and post-processing**. A rest or grasp classification was generated every 100 ms by the FC layer, after which it entered a running buffer of classifications, which in turn was updated with each new classification. This buffer was the voting window, which contained a pre-determined number of classifications (10 and 7 for the medical communication board and the spelling interface, respectively), and in which a given number of those classifications (voting threshold) were required to be grasp in order to initiate a click (Fig. 1g). This voting window and threshold were applied to prevent sporadic grasp classifications from being interpreted as an intention to generate a click. A click triggered the selection of the participant's desired row or column in the switch scanning application (Fig. 1h).

**Switch scanning applications**

A switch scanning application is an augmentation and alternative communication (AAC) technology that allows users with severe motor or cognitive impairments to navigate to and select icons or letters by timing their clicks to the desired row or column during periods in which rows or columns are sequentially highlighted[20-26]. The participant generated a click by attempting a brief grasping movement as described in *Training task*.

**Medical communication board**. As a preliminary assessment of the click detector's sensitivity and false positive detections, we first cued the participant to navigate to and select keys with graphical symbols from a medical communication board (Fig. 3a, Supplementary Fig. 8). Graphical symbols were obtained from https://communicationboard.io/[43] and https://arasaac.org/[44]. We used a 10-vote voting window with a 10-vote threshold (all 10 classifications within the running voting window needed to be grasp to initiate a click) and set the row and column scan rates to 0.67 per s. Finally, we enforced a lock-out period of 1 s, during which no other clicks could be produced, after clicking on a row or a button within a row (Fig. 1g). This prevented multiple clicks being produced from the same attempted grasp.

**Spelling application**. We then developed a switch scanning speller application, in which the participant was prompted to spell sentences (Supplementary Fig. 9). The buttons within the spelling interface were arranged in a grid design that included a center keyboard as well as autocomplete options for both letters and words. Letter and word auto-completion options were both generated by a distilBERT language model[45] hosted on a separate server, providing inference through an API. The distilBERT model was chosen over larger language models for its faster inference speed. We added three pre-selection rows at the beginning of each switch scanning cycle as well as one pre-selection column at the beginning of column scanning cycle. These allowed the participant a brief preparation time if he desired to select the first row, or first column within a selected row. We decided to use a 7-vote voting window with a

7-vote threshold, which decreased latency from attempted grasp onset to click (see *Click latencies*) compared to using a 10-vote threshold with the medical communication board. However, after several sessions of spelling and feedback from the participant, on Day +81 we reduced the voting threshold requirement to a 4-vote threshold (any 4/7 classifications within the voting window needed to be grasp to initiate a click). This was because the participant reported that he preferred increased sensitivity despite a possible increase in false positive detections. We again enforced a lock-out period of 1 s.

**Online switch scanning**. Using the communication board, the participant was instructed to navigate to and select one of the keys verbally cued by the experimenter. If the participant selected the incorrect row, the cued key was changed to be in that row. Once a key was selected, the switch scanning cycle would start anew (Supplementary Movie 1, Fig. 3a, Supplementary Fig. 8). We recorded one session with the communication board on Day +21 after the completion of training data collection (Fig. 2b).

To test online spelling performance using the fixed click detector, the participant was required to type out sentences by using the switch scanning speller application. The sentences were sampled from the Harvard sentence corpus[46] and were presented at the top of the spelling interface in pale gray text. If the participant accidentally clicked a wrong key, resulting in an incorrect letter or autocompleted word, the corresponding output text would be highlighted in red. The participant was then required to delete it using the DEL or A-DEL (auto-delete) keys, respectively. Once the participant completed a sentence, he advanced to the next one by clicking the ENTER key (Supplementary Movie 2, Fig. 3b, Supplementary Fig. 9). A spelling block consisted of 3-4 sentences to complete, and in each session the participant completed 1–5 spelling blocks (Fig. 2b). We recorded blocks with the switch scanning speller application across 17 sessions.

**Performance evaluation**

**Sensitivity and click rates**. Sensitivity was measured as the percentage of correctly detected clicks:

$$Sensitivity = \frac{N_{correct\ clicks}}{N_{attempted\ grasps}} \times 100\% \qquad (1)$$

where in one session $N_{correct\ clicks}$ was the total number of correct clicks and $N_{attempted\ grasps}$ was the total number of attempted grasps, and where $N_{correct\ clicks} \leq N_{attempted\ grasps}$. For a detected click to be correct (i.e., a true positive), it must have appeared on the user interface (as visual feedback to the participant) within 1.5 s after the onset of an attempted grasp. Attempted grasps with no clicks occurring within this time period were considered false negatives. Clicks that occurred outside this time period were assumed to be unrelated to any attempted grasp and were thus considered false positives.

True positive and false positive frequencies (TPF and FPF, respectively) were measured per unit time and for each session were defined as the following:

$$TPF = \frac{N_{TP}}{T} = \frac{N_{correct\ clicks}}{T} \qquad (2)$$

$$FPF = \frac{N_{FP}}{T} \qquad (3)$$

where $N_{TP}$ and $N_{FP}$ were the number of true and false positives in a session, respectively, and $T$ was the total spelling time for that session. Whether the participant clicked the correct or incorrect key had no bearing on sensitivity, TPF, or FPF as these metrics depended only on whether a click truly occurred following an attempted grasp.

**Click latencies**. Movement onsets and offsets were determined from the normalized pose-estimated landmark trajectories of the hand. Specifically, only the landmarks of the fingers with substantial movement during the attempted grasp were considered. Then, for each attempted grasp, movement onset and offset times were visually estimated.

For each correctly detected click, we computed both: (a) the time elapsed between movement onset and algorithm detection, and (b) the time elapsed between movement onset and the click appearing on the spelling application's user interface. The latency to algorithm detection was primarily composed of the time necessary to reach the voting threshold (i.e., a 4-vote threshold produced at minimum 400 ms latency with four sequential grasp classifications). The latency to the on-screen click appearing on the spelling interface depended on the algorithm detection latency along with additional network and computational overhead necessary for displaying the click. This additional overhead was roughly 200 ms (see *Switch scanning performance*).

**Spelling rates**. Spelling rates were measured in units of correct characters per minute (CCPM) and correct words per minute (CWPM). Spelled characters and words were correct if they exactly matched their positions in the prompted sentence. For example, if the participant spelled a sentence with 30 characters (5 words) with 1 character typo, only 29 characters (4 words) contributed to the CCPM (CWPM). The frequency of character typos was measured in units of wrong characters per minute (WCPM). The participant was instructed to correct any mistakes before proceeding to type the rest of the sentence. Note that all spelling was performed with assistance of autocompletion options from the language model and subsequently all analyses of spelling performance were based on this assisted spelling.

**Offline simulations**
**Performance as a function of training trials**. We investigated the relationship between the number of trials used for training the classification model and the resulting simulated performance of a click detector (sensitivity and FPF). This was done to determine whether similar performance to online spelling could have been achieved using a click detector model trained on fewer trials. To do this, we trained classification models with various numbers of training trials and tested them offline on data collected from online spelling sessions. Using the training procedure described above, we trained six models, each trained with an additional block's worth of data (Supplementary Table 1) compared to the preceding model. As such, six models were trained on data containing either 50, 100, 150, 300, 390, or 480 trials (3.77, 7.56, 11.34, 25.43, 34.68, or 43.92 min, respectively). Note that the click detector model used for online spelling was trained on the same 480 trials, the entirety of the original training dataset. The models were tested on data collected from each online spelling block during which the click detector operated with a 4-vote voting threshold. Models were not tested on spelling blocks in which a 7-vote threshold was used because for a majority of these sessions

(Days 46–56), there was no audio-synchronization cue to align the neural data recorded by the Neuroport system (and the resulting click-detections from simulation analysis) with the recorded video frames. As such, it was not possible to accurately determine when the simulated click detections would have occurred relative to the onset of attempted grasp. Sensitivity and FPF were computed as described in *Sensitivity and click rates*. Then, for each specific number of training trials, we computed the across-session median sensitivity and FPF.

**Model updates using previous spelling blocks**. To assess whether the spelling task itself could function as a modality by which to collect further training data, we trained updated classification models with data from spelling blocks of preceding sessions. We then simulated the performance of these models on spelling blocks from the subsequent sessions. For example, the simulated performance of a click detector trained on data from all spelling blocks recorded up until and including day *d* was evaluated on all spelling blocks from day *d + 1*. We used largely the same procedure to train each updated model as we did for the original fixed model, with only two differences. First, we determined the onset and offset times of the re-aligned trial averaged HG power traces relative to the start of attempted movement rather than a Go cue, which was not present during the spelling blocks. Second, since two attempted grasps could have occurred within a very short duration of each other (e.g., clicking into a row followed by clicking into the first column), we excluded from training all attempted grasps which occurred <3 s (the minimum jittered ISI, Supplementary Table 1) after a preceding attempted grasp. As described in *Sensitivity and click rates*, sensitivity using the original fixed detector was computed by determining the number of click detections (occurring as visual feedback to the participant) that occurred within 1.5 s after the onset of an attempted grasp. Since the offline analysis simulated algorithmic detection (and not on-screen clicks), we therefore shifted all click detections by 200 ms to account for the consistent delay between algorithmic detection and on-screen click mentioned above. TPF and FPF were computed as described above. We again only used data from online spelling sessions during which the click detector operated with the 4-vote voting threshold. The seed update model was trained on the third block of the online spelling session on Day +81 (Fig. 2b).

**Channel contributions and offline classification comparisons**
Using the subset of samples in the training data labeled as grasp, we computed each channel's importance to generating a grasp classification given the model architecture. Specifically, we computed the integrated gradients from 10 cross-validated classification models (see *Cross-validation*) with respect to the input features from each sample labeled as grasp in the corresponding validation folds. This generated an attribution map for each sample[47], from which we calculated the L2-norm across all 10 historical time feature vectors[2], resulting in a 1×128 saliency vector. Due to the random initialization of weights in the RNN-FC network, models trained on features from the same set of folds were not guaranteed to converge to one set of final weights. We therefore retrained the set of 10 cross validated models 20 times and similarly recomputed the saliency vectors for each sample. The final saliency map was computed by averaging the attribution maps across all repeated samples and normalizing the resulting mean values between 0 and 1. We repeated this process using HG features from all channels except channel 112, which was located over sensory cortex and showed a relatively high activation compared to other channels during attempted movement. We then repeated this process using HG features from a subset of 12 electrodes over cortical hand-knob (anatomically determined as channels 92, 93, 94, 100, 101, 102, 108, 109, 110, 116, 117, 118; Fig. 5e, Supplementary Figs. 2, 3). Neither of these two model architectures were deployed for online BCI use.

To inform whether models trained with HG features from these smaller subsets of channels could retain robust click performance, we computed offline classification accuracies using 10-fold cross-validation

(see *Cross-validation*) of the training data. We repeated cross-validation such that for each of the 10 validation folds a set of 20 accuracy values was produced. We then took the average of these 20 values to obtain a final accuracy for each fold. For each subset of channels, a confusion matrix and accuracy value were generated using the true and predicted labels across all validation folds and all repetitions. We compared these results to those generated by using features from all channels.

Finally, we computed the confusion matrix and classification accuracy value across all spelling blocks in which the click detector operated with a 7-vote threshold and with a 4-vote threshold using the original fixed model (trained on features from all channels). As described in *Model updates using previous spelling blocks* true grasp labels were assigned to each trial within the bounds of the onset and offset of the re-aligned trial averaged HG power traces relative to the attempted movement start. Again, data corresponding to attempted grasps which occurred <3 s after a preceding attempted grasp were excluded from labeling. All other samples were labeled as rest. An equal amount of rest and grasp samples were used for computing the confusion matrix and corresponding accuracy.

### Cognitive workload
In order to evaluate the participant's experience using the switch scanning speller application, we asked the participant to complete the NASA task load index (NASA-TLX) questionnaire[48,49] using the NASA-TLX iOS application, a commonly used set of questions to evaluate a participant's mental, physical, and temporal demand of a task as well as the perceived performance, effort, and frustration of a task. These categories were each scored from 0-100 where lower and higher scores corresponded, respectively to less and more of each of the six above-mentioned characteristics.

### Statistics and Reproducibility
**Statistical analysis.** Spelling blocks with a specific voting threshold were collected on no more than nine sessions. Given this small sample size, we could not assume normality in the distribution of the sample mean of any of the performance metrics (sensitivity, TPF, FPF, latencies, CCPM, WCPM, and CWPM). Because of this, we computed our 95% confidence intervals (95% CI) of the means using 10,000 bootstrapped replicates (bias-corrected and accelerated) of the samples of each performance metric. Additionally, we used the non-parametric two-sided Wilcoxon Rank-Sum test to determine whether there were significant differences between performance metrics from spelling blocks where different voting thresholds were applied. A *P*-value <0.05 was considered significant. Similarly, we used the two-sided Wilcoxon Rank-Sum test to determine whether there were significant differences in offline classification accuracies when different configurations of channels were used from model training and validation as well as for all simulated performance metrics. We additionally used a Holm-Bonferroni correction to adjust for multiple comparisons.

**Reproducibility of experiments.** Neural data collection, processing and performance of the click detector were reproducible as the participant was able to repeatedly demonstrate click control with stable performance across sessions on different days. Samples of sensitivity, TPF, FPF, latencies, CCPM, WCPM, and CWPM measurements were created with data from online BCI use with voting thresholds ranging from 2 to 7 votes. Only samples from the 4-vote, 6-vote, and 7-vote conditions were statistically compared. Samples of simulated sensitivity, TPF, and FPF were the same size as those created with data from online BCI use with a 4-vote threshold. The sizes of the samples described above were not predefined as they depended on the number of spelling blocks collected with a specific voting threshold, which was adjusted based on participant feedback. We defined the sample size for comparing offline classification accuracies using various channel combinations (All channels, No Channel 112, Hand knob) as the number of folds used for cross-validation. As this study reports on the first participant in this trial so far,

further work will be necessary to test the reproducibility of these results in other participants.

### Reporting summary
Further information on research design is available in the Nature Portfolio Reporting Summary linked to this article.

## Results
### Safety and viability of the study device
We tested the primary outcomes related to the safety and viability of the study device. To date, there have been no adverse events related to the study device or study participation, and the participant has consented to continue the study. At this time, the device has been implanted for >2 years with good recording viability (Supplementary Note 1) and continues to be used for research purposes. Therefore, the primary outcomes related to safety and viability of the study device have been achieved. The results of the primary outcome related to BCI functionality are described in the remainder of the Results section.

### Long-term usage with a fixed click detector
The participant used the fixed click detector to effectively control a switch scanning application for a total of 626 min spanning a 90 day period that started on Day +21 after the completion of training data collection (Fig. 2b). We found that the performance of the click detector remained robust over a period of 111 days.

### Switch scanning performance
With the switch scanning medical communication board, the click-detector achieved 93% sensitivity (percentage of detected clicks per attempted grasps) with a median latency of 1.23 s from movement onset to on-screen click (visual feedback on the user interface) using a 10-vote threshold. No false positives were detected.

Using the switch scanning speller application (from Day +46 to Day +111), the click detector achieved a median detection sensitivity of 94.9% (95% CI [89.4, 95.5]) using a 7-vote threshold, and an increased sensitivity of 97.8% (95% CI [94.0, 97.7]) when using a 4-vote threshold ($W = -1.898$, $P = 0.057$, two-sided Wilcoxon Rank-Sum test; Fig. 4a). Offline classification accuracies across all spelling blocks where a 7-vote and 4-vote threshold were used were 90.8% and 93.6%, respectively (Supplementary Fig. 10). The median true positive frequency (TPF) was 10.671 per min (95% CI [10.07, 10.95]) using a 7-vote threshold, which improved to 11.596 per min (95% CI [11.13, 11.75]) when using a 4-vote threshold ($W = -2.782$, $P = 0.005$, two-sided Wilcoxon Rank-Sum test; Fig. 4b); the median false positive frequency (FPF) was 0.029 (95% CI [0.04, 0.24]) per min (1.74 per h) using a 7-vote threshold and 0.101 (95% CI [0.09, 0.30]) per min (6.03 per h) when using a 4-vote threshold ($W = -1.280$, $P = 0.200$, two-sided Wilcoxon Rank-Sum test; Fig. 4b).

As expected, we observed a decrease in latency from movement onset to algorithmic detection and to on-screen click after switching from the 7-vote to the 4-vote threshold (Fig. 4c). Using the 7-vote threshold, the median detection latency was 0.75 s (95% CI [0.73, 0.77]) and significantly dropped to 0.48 s (95% CI [0.46, 0.49]) using the 4-vote threshold ($W = 2.496$, $P = 0.013$, two-sided Wilcoxon Rank-Sum test). Meanwhile the median on-screen click latency was 0.93 s (95% CI [0.86, 0.94]) using the 7-vote threshold and dropped to 0.68 s (95% CI [0.65, 0.69]) using the 4-vote threshold ($W = 3.576$, $P = 3 \times 10^{-4}$, two-sided Wilcoxon Rank-Sum test). The delay between algorithmic detection and on-screen click was consistently ~200 ms, due to network and computational overhead.

Consequently, the participant was able to achieve high rates of spelling (Fig. 4d, e). Specifically, the median spelling rate was 9.1 (95% CI [7.59, 9.45]) correct characters per minute (CCPM) using the 7-vote threshold, which significantly improved to 10.2 (95% CI [9.66, 10.71]) CCPM using the 4-vote threshold ($W = -2.163$, $P = 0.031$, two-sided Wilcoxon Rank-Sum test). The wrong characters per minute (WCPM) rate was low at 0.2 (95% CI

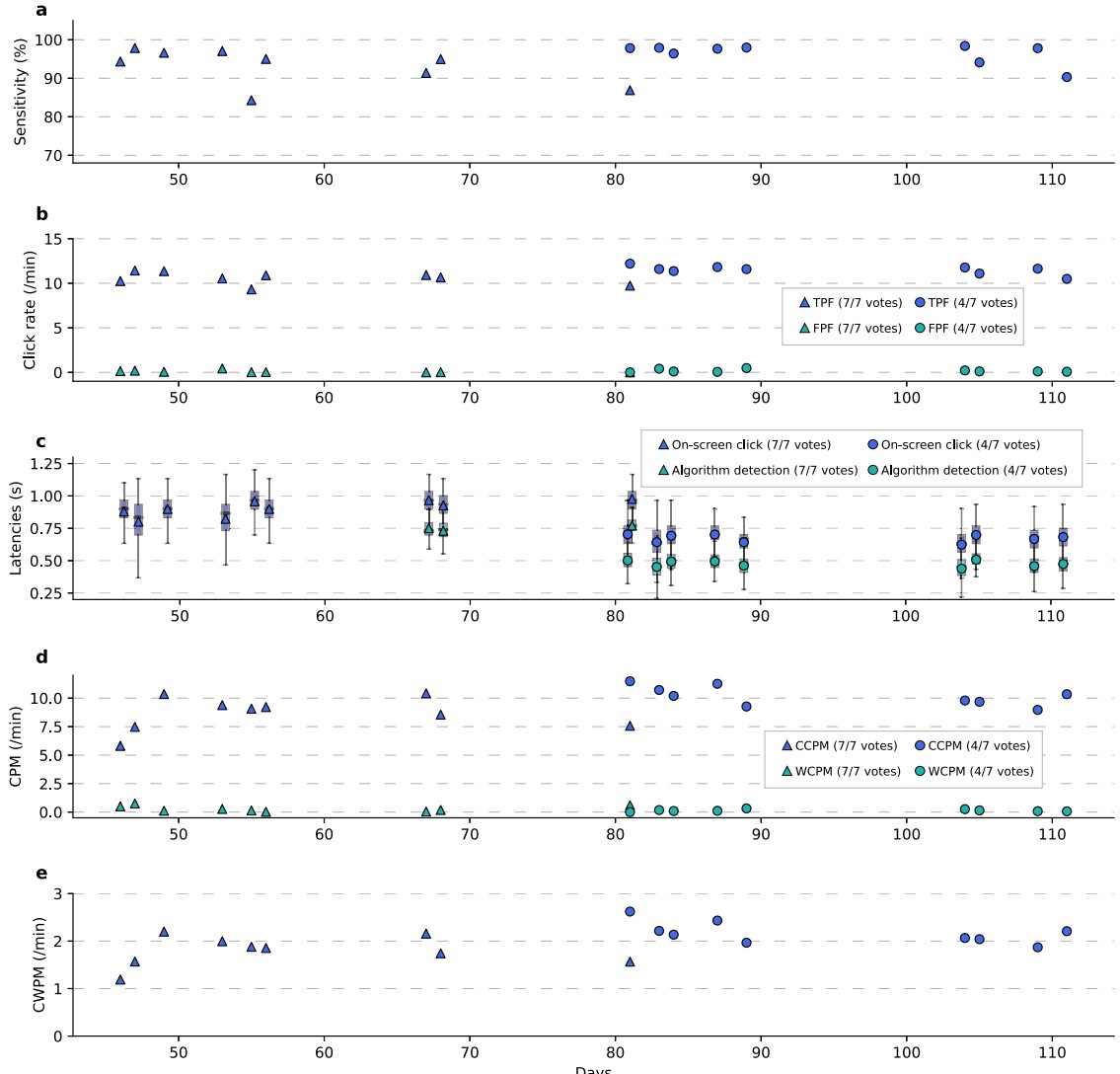

**Fig. 4 | Long-term switch scan spelling performance.** Across all subplots, triangular and circular markers represent metrics using a 7-vote and 4-vote voting threshold, respectively. **a** Sensitivity of click detection for each session. **b** True positive and false positive frequencies (TPF and FPF, which are represented by blue and green markers, respectively) were measured as detections per minute. **c** Latencies of grasp onset to correct algorithm detection (green markers) and on-screen click (blue markers). For each of the sessions using a 7-vote threshold, there were 284, 182, 372, 264, 451, 382, 233, 453, and 292 latency measurements, respectively. For each of the sessions using a 4-vote threshold, there were 135, 513, 591, 209, 289, 547, 511, 579, and 466 latency measurements, respectively. Mean latencies are shown as triangular or circular markers that are overlayed on top of box-and-whisker plots. The distribution of latencies across all spelling blocks in a session was used to compute the mean latency and box-and-whisker plot for that session. For each box-and-whisker plot, the median is shown as the center line, the quartiles are shown as the top and bottom edges of the box, and the whiskers are shown at 1.5 times the interquartile range. Using 7-vote and 4-vote voting thresholds, on-screen clicks happened an average of 207 ms and 203 ms, respectively after detection. Note that algorithmic detection latencies were not registered in the first six sessions. **d** Characters per minute (CPM) are assessed in terms of correct and wrong characters per minute (CCPM and WCPM, which are represented by blue and green markers, respectively). **e** Correct words per minute (CWPM).

[0.16, 0.48]) using the 7-vote threshold and remained low at 0.1 (95% CI [0.08, 0.21]) after switching to the 4-vote threshold ($W = 1.192$, $P = 0.233$, two-sided Wilcoxon Rank-Sum test). Additionally, the participant achieved 1.85 (95% CI [1.57, 1.97]) correct words per minute (CWPM) using the 7-vote threshold, which significantly improved to 2.14 (95% CI [2.05, 2.35]) CWPM using the 4-vote threshold ($W = -2.428$, $P = 0.015$, two-sided Wilcoxon Rank-Sum test). In one session, the participant achieved a spelling rate >11 CCPM with the 4-vote threshold, which to our knowledge is the highest spelling rate achieved using single-command BCI control with a switch scanning speller paradigm. Note that the CCPM for both voting conditions was lower than the respective TPFs, most likely because two true positive detections were necessary for one correct button click (be it a letter on the static keyboard or predicted letter or word).

## Simulation performance

The six click detectors trained on 50, 100, 150, 300, 390, and 480 trials achieved simulated median sensitivities of 84.3% (95% CI [79.8, 87.6]), 91.4% (95% CI [86.5, 92.7]), 93.5% (95% CI [91.5, 96.7]), 87.16% (95% CI [84.1, 92.3]), 95.8% (95% CI [91.6, 96.3]), and 95.8% (95% CI [92.0, 96.7]) and false positive frequencies (FPFs) of 0.154 (95% CI [0.10, 0.26]), 0.221 (95% CI [0.21, 0.43]), 0.321 (95% CI [0.23, 0.96]), 0.177 (95% CI [0.12, 0.30]), 0.195 (95% CI [0.14, 0.32]), and 0.096 (95% CI [0.10, 0.28]) per min respectively, (Supplementary Fig. 11). The simulated median sensitivity and FPF of any of the click detectors trained on 100, 150, 300, or 390 trials were not significantly different from the simulated median sensitivity and FPF of the click detector trained on all the original training data (480 trials) (for all comparisons $P > 0.05$, two-sided Wilcoxon Rank-Sum test). For the click

detector trained on 50 trials, the simulated median sensitivity was lower ($W = -3.135$, $P = 0.002$, two-sided Wilcoxon Rank-Sum test with 5-way Holm-Bonferroni correction) but the simulated median FPF was not.

We compared the simulated performance metrics (sensitivity, TPF and FPF) of the original fixed click detector to simulated metrics from click detectors with updated models trained on data from all preceding spelling blocks (Supplementary Fig. 12). The simulated median detection sensitivity of these updated click detectors was 99.1% (95% CI [98.4, 99.1]), which was higher than the 97.3% (95% CI [91.7, 97.0]) simulated sensitivity of the original fixed detector ($W = 3.098$, $P = 0.002$, two-sided Wilcoxon Rank-Sum test). Correspondingly, the simulated median TPF of the updated click detectors was 11.711 (95% CI [11.67, 11.84]) per min, slightly higher than the 11.574 (95% CI [10.78, 11.55]) per min simulated TPF of the original fixed detector ($W = 2.468$, $P = 0.014$, two-sided Wilcoxon Rank-Sum test). However, the simulated FPF of the updated click detectors was 0.641 (95% CI [0.78, 6.78]) per min (38.46 per h), higher than the 0.157 (95% CI [0.10, 0.30]) per min (9.42 per h) simulated FPF of the original fixed detector ($W = 2.941$, $P = 0.003$, two-sided Wilcoxon Rank-Sum test).

### Click detector retraining due to transient performance drop

On Day +118 (Supplementary Fig. 13 for timeline), the detector sensitivity markedly decreased (Supplementary Fig. 14), which was likely due to a decrease in the movement-aligned event-related synchronization (ERS) of the HG response across a subset of channels (Supplementary Fig. 15). Conversely, we found an increase in the event-related desynchronization (ERD) of low frequency power (10-30 Hz) (Supplementary Fig. 16). We found no hardware or software causes for the observed deviations in HG responses. Moreover, the participant had no subjective change in strength, no changes on detailed neurological examination or cognitive testing, and no new findings on brain computerized tomography images.

To ensure that BCI performance was not permanently affected, we retrained and tested a click detector with the same model architecture using data collected roughly 4 months after the observed performance drop (Supplementary Note 5). The new click detection algorithm used a total of 15 min of training data, which was all collected within one day, six days before BCI use (Day 309 post-surgical implantation, Supplementary Figs. 13, 17a); afterward, the model weights remained fixed. To determine the optimal voting threshold for long-term use, we additionally evaluated online click performance using all voting thresholds from 2 to 7 votes with this new click detection algorithm (Supplementary Fig. 18).

The participant used this retrained fixed click detector for a total of 428 min during six sessions spanning a 21 day period after retraining (Supplementary Fig. 17b). The optimal combination of sensitivity and false detections (Supplementary Note 6) was achieved using a 6-vote threshold. Using this threshold, we achieved similar performance metrics to those from the original click detector with a 4-vote threshold, namely a median detection sensitivity of 94.8% (95% CI [91.6, 95.9]), median TPF and FPF of 11.3 (95% CI [10.98, 11.84]) per min and 0.20 (95% CI [0.03, 0.53]) per min respectively, and a median CCPM, WCPM, and CWPM of 10.1 (95% CI [9.02, 10.62]), 0.1 (95% CI [0.08, 0.55]) and 2.2 (95% CI [1.95, 2.41]), respectively (for all comparisons $P > 0.05$, two-sided Wilcoxon Rank-Sum test) (Supplementary Fig. 19). Expectedly the median on-screen click latency was 0.86 s (95% CI [0.86, 0.91]), roughly 200 ms higher compared to the previous 4-vote threshold, due to the two extra votes required for generating a click ($W = 3.181$, $P = 10^{-3}$, two-sided Wilcoxon Rank-Sum test).

We additionally evaluated the participant's subjective cognitive workload of switch scan spelling with the click detector using the NASA-TLX iOS application. Across the six sessions using the retrained click detector the participant reported scores of $7.5 \pm 2.7$ (mean ± standard deviation) for mental demand, $8.3 \pm 2.6$ for physical demand, $5.8 \pm 3.7$ for temporal demand, $6.7 \pm 5.2$ for performance, $6.7 \pm 5.2$ for effort, and $6.7 \pm 2.6$ for frustration. These low scores indicate that the participant did not have difficulty in controlling the switch scanning speller application via click-detection.

### Channel contributions to grasp classification

To assess which channels produced the most important HG features for classification of attempted grasp, we generated a saliency map across all channels used to train the original model (Fig. 5a). As expected, channels covering cortical face region were generally not salient for grasp classification. The channel producing the most salient HG features was located in the upper-limb area of somatosensory cortex (channel 112, Fig. 5a), with a saliency value 55% and 88% higher than the next two most salient channels (channels 108 and 118), respectively (Supplementary Fig. 20a). Indeed, prior to the observed performance drop, this channel had a relatively amplified spectral response compared to other channels during attempted grasp (Supplementary Figs. 2, 3, 15). We then computed the corresponding offline classification accuracy of cross-validated models using the original model architecture for comparison to accuracies of models using a model architecture without channel 112 and an architecture using channels only over cortical hand-knob (see *Channel contributions and offline classification comparisons*); the mean accuracy using the original model architecture with repeated 10-fold cross-validation (CV) was 92.9% (Fig. 5b).

To check that the high online performance was not entirely driven by channel 112, we computed the offline classification accuracy of cross-validated models trained on HG features from all other channels. As expected, this model architecture relied strongly on channels covering the cortical hand-knob region (Fig. 5c), and notably was not as dependent on a single channel; the saliency of the most important channel (channel 108) was only 23% and 60% larger than the next two most salient channels (channels 118 and 110), respectively (Supplementary Fig. 20b). The offline mean classification accuracy from repeated 10-fold CV was 91.7% (Fig. 5d), which was not significantly lower compared to the mean accuracy using all channels ($W = 1.814$, $P = 0.139$, two-sided Wilcoxon Rank-Sum test with 3-way Holm-Bonferroni correction, Fig. 5g).

As channels covering the cortical hand-knob region made relatively larger contributions to decoding results, we investigated the classification accuracy of a model trained on HG features from a subset of electrodes covering only this region (Fig. 5e). Saliency values followed a flatter distribution; the saliency of the most important channel (channel 108) was only 21% and 44% larger than the next two most salient channels (channels 118 and 110), respectively (Supplementary Fig. 20c). Though the offline mean classification accuracy from repeated 10-fold CV remained high at 90.4% (Fig. 5f), it was slightly lower compared to the mean accuracy using all channels ($W = 2.800$, $P = 0.015$, two-sided Wilcoxon Rank-Sum test with 3-way Holm-Bonferroni correction, Fig. 5g). This suggests that a model trained on HG features from only the cortical hand-knob could still produce effective click detection, but parameters used for data labeling, model training, and post-processing may need to be more thoroughly explored to optimize click performance.

Finally, we assessed which channels produced the most important HG features for attempted grasping during the period with the performance drop (Supplementary Note 7). Despite lower performance (Supplementary Fig. 14), relative channel contributions remained largely the same (Supplementary Fig. 21) with the saliency value from channel 112 being 30% and 42% larger than the next two most salient channels (channels 108 and 118), respectively. Indeed, despite the drop in the movement-aligned HG responses across many channels, these relative saliency values suggest that some structure of channel importance was conserved.

## Discussion

In this study we demonstrated that a clinical trial participant with ALS was able to use a fixed click detector trained with a limited multichannel electrocorticographic (ECoG) dataset to generate stable online clicks over a period of three months. Using this click detector the participant formed sentences using a switch scanning speller application coupled with a language model. Our detector's high sensitivity, low false positive frequency, and minimal latency between onset of attempted grasp and algorithmic click detection allowed him to quickly and reliably spell sentences over several months without retraining the model.

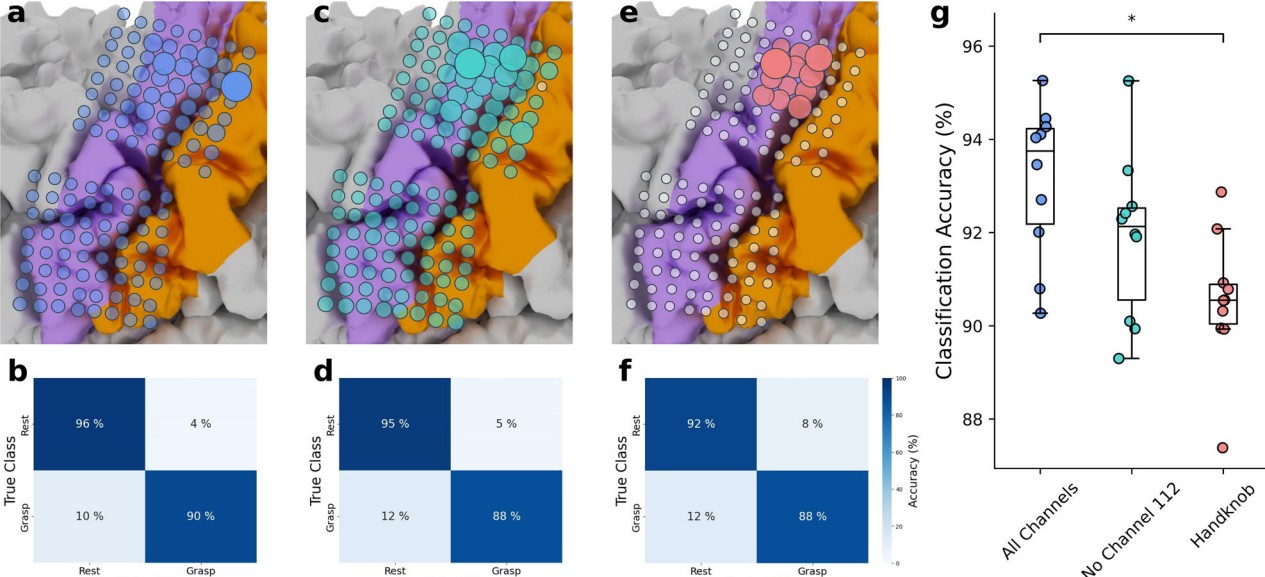

**Fig. 5 | Channel importance for grasp classification.** Saliency maps for the model used online, a model using HG features from all channels except from channel 112, and a model using HG features only from channels covering cortical hand-knob are shown in (**a**), (**c**) and (**e**), respectively. Channels overlaid with larger and more opaque circles represent greater importance for grasp classification. White and transparent circles represent channels which were not used for model training. Mean confusion matrices from repeated 10-fold CV using models trained on HG features from all channels, all channels except for channel 112, and channels covering only the cortical hand-knob are shown in (**b**), (**d**), and (**f**), respectively. **g** Box and whisker plot showing the offline classification accuracies from 10 cross-validated testing folds using models with the above-mentioned channel subsets. Specifically, for one model configuration, each dot represents the average accuracy of the same validation fold across 20 repetitions of 10-fold CV (see Channel contributions and offline classification comparisons). Offline classification accuracies from CV-models trained on features from all channels were statistically higher than CV-models trained on features from channels only over cortical hand-knob (* $P = 0.015$, two-sided Wilcoxon Rank-Sum test with 3-way Holm-Bonferroni correction).

A substantial barrier to the use of BCI systems by clinical populations outside of the laboratory is that users must often undergo an extensive period of training for optimizing fixed decoders[1], or daily model retraining or updating[34]. For example, reliable switch scan spelling was demonstrated for up to 36 months using a fixed decoder but required several months of data collection to optimize parameters for inhibiting unintentional brain clicks[1,18]. However, our click detector's long-term performance with a relatively small training dataset suggests a potentially reduced need for model optimization using ECoG signals with higher spatial density. Similarly, an endovascular electrode stent-array was recently used to train an attempted movement detector[3]. Though this is an extremely promising BCI technology for click decoding, the anatomical constraints on the number of electrodes and their proximity to motor cortex limit scaling to more complex BCI commands[34,50,51]. The device used in this study may have included more electrodes over upper-limb cortex than was necessary for click detection, but it allowed us to explore the upper bounds of click performance that might be expected from an ECoG array implanted for decoding attempted speech or complex upper limb movements. A post-hoc analysis suggested that similar performance could be obtained using a smaller grid confined to the hand knob region of motor cortex.

Our click detector performed with high sensitivity and low false positive frequency (FPF). The high sensitivity was likely attributable to the large increases in high gamma (HG) modulation during attempted movement compared to rest or baseline conditions. The voting window was a simple yet effective heuristic strategy for inhibiting false detections; post-hoc analysis of online performance revealed that false detections particularly increased when less than three votes were required for generating a click. The participant preferred an increased sensitivity at the expense of a slight increase in false detections. This illustrates the utility of the voting window as an adjustable parameter and the importance of allowing BCI users to fine-tune parameters according to their preferences, which may vary substantially among users and among different click-based applications.

The robust changes in HG modulation likely contributed to the high simulated performance of the click detectors that were trained on most of the subsets of the original training data. Nonetheless, the results of our simulated model updates suggest that periodically updating fixed models with recent training data may enable higher sensitivity. The concurrent increase in FPF was likely due to training on false positive-inducing features that occurred independently of attempted grasping and were consequently labeled as rest. Though recent advances in unsupervised label correction have been primarily used for online retraining of speech models[16], it may be possible to apply analogous methods to click detector outputs for relabeling such false positive-inducing features.

Our results improve upon the previous switch scanning performance reported by Vansteensel et al.[1]. In that study a participant with ALS was implanted with four contacts over hand motor cortex and achieved a spelling rate of 1.8 letters per minute with a latency of at least 1 s per click (compared to a spelling rate of 10.2 CCPM and 0.68 s latency reported in the present work). The authors were not able to measure the latency from movement attempts to click detection due to the locked-in state of their participant. Because their click detector required five consecutive 200 ms epochs with neural features exceeding a pre-determined threshold, their latency could not have been <1 s. These results may have been limited by lower sensitivity for high frequency activity, a limited number of ECoG channels (one bipolar derivation), and a 5 Hz rate for transmitting power values (constrained by energy consumption of wireless signal transmission), see Vansteensel et al.[1]. Although spelling rates were slightly lower than those observed in Oxley et al.[19] (14–18 CCPM) the participants in that study used eye-tracking (ET) to first navigate to the appropriate letter before clicking it. Additionally, though the same group reported accuracies of 97.4% and ~82% in selecting one of five targets (without ET), these corresponded to relatively longer latencies of 2.5 s and 0.9 s, respectively[3].

Our spelling rates were comparable to those from other clinical populations who have used switch scanning keyboards by leveraging residual movements (and without a BCI), including people living with ALS[52] or

other causes of motor impairments[53]. Integration of eye-tracking with click decoding may enable even faster user interface navigation and spelling rates[3,54]. Meanwhile, non-invasive BCIs using visually evoked potentials are estimated to produce comparable spelling rates[55] and can potentially be trained with little or no neural data[56]. BCIs based on P300 spellers have achieved a wide range of accuracies (65–100%) and have achieved spelling rates of 1.2–6 CCPM for people with ALS[57–60], while typing speeds of 17 characters per min using eye-tracking alone have been reported[61]. However, control strategies based on eye movements may cause eyestrain during long periods of use[27–29] and worsen as residual eye movements can deteriorate in late-stage ALS[30–33].

After nearly 4 months without retraining or updating our model, we observed a drop in BCI performance. This was caused by a modest decrease in the modulation of upper-limb HG power in several electrodes over hand area of sensorimotor cortex. This decrease was especially pronounced in the most salient channel used to train the original detector, so it was not unexpected that BCI performance was affected. There were no accompanying new neurological symptoms or changes in cognitive testing, nor any evidence of adverse events or device malfunction. Variations in signal amplitude and spectral energy similar to those we observed in the participant have been reported in ECoG signals recorded for several years by the Neuropace (TM) RNS system[62]. However, the RNS system has been used in patients with epilepsy, not people living with ALS, for whom there is scarce data on long-term ECoG. We are aware of only one such study[18], but this study did not report signal characteristics on the granular timescale necessary for comparison with our results. By retraining the model with a similar workflow, but with even less data, we achieved equally robust performance in subsequent testing sessions, suggesting that long-term discernability of HG activity was not affected.

There were several limitations to this study. Though the participant retained the ability to perform partial grasping movements, additional work is needed to determine how performance of a rapidly trained click detector would generalize to individuals with more severe movement impairments. Click detection using signals from less affected regions of the cortex and control strategies that are not based on attempted movement, but rather on imagined movements or responses to sensory input could serve as alternative control strategies. Additionally, we did not explore the utility of low frequency power suppression, which has been stable and useful in other studies[60,61]. It is possible that by optimizing our training paradigm to minimize anticipatory low-frequency ERD and by appropriately labeling rebound ERS for model training, we could further improve the robustness of our click detector. Spelling rates were likely hindered by preparatory periods (Supplementary Note 8), and by the divergent linguistic statistics between the Harvard sentence prompts and the language model used for letter and word-autocompletion. We expect that spelling rates would improve with free-form spelling and a language model fine-tuned to the BCI user's preferences. Furthermore, regular user-specific model updates may better optimize the balance between independent long-term BCI use and technician intervention, which is particularly relevant for home-use.

This study adds to the growing literature demonstrating the functional utility of ECoG-based BCI implants. Robust click-detection complements recent major advancements in online spelling[2,4] and speech decoding[63,64] by providing a more application-agnostic interface. Click detectors, in addition to their utility as a communication tool, enable navigation of menus and accessibility software such as web-browsers, internet of things (IoT), and multimedia platforms, and thus merit further investigation.

## Data availability
Source data for rendering the figures in the main text and supplementary information is publicly available at https://doi.org/10.17605/OSF.IO/46SKB[65]. Beginning immediately after publication, neural data from the study participant and the study protocol will be available from the corresponding author upon reasonable request.

## Code availability
The analytical code for rendering the figures in the main text and supplementary information is publicly available at https://doi.org/10.17605/OSF.IO/46SKB[65]. Code used for offline model development and post-hoc analysis is also included. Model training and offline analysis were done using Python (version 3.9.13). The recurrent neural network was built using Keras with a TensorFlow backend (version 2.8.0). Real time decoding was done in Python (version 3.10.12) using the ezmsg[42] messaging architecture (version 3.0.0).

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

## Acknowledgements

The authors wish to express their sincere gratitude to participant CC01, whose dedication and efforts made this study possible. The authors thank and Chad Gordon for medical and surgical support during device implantation; the ALS clinic at the Johns Hopkins Hospital and the Johns Hopkins ALS Clinical Trials Unit for their consultations on ALS-related topics and care of the participant; Mathijs Raemaekers for analysis of fMRI data; Yujing Wang for 3D reconstruction of the brain MRI and electrode coregistration with the post-operative CT. The authors were supported by the National Institutes of Health under award number UH3NS114439 (NINDS).

## Author contributions

D.N.C, M.S.F., and N.E.C. conceived and designed the experiments. D.N.C. designed and implemented the detection algorithm and the neural classifier. D.N.C., M.A., G.W.M, S.L., and C.C. implemented the online decoding pipeline. D.N.C., S.S., and C.C. designed the user interfaces. D.N.C., S.L., and Q.R. contributed to data collection. D.N.C. and S.S. analyzed the data. D.N.C, S.S., and B.A.W. prepared data visualizations. D.N.C., S.L., S.S., M.A., Q.R, M.S.F., and N.E.C. contributed to study methodology. M.V., F.V.T., N.F.R. M.S.F., and N.E.C. contributed to patient recruitment and regulatory approval. N.J.M., L.C. and A.U. contributed to the clinical care of the participant. W.S.A., K.R.R., and N.E.C. planned and performed device implantation; D.N.C., S.L., B.A.W., F.V.T., N.F.R., and M.S.F. were also involved in surgical planning and intraoperative functional mapping. K.C.N. recorded and analyzed fMRI data for informing device implantation. N.F.R. and N.E.C. obtained funding. N.E.C. supervised the study. D.N.C., M.S.F., and N.E.C. prepared the manuscript. All authors reviewed and revised the manuscript.

## Competing interests

The authors declare no competing interests.
