## [Peer Review File · Communications Medicine]

Reviewers' comments:

Reviewer #1 (Remarks to the Author):

I thank the authors for their great contribution to the BCI literature on ECoG-based selection decoding in an individual with ALS. The novelty of this proposal is high, and demonstrates improvements from previous efforts which by using a more extensive cortical grid and achieving higher communication rates.

This paper describes the implant of two ECoG grids in an individual with ALS, who, over the course of about a year trains and eventually uses the device to operate a switch-scanning interface. The authors describe the longitudinal performance of the system that used two different scanning interfaces, as well as the longitudinal performance, and the adaptations made over the course of the study. I think it is an instructive piece of work and sets the stage for future exciting results out of this group.

I have a few major concerns which I would like addressed, in addition to multiple other comments I have included in a revised manuscript and supplementary materials.

1) Better description is needed of the participant in terms of established clinical measures, such as ALSFRS-R scores. Which cognitive and neurological tests, mentioned first in the results section on decoder retraining, were performed?

2) The manuscript contains some blocks of text which this reviewer finds unnecessary for the understanding of the manuscript, or which better belong in other locations.

a) Most of the last paragraph of the introduction belongs in the results and methods.

b) Most of the first paragraph of the methods, along with Supplementary notes 1 and 4 can be excluded.

c) Related to b), the participant paragraph of the methods tries to make statements related to the whole CortiCom clinical trial, which I do not think are necessary for this paper.

d) others noted in attached manuscript.

3) Cue-aligned vs trial-aligned HG power. The description of this epoch shifting/time warping was not well understood by me. Also, the need for performing this was not well described. Did decoder performance suffer if this shifting of training epochs was not performed? Better description needs to be given of how shifting/warping occurred, what extent of shifting/warping was performed, and how this affected classification performance. Other than the technical demonstration that shifting trials to match the peak of each other increases the correlation between them, I don't think Supplementary figures 4 & 5 are useful.

3.1) My understanding is that video was not used to capture movement onset and offset required to perform shifting. Why?

4) Some discussion is warranted for why no attempt was made at performing imagination of hand grasp to determine similarities and differences with actual hand grasp in terms of cortical representation and performance.

Reviewer #2 (Remarks to the Author):

Dear Nathan Crone and colleagues,

With great interest I read your manuscript entitled "A click-based electrocorticographic brain-computer interface enables long-term high-performance switch-scan spelling". This research demonstrates the effective and long-term use of a brain-controlled communication aid that allowed a person with amyotrophic lateral sclerosis (ALS) to select letters or icons on a computer screen. Novel features of this research include training a classifier on a limited amount of training data (44 minutes of data were acquired across 4 different days) and testing it across a period of 3 months (18 different test days) after without retraining the classifier. The use of chronically implanted ECoG is also relatively new in the field of brain-computer interfacing (BCI).

The achieved classification speed and performance outperform those of traditional P300 spellers and are comparable to those of other BCI spelling applications. For example, Sutter (1992) achieved a performance of 10-12 bits (full words or characters) per minute using a visual ECoG-driven brain-computer interface. Later on, this same approach based on code-modulated visual

evoked potentials (cVEP) was effectively demonstrated in a group of people with ALS using EEG, also yielding average speeds of 10 characters per minute (Verbaarschot et al., 2021).

Demonstrations of effective BCI performance within the target population are rare. Even though the current manuscript reports results of one participant only, this relatively long term study is a relevant contribution to this field.

Major remarks:

1) The advantage of using invasive neuroimaging methods such as ECoG are not immediately clear to me. Comparable results have been achieved using non-invasive methods such as EEG. Could you elaborate what benefits the use of ECoG provide over non-invasive methods? Moreover, could you compare the results of this study to other commonly available communication aids that a person with ALS may use, such as eye-tracking or applications that are directly controlled by residual movement? Does an invasive BCI have a clear advantage over these less invasive and less expensive methods? Since your approach currently relies on residual movement, the direct benefit is not clear to me.

2) I miss information on the opinion of the user in this manuscript. What did the participant think of the spelling application? Did they ever use it autonomously? Would they want to use it in their daily life? Did they enjoy using the application? Or was it frustrating?

3) It is not clear to me why a relatively complex decoding strategy was used for this application. You are trying to decode grasp intentions vs. rest, which induces relatively large changes in high gamma power activity. It seems to me that a simple linear decoder may also do this job. What were the reasons for using a non-linear neural network approach instead? Moreover, specific design choices for the network are not well argued (number of layers, number of units per layer). Could you elaborate your design choices in the manuscript?

4) ALS affects motor control and is reflected by changes in motor areas in the brain. Why do you choose to use a movement for BCI control and record from motor areas in the brain as both these things are likely affected as ALS progresses? Would another, less affected brain area not be a better recording site? And would a task that the participant can always perform with ease (response to sensory input, mental calculation, imagined sensations) not be a better and more pleasant task for them?

5) The main novelty of this paper is the fact that the classifier is trained on limited test data, and used across a large period of time without any retraining. However, non-invasive methods such as the c-VEP based BCI speller (Verbaarschot et al., 2021) do not require any training data. What are the benefits of your approach to such spellers?

Minor remarks:

(1) Line 36: please clarify what happened when you re-trained the classifier. You collected new training data on that day and tested in the following 21 days? Or did you train and test on the same day?

(2) Line 50: Add that the training data was recorded on different days than you tested. For example: "Our algorithm was trained using less than one hour's worth of brain signals recorded several days prior to testing the algorithm. The algorithm then performed reliably for a period of three months without any retraining."

(3) Line 62-63: you comment on the use of micro-electrodes on the long term, but micro-electrode arrays have been shown to record reliably for up to 7 years in a row (Hughes et al., 2020)! In addition, deep-brain stimulation electrodes have been shown to reliably perform for extended periods of time too.

(4) Line 67: please define what you mean by "chronic". How long does the implant need to exist for it to become "chronic"?

(5) Line 148-149: what task did the participant perform to assess the relevant cortical areas for implantation?

(6) Figure 1: can you increase the size of subfigure h? It is a bit small to see clearly

(7) Line 188: the ISI is chosen between a lower and upper bound. What were these bounds?

(8) Line 236: what does "randomly downsample" mean? How can the rest class be overrepresented if you have balanced rest and grasp classes?

(9) Line 283: why a lock-out period of 1 second? Why not longer or shorter?

(10) Line 347: please include your reasoning for choosing specific voting thresholds. This information is provided in Supplementary Note 6. I would include this in the main paper.

(11) Lines 413-420: this seems more Methods than Results. It feels like this is a repetitions of information that I already read. Maybe only mention this in the Methods?

(12) Figure 2: please increase the font size on the x and y axes.

(13) Figure 4: please increase the font size of b, d, and f.

(14) Please include Supplementary figures 8 and 9 in the main text. These depict your main experimental task and belong in your main text.

References:

Sutter, E. E. (1992). The brain response interface: communication through visually-induced electrical brain responses. *Journal of Microcomputer Applications*, 15(1), 31-45.

Verbaarschot, C., Tump, D., Lutu, A., Borhanazad, M., Thielen, J., van den Broek, P., ... & Desain, P. (2021). A visual brain-computer interface as communication aid for patients with amyotrophic lateral sclerosis. *Clinical Neurophysiology*, 132(10), 2404-2415.

Hughes, C. L., Flesher, S. N., Weiss, J. M., Downey, J. E., Collinger, J. L., & Gaunt, R. A. (2020). Neural stimulation and recording performance in human somatosensory cortex over 1500 days. *medRxiv*, 2020-01.

Reviewer #3 (Remarks to the Author):

This study demonstrates an ECoG-based single click decoder that has long-term stability for over three months, trained on very small amount of neural data. The authors address a fundamental requirement of stable decoder performance over long periods without recalibration for practical BCI applications, leveraging the well-known stability of ECoG signals. The manuscript is clearly presented. I have some comments, questions and suggestions regarding framing of the problem in the context of current click BCIs, exploration of some results and limitations of this approach.

Click BCI is not a new concept, as authors have discuss, however, I feel that the novelty of this particular work could be fleshed out more in comparison to the merits of ECoG as a stable neural signal. Authors point out in introduction (line 62) that MEA BCIs can have sophisticated capabilities but use outside research environment is limited. However, this is true of all types of current BCIs. Also, long-term safety and efficacy of MEAs in humans have been demonstrated consistently for two decades, and is not just the characteristic of ECoG (line 65). New research in MEA BCIs show that frequent decoder recalibration might not be required with advance online unsupervised training methods that run in the background during online BCI use (e.g., CORP) so this might not be a limiting factor. Regarding the amount of training data required, it is mentioned that stent-electrode based BCI studies also required 1 or more sessions of data (line 77) which indicates that training on small amounts of data from single session has been shown before with ECoG-like signals. In view of these arguments, I think the introduction should be framed to justify strengths of this work. Single command BCIs are common with EEG and have shown decent performance in various studies using different paradigms, hence this should also be discussed.

The stability of the decoder shown in the results is remarkable. It could be useful to show a plot with the accuracy of each real-time session (computed offline) using the most updated decoder trained on all available prior sessions data to assess the maximum performance (upper bound on the accuracy) the decoder could have had in comparison to the fixed decoder currently used.

In figure 3, please also show the accuracy of the system along with sensitivity as well as the confusion matrix of classification (like fig 4) for three groups: (1) cross-validated results on training sessions, (2) 7-vote real-time sessions and (3) 4-vote real-time sessions for complete understanding of the decoder performance.

Looking retrospectively, what is the minimum amount of data required to get the highest real-time accuracy? In other words, what would be the decoder accuracy as a function of the amount of training data.

It's not clear to me whether the correct characters per min and correct words per min reported in fig 3 are calculated for selecting single characters (number of actual clicks) or on the output of autocomplete, which would require fewer clicks but inflate the CCPM/CWPM.

Supplementary fig 2 shows very strong event-related desynchronization (ERD) in lower frequencies around beta band uniformly in majority of channels. Could this be used as more robust feature for click detection to supplement high gamma power? The robustness of ERD in EEG literature for movement detection is well known, which can also be consistently observed here. Could that be leveraged for better click classification and improved stability? It would be interesting to see if ERD also reduced after day 118 when HG was reduced. Why were the features restricted to HG?

Again, in supplementary figure 2, some ERD can be observed before go cue, is this due to anticipatory behaviour?

Why was RNN chosen for binary classification (instead of simply flattening the sequential features which can be sufficient for simple binary classification)? Did you try a simpler classifier e.g., logistic regression, SVM or LDA? These can be faster to train and require less data.

It would be helpful to readers to put your CCPM and CWPM in context of other similar BCI studies in the discussion. It will be useful to also compare the performance with similar EEG based systems. What advantage would this BCI system provide over a binary classification EEG BCI? How does it compare with P300 speller in terms of speed and accuracy?

Also, how does this BCI performance compare with other technologies/aids that you participant uses for communication in his daily life?

While single click decoding can be useful and robust, this is still not practical and very slow for use with the speed of ~2.2 words/min speed. Please discuss limitations of this approach. It seems to me that the bottleneck here is the switch scanning application used which traverses each row and column and pauses for 1s at each switch. Would a faster switch scanning application design (something similar to Dasher) improve the speed? Please discuss how can overall speed of the whole BCI system be improved? What computational overheads cause latency of 200ms for producing the click after its detection and can this be reduced?

And finally, would this decoder with HG features work robustly in cases where there is no residual arm/hand movement in a participant to perform actual hand movement for initiating clicks?

Minor: Which channel activity is shown in supplementary fig 12 b,c,d

Reviewer #4 (Remarks to the Author):

Summary

In this well written and interesting manuscript, the authors describe a brain-computer interface (BCI) that can translate cortical signals of a person with (incomplete) hand paralysis as they attempt to grasp their right hand into clicks, which are subsequently used to drive a switch-scan spelling interface to communicate in full sentences.

Importantly, they also show that the interface can achieve stable performance for months without recalibration.

The methods appear sound; however, some comments should be addressed during revision:

Comments

The sudden performance drop is perplexing. Do saliencies appear consistent before / after the performance drop?

Did the authors try other types of BCI control, such as cursor control or having, for example, 3 different hand target? Given the performance, it seems likely that 128 channels of ECoG could facilitate control over a richer output space to get faster communication rates. Some explanation for why the authors settled on a single type of output could be helpful.

In Figure 1, can the authors add some visual indication that the patient is attempting to grasp their hand? It would help convey the system use a little more clearly.

Can the authors comment on why they chose 110-170 Hz as high-gamma range? It seems that the range that groups use varies in the literature.

The Discussion spends a lot of time simply restating content from the Results. It is preferred for Discussions to instead frame and interpret the results, not simply repeat them. The Discussion could be heavily trimmed and condensed.

Title

A click-based electrocorticographic brain-computer interface enables long-term high-performance switch-scan spelling

Authors

Daniel N. Candrea¹, Samyak Shah², Shiyu Luo¹, Miguel Angrick², Qinwan Rabbani³, Christopher Coogan², Griffin W. Milsap⁴, Kevin C. Nathan², Brock A. Wester⁴, William S. Anderson⁵, Kathryn R. Rosenblatt⁶, Alpa Uchil², Lora Clawson², Nicholas J. Maragakis², Mariska J. Vansteensel⁷, Francesco V. Tenore⁴, Nicolas F. Ramsey⁷, Matthew S. Fifer⁴, Nathan E. Crone²

Department of Biomedical Engineering, Johns Hopkins University School of Medicine, Baltimore, MD

Department of Neurology, Johns Hopkins University School of Medicine, Baltimore, MD

Department of Electrical and Computer Engineering, Johns Hopkins University, Baltimore, MD

Research and Exploratory Development Department, Johns Hopkins University Applied Physics Laboratory, Laurel, MD

Department of Neurosurgery, Johns Hopkins University School of Medicine, Baltimore, MD

Department of Anesthesiology and Critical Care Medicine, Johns Hopkins University School of Medicine, Baltimore, MD

Department of Neurology and Neurosurgery, UMC Utrecht Brain Center, Utrecht, The Netherlands

Abstract

Background

Brain-computer interfaces (BCIs) can restore communication in movement- and/or speech-impaired individuals by enabling neural control of computer typing applications. Single command “click” decoders provide a basic yet highly functional capability.

Methods

We sought to test the performance and long-term stability of click-decoding using a chronically implanted high density electrocorticographic (ECoG) BCI with coverage of the sensorimotor cortex in a human clinical trial participant (ClinicalTrials.gov, NCT03567213) with amyotrophic lateral sclerosis (ALS). We trained the participant’s click decoder using a small amount of training data (<44 minutes across four days) collected up to 21 days prior to BCI use, and then tested it over a period of 90 days without any retraining or updating.

Results

Using this click decoder to navigate a switch-scanning spelling interface, the study participant was able to maintain a median spelling rate of 10.2 characters per min. Though a transient reduction in signal power modulation interrupted testing with this fixed model, a new click decoder achieved comparable performance despite being trained with even less data (<15 min, within one day).

Conclusion

These results demonstrate that a click decoder can be trained with a small ECoG dataset while retaining robust performance for extended periods, providing functional text-based communication to BCI users.

Plain Language Summary

People living with amyotrophic lateral sclerosis (ALS) struggle to communicate with family and caregivers due to progressive muscle weakness. This study investigated whether the brain signals of a participant with ALS could be used to control a spelling application. Specifically, when the participant attempted to make a fist, a computer algorithm detected increased neural activity from electrodes implanted on the surface of his brain, and thereby generated a mouse-click. The participant used these self-generated clicks to select letters or words from a spelling application to type sentences. Our algorithm was trained using less than one hour's worth of recorded brain signals and then performed reliably for a period of three months. This approach can potentially be used to restore communication to other severely paralyzed individuals over an extended period of time and after only a short training period.

Introduction

Brain-computer interfaces (BCIs) can allow individuals with a variety of motor impairments to control assistive devices using their neural signals¹⁻¹¹. In particular, implantable BCIs have the potential to provide higher performance compared to non-invasive BCIs and may provide round-the-clock availability. These capabilities are derived either from single neuron activity recorded by microelectrode arrays (MEAs), or from neural population activity recorded by macroelectrodes (typically consisting of electrocorticographic (ECoG) arrays on the cortical surface)¹². Although sophisticated capabilities and high performance of MEA BCIs have been reported, their use outside of research environments has been limited due to varying degrees of long-term signal attrition^{13,14} and day-to-day instability in decoding models trained on single neuron activity, often requiring frequent recalibration¹⁵. On the other hand, extensive safety and efficacy data from the use of chronic ECoG recordings for epilepsy management¹⁶ suggests that ECoG implants have the potential to deliver greater long-term signal stability. However, the utility of ECoG for chronically implanted BCIs has only been tested in a few participants.

In the first clinical trial of a chronic ECoG BCI¹, a participant with quadriplegia and anarthria due to amyotrophic lateral sclerosis (ALS) attempted hand movements to generate "brain clicks", in turn controlling a switch-scanning spelling application. These brain clicks were detected as spectral changes in ECoG signals recorded from a single pair of electrodes on the surface of hand area of contralateral motor cortex. Though the participant used these brain clicks to communicate in her daily life for more than 3 years¹⁷, several months of data collection were necessary for parameter optimization. In a separate clinical trial^{4,18}, participants with severe upper limb paralysis due to ALS or primary lateral sclerosis were implanted with an endovascular stent-electrode array and required 1-12 sessions of training with their brain click BCI before long-term use⁴. However, due to the location of electrodes in the superior sagittal sinus, the participants triggered brain clicks with attempted foot movements, which may not be intuitive for computer control. Moreover, device limitations in both clinical trials may have constrained brain click speed and overall performance of the BCIs. Vansteensel et al. reported 87-91% click accuracy (comprised of correctly detected and withheld clicks) with a 1 s latency¹ while Mitchell et al. reported ~82% accuracy with a 0.9 s latency or a 97% accuracy with a 2.5 s latency⁴.

In this study, we tested whether improved click performance could be achieved using high density ECoG recordings from sensorimotor cortex. We implanted two 8 x 8 ECoG grids (4 mm pitch, PMT Corp.,

Chanhasen, MN) over left hand and face cortical regions in a clinical trial participant with ALS (ClinicalTrials.gov, NCT03567213). The participant generated clicks using the implanted BCI to spell sentences at a significantly improved spelling rate compared to prior brain click work using a switch-scanning paradigm¹. Moreover, the participant achieved high click-detection accuracies with low false-positive rates and low latencies from attempted movement onset to click. We found that a fixed ECoG-based click decoder trained on a limited dataset maintained high performance over a period of several months without requiring re-training or daily model adaptation. Finally, offline analysis suggested that similar performance is achievable with a smaller number of ECoG electrodes over only the cortical hand-knob region.

[revised manuscript text omitted]

After computing each channel's trial-aligned HG power (-1 s to 2.5 s post-cue), we accounted for the inter-trial variability due to reaction delay by re-aligning each trial's HG power using a subset of highly activated channels²². This resulted in generally increased HG power correlations between trials (Supplementary Figs. 3-5). We visually determined the onset and offset of the re-aligned trial-averaged HG power from the channels used for re-alignment (Supplementary Fig. 6). The average neural activity onset and offset were manually estimated from the aligned neural data to be roughly 0.2 s and 1.2 s post-cue, respectively, with neural activity more clearly differentiating from rest activity starting at 0.3 s post-cue and ending at 1.1 s post-cue. We consequently assigned grasp labels to ECoG feature vectors falling between 0.3 s and 1.1 s post-cue for each trial, and rest labels to all other feature vectors. Since this overall strategy relies only on the visual inspection of neural signals, we believe it to be compatible with reduced availability of ground truth signals, like movement, as might be the case in locked-in participants.

Model architecture and training

We designed a recurrent neural network in a many-to-one configuration to learn changes in HG power over sequences of 1 s (Supplementary Fig. 7). Each 128-channel HG power vector was input into a long short-term memory (LSTM) layer with 25 hidden units for modelling sequential dependencies. From here, 2 consecutive fully-connected (FC) layers with 10 and 2 hidden units, respectively, determined probabilities of the rest or grasp class. The former utilized an eLU activation function while the latter employed softmax to output normalized probability values. In total, the architecture consisted of 17,932 trainable parameters, and was trained on a balanced dataset of rest and attempted grasping sequences by randomly downsampling from the overrepresented rest class.

[revised manuscript text omitted]

Reproducibility of experiments

Neural data collection and processing as well as decoder performance were reproducible across sessions as the participant was able to repeatedly demonstrate click control using neural signals from attempted hand movements to spell sentences. However, as this study reports on the first and only participant in this trial so far, further work will be necessary to test the reproducibility of these results in other participants.

Results

Long-term usage with a fixed click detector

The participant used the fixed click detector to effectively control a switch-scanning application for a total of 626 min spanning a 90-day period that started on Day 21 after the completion of training data collection (Fig. 2b). Specifically, we recorded one session with the medical communication board and 17 sessions with the spelling application. We defined Day 0 as the last session of training data collection. We used a voting threshold of 10/10 votes with the communication board. Using the spelling application, we initially used a voting threshold of 7/7 votes, but reduced this threshold to 4/7 votes on Day +81 as the participant reported that he preferred an increased sensitivity despite the resulting increase in false positive detections. We found that the decoder performance remained robust for 111 days.

Figure 2 | Long-term use of a fixed click detector. (a) Training data was collected during 4 sessions that occurred within a period of 15 days. For each day, each bar segment represents a separate block of training data collection (6 training blocks total). (b) Using the fixed decoder, one block of switch-scanning with the communication board was performed +21 days post-training data collection (purple). From Day +46 to Day +81, the fixed decoder was used for switch-scan spelling with a 7-vote threshold (blue). From Day +81 to Day +111, the fixed decoder was used for switch-scan spelling with a 4-vote threshold (teal). For each day, each bar segment represents a separate spelling block of 3-4 sentences. The horizontal axis spanning both (a) and (b) represents the number of days relative to the last day of training data collection (Day 0).

Switch-scanning performance

With the switch-scanning medical communication board, the click-detection model achieved 93% sensitivity (percentage of detected clicks per attempted grasps) with a median latency of 1.23 s from movement onset to on-screen click (visual feedback on the user interface) using a 10-vote threshold. No false positives were detected.

Using the switch-scanning spelling application (from Day +46 to Day +111), the click detector achieved a median detection sensitivity of 94.9% using a 7-vote threshold, and a sensitivity of 97.8% when using a 4-vote threshold ($P = 0.057$, Wilcoxon Rank-Sum test; Fig. 3a). The median true positive frequency (TPF) was 10.7 per min using a 7-vote threshold, which improved to 11.6 per min when using a 4-vote threshold ($P = 0.005$, Wilcoxon Rank-Sum test; Fig. 3b); the median false positive frequency (FPF) was 0.029 per min (1.74 per h) using a 7-vote threshold and 0.101 per min (6.03 per h) when using a 4-vote threshold ($P = 0.20$, Wilcoxon Rank-Sum test; Fig. 3b).

As expected, we observed a decrease in latency from movement onset to algorithmic detection and on-screen click when switching from the 7-vote to the 4-vote threshold (Fig. 3c). Using the 7-vote threshold, the median detection latency was 0.75 s and significantly dropped to 0.48 s using the 4-vote threshold ($P = 0.013$, Wilcoxon Rank-Sum test). Meanwhile the median on-screen click latency was 0.93 s using the 7-vote threshold and dropped to 0.68 s using the 4-vote threshold ($P = 3 \times 10^{-4}$, Wilcoxon Rank-Sum test). The delay between algorithmic detection and on-screen click was consistently ~200 msec, due to network and computational overhead.

Consequently, the participant was able to achieve high rates of spelling (Fig. 3d). Specifically, median spelling rate was 9.1 correct characters per minute (CCPM) using the 7-vote threshold, which significantly improved to 10.2 CCPM using the 4-vote threshold ($P = 0.031$, Wilcoxon Rank-Sum test). Similarly, he achieved 1.85 correct words per minute (CWPM) using the 7-vote threshold, which significantly improved to 2.14 CWPM using the 4-vote threshold ($P = 0.015$, Wilcoxon Rank-Sum test). In

one session, the participant achieved a spelling rate greater than 11 CCPM with the 4-vote threshold, which to our knowledge is the highest spelling rate achieved using single-command BCI control with a switch-scanning spelling paradigm.

Figure 3 | Long-term switch-scanning spelling performance. Across all subplots, triangular and circular markers represent metrics using a 7-vote and 4-vote voting threshold respectively. (a) Sensitivity of grasp detection for each session. Dashed line delineates 100% sensitivity. (b) True-positive and false-positive frequencies (TPF and FPF) measured as detections per minute. Dashed line delineates 0 FPF. (c) Average latencies with standard deviation error bars of grasp onset to algorithm detection and to on-screen click. The averages and standard deviations were computed from latency measurements across all spelling blocks from one session using the same voting threshold. Using 7-vote and 4-vote voting thresholds, on-screen clicks happened an average of 207 ms and 203 ms respectively after detection. Note that detection latencies were not registered in the first six sessions. (d) Correct characters and words per minute (CCPM and CWPM).

Decoder retraining due to transient performance drop

On Day +118 (Supplementary Fig. 10 for timeline), the detector sensitivity fell below the pre-set performance threshold of 80% (Supplementary Fig. 11), which was likely due to a drop in the movement-aligned HG response across a subset of channels (Supplementary Fig. 12). We found no hardware or software causes for the observed deviations in HG responses. Moreover, the participant had no subjective change in strength, no changes on detailed neurological examination or cognitive testing, and no new findings on brain computerized tomography images.

To ensure that BCI performance was not permanently affected, we retrained and tested a click detector with the same model architecture using data collected roughly four months after the observed performance drop (Supplementary Note 5). The new click detection algorithm used a total of 15 min of training data, which was all collected within one day (Supplementary Fig. 13a); afterward, the model weights remained fixed again. To determine the optimal voting threshold for continued long-term use, we additionally evaluated real-time click performance using all voting thresholds from 2/7 to 7/7 votes with this new click detection algorithm (Supplementary Fig. 14).

The participant used this retrained click detector for a total of 428 min in six sessions spanning a 21-day period after re-training (Supplementary Fig. 13b). The optimal combination of sensitivity (Supplementary Note 6) and false detections was achieved using a 6-vote threshold. Using this threshold, we achieved similar performance metrics to those from the original click detector with a 4-vote threshold, namely a median detection sensitivity of 94.8%, median TPF and FPF of 11.3 per min and 0.20 per min respectively, and a median CCPM and CWPM of 10.1 and 2.2 respectively (for all comparisons $P > 0.05$, Wilcoxon Rank-Sum test) (Supplementary Fig. 15). Expectedly the median on-screen click latency was 0.86 s, roughly 200 ms higher compared to the previous 4-vote threshold, due to the two extra votes required for generating a click ($P = 10^{-3}$, Wilcoxon Rank-Sum test).

Electrode contributions to grasp classification

To assess which channels produced the most important HG features for classification of attempted grasp, we generated a saliency map across all channels used to train our original model (Fig. 4a). As expected, channels covering cortical face region were generally not salient for grasp classification. The channel producing the most salient HG features was located in the upper-limb area of somatosensory cortex (channel 112, Supplementary Fig. 16), with a saliency value 55% and 88% higher than the next two most salient channels respectively (Supplementary Fig. 16a). Indeed, prior to the observed performance drop, this channel had a relatively amplified spectral response compared to other channels during attempted grasp. We then computed the corresponding offline classification accuracy of our original model architecture for comparison to a model architecture without channel 112 and an architecture using channels only over cortical hand-knob (see *Methods: Channel contributions and offline classification comparisons*); the mean accuracy from repeated 10-fold cross-validation (CV) was 92.9% (Fig. 4b).

To ensure that real-time classification accuracy was not entirely driven by channel 112, we evaluated a model trained on HG features from all other channels offline. As expected, this model relied strongly on channels covering the cortical hand-knob region (Fig. 4c), and notably was not as dependent on a single channel; the saliency of the most important channel was only 23% and 60% larger than the next two most salient channels, respectively (Supplementary Fig. 16b). The offline mean classification accuracy from repeated 10-fold CV was 91.7% (Fig. 4d), which was not significantly lower compared to the mean accuracy using all channels ($P = 0.139$, Wilcoxon Rank-Sum test with 3-way Bonferroni-Holm correction, Fig. 4g).

As channels covering the cortical hand-knob region made relatively larger contributions to decoding results, we investigated the classification accuracy of a model trained on HG features from a subset of electrodes covering only this region (Fig. 4e). Saliency values followed a flatter distribution; the saliency of the most important channel was only 21% and 44% larger than the next two most salient channels

respectively (Supplementary Fig. 16c). Though the offline mean classification accuracy from repeated 10-fold CV remained high at 90.4% (Fig. 4f), it was statistically lower compared to the mean accuracy using all channels ($P = 0.015$, Wilcoxon Rank-Sum test with 3-way Bonferroni-Holm correction, Fig. 4g). This suggests that a model trained on HG features from only the cortical hand-knob could still produce effective click detection, but parameters used for data labeling, model training, and post-processing may need to be more thoroughly explored to optimize click performance.

Figure 4 | Channel importance for grasp classification. Saliency maps for: the model used in online decoding, a model using HG features from all channels except from channel 112, and a model using HG features only from channels covering cortical hand-knob are shown in (a), (c) and (e) respectively. Electrodes overlaid with larger circles represent greater importance for grasp classification. White and transparent circles represent electrodes which were not used for model training. Mean confusion matrices from repeated 10-fold CV using models trained on HG features from all channels, all channels except for channel 112, and channels covering only the cortical hand-knob are shown in (b), (d), and (f) respectively. (g) Box and whisker plot showing the offline classification accuracies from 10 cross-validated testing folds using models with the above-mentioned channel subsets. Specifically, for one model configuration, each dot represents the average accuracy of the same validation fold across 20 repetitions of 10-fold CV (see *Methods: Channel contributions*). Offline classification accuracies from CV-models trained on all features from all channels were statistically higher than CV-models trained on features from channels only over cortical hand-knob ($*P = 0.015$, Wilcoxon Rank-Sum test with 3-way Bonferroni-Holm correction).

Discussion

In this study we show that a clinical trial participant with ALS was able to use a fixed decoder trained on a limited multichannel electrocorticographic (ECoG) dataset to generate stable real-time clicks over a period of three months. Specifically, the participant used his click detector to select the appropriate letters and words to form sentences using a switch-scanning spelling application. Our detector's high sensitivity (97.8%), low false positive frequency (0.101 per min) and minimal latency between onset of attempted grasp and click (0.48 s) allowed him to quickly and reliably spell sentences over a several months without retraining the model.

A significant barrier to the use of BCI systems by clinical populations outside of the laboratory is that users must often undergo an extensive period of training for optimizing fixed decoders¹, or daily model retraining or updating³. For example, reliable switch-scan spelling was demonstrated for up to 36

months using a fixed decoder but required several months of data collection to optimize parameters for inhibiting unintentional brain clicks^{1,17}. However, our click detector's long-term performance with a relatively small training dataset suggests a potentially reduced need for model optimization using ECoG signals with higher spatial density (for example, 12 electrodes with 4 mm pitch covering the cortical hand-knob region in this study compared to 4 electrodes with 10 mm pitch in the aforementioned one). Similarly, an endovascular electrode stent-array was recently used to train an attempted movement detector⁴. Though this is an extremely promising BCI technology for click decoding, the anatomical constraints on the number and proximity of electrodes in the stent-array to motor cortex may make it difficult to scale up from simple brain-clicks to more complex BCI commands^{3,36,37}. The device we used in this study may have included more electrodes over upper-limb cortex than was necessary for click detection, but it allowed us to explore the upper bounds of click performance that might be expected for a device with these capabilities in a participant with ALS.

Our model detected intended clicks with high sensitivity and low false positive rates. The high sensitivity was likely attributable to the high contrast between HG power during movement vs. rest, or baseline conditions, which had previously enabled real-time grasp detection^{3,38,39} but may have not been as robustly detected by the Acliva PC+S device⁴⁰ in previous work¹. The voting window provided a simple yet effective heuristic strategy for inhibiting false detections; post-hoc analysis of real-time performance revealed that false detections particularly increased when less than three votes were required for producing a click (Supplementary Fig. 13). We initially chose a conservative voting threshold of 100% (7/7 votes), but later adjusted it to 57% (4/7 votes), as the participant reported that he preferred an increased sensitivity and reduced click latency despite a slight increase in false detections. This experience supports the utility of allowing users to fine-tune algorithmic parameters that can affect BCI performance and the user experience and that may vary significantly among users and among different applications that use a click-detector.

Our results improve upon the previous work by Vansteensel et al. (2016) in which a participant with ALS was implanted with four contacts over hand motor cortex and achieved a spelling rate of 1.8 CPM and a latency of 1 s per click. These results may have been limited by lower sensitivity for high frequency activity, a single bipolar channel, and a 5 Hz transmission rate of power values (related to energy consumption of wireless signal transmission, see Vansteensel et al., 2016). In fact, our spelling rates were comparable to those from other clinical populations who have used switch scanning keyboards without a BCI, including people living with ALS⁴¹ or other causes of motor impairments⁴². It is worth noting that although the integration of eye-tracking with click decoding may enable even faster user interface navigation and spelling rates^{4,43}, it may also cause eyestrain during long periods of use⁴⁴ and worsen as residual eye movements deteriorate in late-stage ALS⁴⁵⁻⁴⁷.

To explore whether more limited electrode coverage of sensorimotor cortex would be sufficient for comparable click performance, we conducted a channel-wise saliency analysis. Despite the substantially higher saliency of one channel in post-central gyrus adjacent to the cortical hand knob, many of our highly salient channels were located over the pre-central gyrus at the cortical hand knob^{11,48}, and a virtual grid confined to this area had only a slight reduction in grasp classification accuracy (90.4%, vs. 92.9% in the all-channel model). As suggested by our high offline accuracy, a click detector of

comparable performance might be effective using this smaller cortical coverage while leaving open the possibility of training models for multiclass or cursor-based control.

After nearly four months without re-training or updating our model, we observed a drop in BCI performance caused by a modest decrease in the modulation of upper-limb HG power in several electrodes over hand area of sensorimotor cortex. This decrease was especially pronounced in the most salient channel used to train the original detector, so it was not unexpected that BCI performance was affected. There were no accompanying new neurological symptoms or changes in cognitive testing, nor any evidence of adverse events or device malfunction. Variations in signal amplitude and spectral energy similar to those we observed in our participant have been reported in ECoG signals recorded for several years by the Neuropace (TM) RNS system⁴⁹. However, the RNS system typically stores samples of ECoG from only 4 bipolar channels (8 contacts) in each patient, and is indicated for patients with epilepsy, not patients with ALS, for which there is scarce data on long-term ECoG. We are aware of only one such study¹⁷, but this study did not report signal characteristics on the granular timescale necessary for comparison to our results. Regardless of the cause, our click detector's small amount of training data did not include the signal regime we observed during the performance drop. Nevertheless, we successfully tested another click detector, which was retrained with even less data using a similar workflow, and achieved equally robust performance in subsequent testing sessions, suggesting that long-term discernability of HG activity was not affected. In the future, it may be possible to achieve both high performance and longevity by updating our model periodically, for example once every few months, simulating a periodic in-lab or outpatient checkup.

Our study adds to the expanding literature on ECoG as an effective recording modality for long-term BCI use. Importantly, due to the participant's residual upper limb movement, we were able to assess click performance using his "ground-truth" movement attempts. However, more work is needed to determine how a rapidly trained fixed click detector can provide long-term efficacy for the population of individuals suffering from more severe movement impairments. Robust click-detection capability complements recent major advancements in real-time spelling^{2,50} and speech decoding^{51,52} and provides a more application-agnostic capability for navigating menus and applications. Optimal spelling performance, however, was likely not realized as the linguistic statistics of our Harvard sentence prompts were not sufficiently representative of the word sequences on which our language model was trained. Therefore, we expect that spelling rates could be substantially improved during free-form spelling and even more so with a language model tuned to the linguistic preferences of the participant. Further, there is likely a user-specific regularity of model updates that would optimize the balance between independent long-term BCI use and technician intervention, which is especially relevant during home-use. Finally, we expect that click detectors, in addition to their utility as a communication tool, may be critical for accessibility software beyond spelling interfaces or communication boards such as web-browsers, internet of things (IoT), and multimedia platforms, and thus merit further investigation.

Data availability

Source data for Figs. 2- 4 can be accessed in Supplementary Data 1 (accessible upon manuscript acceptance). Beginning immediately after publication, individual participant data (neural, and

behavioral) and study protocol will be available from the corresponding author upon reasonable request.

Code availability

The analytical code for regenerating Figs.2-4 is available at [Zenodo link (accessible upon manuscript acceptance)]. Code used for offline model development and post-hoc analysis is freely available at [Zenodo link (accessible upon manuscript acceptance)]. Model training and offline analysis were done using Python (version 3.9.13). The recurrent neural network was built using Keras with a TensorFlow backend (version 2.8.0). Real time decoding was done in Python (version 3.10.12) using the ezmsg²⁵ messaging architecture (version 3.0.0).

Author contributions

[revised manuscript text omitted]

38. Yanagisawa, T. *et al.* Electrocorticographic control of a prosthetic arm in paralyzed patients. *Annals of Neurology* **71**, 353–361 (2012).
39. Yanagisawa, T. *et al.* Real-time control of a prosthetic hand using human electrocorticography signals. *Journal of Neurosurgery* 1715–1722 (2011) doi:10.3171/2011.1.JNS101421.
40. Swann, N. C. *et al.* Chronic multisite brain recordings from a totally implantable bidirectional neural interface: experience in 5 patients with Parkinson’s disease. *Journal of Neurosurgery* **128**, 605–616 (2018).
41. Lancioni, G. E. *et al.* A man with amyotrophic lateral sclerosis uses a mouth pressure microswitch to operate a text messaging system with a word prediction function. *Developmental Neurorehabilitation* **16**, 315–320 (2013).
42. Koester, H. H. & Arthanat, S. Text entry rate of access interfaces used by people with physical disabilities: A systematic review. *Assistive Technology* **30**, 151–163 (2018).
43. Oxley, T. Long-Term Safety of a Fully Implanted Endovascular Brain-Computer Interface for Severe Paralysis. *Archives of Physical Medicine and Rehabilitation* **103**, e53 (2022).
44. Käthner, I., Kübler, A. & Halder, S. Comparison of eye tracking, electrooculography and an auditory brain-computer interface for binary communication: a case study with a participant in the locked-in state. *J NeuroEngineering Rehabil* **12**, 76 (2015).
45. Sharma, R. Oculomotor Dysfunction in Amyotrophic Lateral Sclerosis: A Comprehensive Review. *Arch Neurol* **68**, 857 (2011).
46. Kang, B.-H., Kim, J.-I., Lim, Y.-M. & Kim, K.-K. Abnormal Oculomotor Functions in Amyotrophic Lateral Sclerosis. *J Clin Neurol* **14**, 464 (2018).
47. Farr, E., Altonji, K. & Harvey, R. L. LOCKED-IN Syndrome: Practical Rehabilitation Management. *PM&R* **13**, 1418–1428 (2021).

48. Pistoht, T., Schulze-Bonhage, A., Aertsen, A., Mehring, C. & Ball, T. Decoding natural grasp types from human ECoG. *NeuroImage* **59**, 248–260 (2012).
49. Sun, F. T., Arcot Desai, S., Tchong, T. K. & Morrell, M. J. Changes in the electrocorticogram after implantation of intracranial electrodes in humans: The implant effect. *Clinical Neurophysiology* **129**, 676–686 (2018).
50. Willett, F. R., Avansino, D. T., Hochberg, L. R., Henderson, J. M. & Shenoy, K. V. High-performance brain-to-text communication via handwriting. *Nature* **593**, 249–254 (2021).
51. Moses, D. A. *et al.* Neuroprosthesis for Decoding Speech in a Paralyzed Person with Anarthria. *N Engl J Med* **385**, 217–227 (2021).
52. Willett, F. R. *et al.* *A high-performance speech neuroprosthesis.*
<http://biorxiv.org/lookup/doi/10.1101/2023.01.21.524489> (2023) doi:10.1101/2023.01.21.524489.

A click-based electrocorticographic brain-computer interface enables long-term high-performance switch-scan spelling

Supplementary Materials

Table of Contents

Supplementary Notes

Supplementary Note 1 Exploratory nature of the clinical trial.	Page 3
Supplementary Note 2 Safety of the study device.	Page 3
Supplementary Note 3 Recording viability of the study device.	Page 3
Supplementary Note 4 BCI functionality of the study device.	Page 4
Supplementary Note 5 Retraining a new brain-click detection after performance drop.	Page 4
Supplementary Note 6 Selecting the optimal voting threshold.	Page 5

Supplementary Figures

Supplementary Fig. 1 Training paradigm.	Page 6
Supplementary Fig. 2 Cue-aligned spectrograms of upper-limb electrodes.	Page 7
Supplementary Fig. 3 Alignment of power trial rasters.	Page 8
Supplementary Fig. 4 Correlation increase of HG power in re-aligned signals.	Page 9
Supplementary Fig. 5 Correlation analysis of channel 112 using re-aligned trials.	Page 10
Supplementary Fig. 6 Labelling trial-averaged re-aligned HG power.	Page 11
Supplementary Fig. 7 Classification model architecture.	Page 12
Supplementary Fig. 8 User interface of communication board.	Page 13
Supplementary Fig. 9 User interface of switch-scanning speller.	Page 14
Supplementary Fig. 10 Timelines of brain-click use after training.	Page 15
Supplementary Fig. 11 Performance decline roughly 4 months post-training.	Page 16
Supplementary Fig. 12 Deviation in movement-aligned high-gamma power.	Page 17
Supplementary Fig. 13 Long-term use of a fixed retrained brain-click detector.	Page 18
Supplementary Fig. 14 Performance evaluation across all voting thresholds.	Page 19
Supplementary Fig. 15 Switch-scanning spelling performance with a retrained click detector.	Page 20
Supplementary Fig. 16 Normalized saliency values.	Page 21

Supplementary Tables

Supplementary Table 1 Experimental parameters for training data collection.	Page 22
---	---------

Supplementary Videos Descriptions

Supplementary Video 1 Switch-scan control of communication board.	Page 22
Supplementary Video 2 Switch-scan spelling.	Page 22

References

Page 23

Supplementary Notes

Supplementary Note 1. Exploratory nature of the clinical trial.

The clinical trial from which the results in this study are being reported is a single-center Phase 1 early feasibility trial to investigate the safety and preliminary efficacy of an investigational brain-computer interface (BCI) device. No more than 5 participants are planned to be enrolled and implanted. Due to the exploratory nature of the study, the primary outcomes are stated in general terms evaluating the safety and recording viability of the implanted device, as well as preliminary assessment of BCI functionality enabled by the device. The metrics and statistics for BCI functionality are not predefined due to the limited number of trial participants and the exploratory nature of the study with respect to strategies for achieving BCI functionality. Nevertheless, the results reported in each participant are to be evaluated to the highest statistical rigor in keeping with comparable studies, which also limited to individual participants¹⁻⁶.

Supplementary Note 2. Safety of the study device.

To date, there have been no adverse events related to the study device or study participation, and the participant has consented to continue the study. At this time the device has been implanted for more than a year and continues to be used for research purposes.

Supplementary Note 3. Recording viability of the study device.

The implant recording viability was assessed by visual inspection of the channel raw voltage signals as well as by measuring electrode impedances. Visual inspection of the voltage signals occurred at the beginning of each session, while electrode impedances were measured once per week.

Voltage signals were visually inspected for the presence of noise, signal amplitudes exceeding the digital bit-range of the recording hardware, signals with unusually low amplitude, or presence of movement artifact. Throughout the study, there were no fewer than 124 viable channels. Suboptimal signal quality was observed in four channels (chan19, chan38, chan48, and chan52, Fig. 1b) all of which were located on the electrode grid covering cortical speech areas. However only chan38 was consistently marked for suboptimal signal quality throughout the duration of the trial.

It is important to note that the decision to exclude chan38 from model training was made only after the initial click detection algorithm was deployed for real-time use. Meanwhile chan19, chan48, and chan52 were never marked for poor signal quality during data collection for initial model training but were marked as such later in the trial. The click detection algorithm trained without features from these channels (Supplementary Note 5) performed comparably to the model trained with these channel features included (see *Results: Decoder retraining due to transient performance drop*). Indeed, features from these four channels provided minimal contributions to classification (Supplementary Fig. 16) and by themselves produced a mean classification accuracy of 55.2% (chance 50%) using repeated 10-fold cross validation (see *Methods: Channel contributions and offline classification comparisons*).

Electrode impedances were measured using the Impedance Tester tool on Central Suite (Blackrock Neurotech Corp.). Particularly, electrodes whose impedance values exceeded a 15 k Ω threshold by an

order of magnitude were compared to channels with suboptimal signal quality and would be additionally excluded from analysis if they were not already marked by visual inspection. To date, chan38 has presented with an impedance consistently roughly two orders of magnitude higher than the 15 k Ω threshold. On one session chan121 also presented with a similarly high impedance.

Supplementary Note 4. BCI functionality of the study device.

The goal of this aspect of the study is to demonstrate control of external devices through speech and/or motor strategies, as well as a performance assessment of each strategy. Because of the exploratory nature of the study and the limited number of participants, specific strategies, tasks, performance metrics, and longitudinal assessments were not predefined. Rather, we anticipated that these methods would evolve during the trial and would be customized to each participant. Likewise, we did not formalize a statistical analysis plan in the protocol. In keeping with this protocol, the present work demonstrates a proof-of-concept spelling system controlled by attempted grasping movements and an ECoG-based BCI in a single participant. Our approach to reporting these results is similar to that of numerous published single-subject studies of implantable BCI functionality¹⁻⁶.

Supplementary Note 5. Retraining a new brain-click detection algorithm after performance drop.

We collected three blocks of training data using the “Go” task as described in *Methods: Training task*. Each block consisted of 60 trials during which the participant attempted to make a grasping motion, and each trial was followed by an interstimulus interval (ISI) during which the participant remained still and fixated his gaze on a crosshair in the center of the monitor. The ISI length was randomly chosen between 3.5-4.5 s. Each block started with a 30 s rest period during which the participant was instructed to remain still with his gaze fixated on a “Rest” stimulus. In total, 180 trials were collected for a total of almost 15 min of data. All three blocks were collected on the same day. Neural signals were recorded by the Neuroport system with a sampling rate of 2 kHz. We excluded four channels (chan19, chan38, chan48, chan52) due to poor raw voltage signal quality determined by visual inspection. High gamma (HG) features were computed and re-aligned for inter-trial variability as described in *Methods: Feature extraction and label assignment*. We assigned grasp labels to ECoG feature vectors falling between 0.4 s and 1.2 s post-cue for each trial and rest labels to all other feature vectors. We used the model architecture and techniques described in *Methods: Model design and training* to train our model. During spelling sessions, video of the participant’s right hand was collected at 60 frames per second (FPS) during all sessions with the retrained algorithm.

Supplementary Note 6. Selecting the optimal voting threshold.

We defined the optimal voting threshold as the one that produced the highest F_1 -score:

$$F_1 = \frac{2N_{TP}}{2N_{TP} + N_{FP} + N_{FN}}$$

where N_{TP} and N_{FP} are the number of true and false positive clicks respectively, and N_{FN} is the number of missed clicks. N_{FN} can be rewritten as $N_{attempted\ grasps} - N_{TP}$, and so our F_1 -score can be rewritten as:

$$F_1 = \frac{2N_{TP}}{N_{TP} + N_{FP} + N_{attempted\ grasps}}$$

The F_1 -scores for all voting thresholds are calculated using the total numbers of attempted grasps, true and false positives across all six sessions with the retrained brain-click detector:

Voting threshold	2/7	3/7	4/7	5/7	6/7	7/7
$N_{attempted\ grasps}$	597	582	476	421	423	461
N_{TP}	563	519	435	382	399	382
N_{FP}	57	23	20	13	14	0
F_1 -score	0.925	0.923	0.934	0.936	0.955	0.906

The 6-vote threshold produced the highest F_1 -score of 0.955.

Supplementary Figures

Supplementary Fig. 1 | Training paradigm. As the participant was seated in a chair with his forearms on the armrests, he was instructed to attempt a brisk grasp with his right hand immediately after the visual stimulus “Go” appeared on the monitor. One trial consisted of one 100 ms “Go” stimulus followed by an interstimulus interval (ISI) during which a centered white crosshair remained in the center of the monitor. The length of each ISI was randomly chosen between a lower and upper bound (3.5 - 4.5 s or 3 - 6 s, see Supplementary Table 1) to reduce anticipatory behavior.

Supplementary Fig. 2 | Cue-aligned spectrograms of upper-limb electrodes. Across trials from all training blocks for the original click detector, the cue-aligned trial-averaged spectrogram for all electrodes in the upper-limb grid is shown (-1 to 2.5 s post-cue). The vertical black lines for each electrode represent cue onset. Spectral power at each frequency is standardized to the statistics of the 1 min calibration period. The slightly increased broadband activity observed in some channels' pre-cue interval is likely due to the relatively heightened baseline activity during the training paradigm. The HG increase is followed by a low frequency rebound, which across many channels (especially in the center of the grid) reaches 100 Hz. The approximate central sulcus location is delineated by a thick black line (CS) and widens at the top such that electrodes B111, B119, and B127 are over it. The pre-central sulcus is delineated by a thick green line (Pre-CS).

Supplementary Fig. 3 | Alignment of power trial rasters. Across all training blocks for the original click detector, the cue-aligned HG power (110-170 Hz) for each trial is shown for all electrodes in the upper-limb grid (Cue-aligned). To account for inter-trial variability (especially for reaction delay in attempted movements), HG power between -1 s and 2.5 s post-cue from channels 86, 88, 94, 101, 102, 108, 109, 110, 112 were used to compute a per-trial shift, which was then used to re-align the HG power across all electrodes (Re-aligned). The approximate central sulcus location is delineated by a thick black line (CS) and widens at the top such that electrodes 111, 119, and 127 are over it. The pre-central sulcus is delineated by a thick green line (Pre-CS).

Supplementary Fig. 4 | Correlation increase of HG power in re-aligned signals. The change in correlation between the re-aligned and cue-aligned HG power trials for each electrode. Electrodes in the upper-limb grid (a) generally increased their inter-trial HG power correlation, whereas those in the speech grid (b) generally stayed the same. The central sulcus (CS) is delineated by a thick black line and widens at the top such that electrodes 111, 119, and 127 are over it. The pre-central sulcus (Pre-CS) is delineated by a thick green line.

Supplementary Fig. 5 | Correlation analysis of channel 112 using re-aligned trials. Channel 112 is presented as an example of increased inter-trial correlation of HG power after re-alignment. **(a)** Cue-aligned HG power across all trials. **(b)** Pair-wise correlation between all trials for channel 112. The mean correlation is computed by taking the mean of all inter-trial correlation values and is computed similarly in **(d)** and **(e)**. **(c)** Re-aligned HG power across all trials. **(d)** Increased pair-wise correlation between all trials. **(e)** Difference between Re-aligned and Cue-aligned HG power across all trials. **(f)** Mean pair-wise correlation change for each trial. For one trial, the mean pair-wise correlation change is computed by taking the mean of correlations between the HG power for that trial with those from all other trials.

Supplementary Fig. 6 | Labelling trial-averaged re-aligned HG power. Trial-averaged re-aligned HG power is shown for each of the channels that were used to compute the per-trial shift. Each 100 ms feature vector between 0.3 s and 1.1 s post-cue was labelled grasp while all other time points were labelled rest. The outlined grasp label at 0.9 s (dark green outline) displays an average representation of 1 s time history from the subset of channels used to train the original click detector.

Supplementary Fig. 7 | Classification model architecture. We used a recurrent neural network that utilizes long short-term memory (LSTM) cells in a many-to-one configuration to predict output probabilities. The architecture of the network is comprised of 3 layers: 1 LSTM layer with 25 units followed by 2 fully-connected layers with 10 and 2 units, respectively, that incorporate eLU and softmax activation functions for non-linear transformations. We trained this network on high-gamma power sequences with a fixed length of 1 s and a framesize of 100 ms using backpropagation through time (BPTT). Here, we relied on the Adam optimizer with an initial learning rate of $10e-4$. In total, the network was trained for 75 epochs and a batch size of 45.

Supplementary Fig. 8 | User interface of communication board. The participant was instructed to select an experimenter-cued graphical button from a 4x8 grid by timing his brain-clicks to the appropriate highlighted row or column during the switch-scanning cycle. Switch-scanning started by sequentially highlighting each row in green (not shown) for 1 s until the participant brain-clicked on the row containing the cued button. Once a row was selected, all eight buttons in that row became outlined in gold and then each column was sequentially highlighted for 1 s in red. Once the participant selected a button by brain-clicking, the button became green for 1 s before the switch-scanning process was reset at row 1 and the participant received another cue.

Supplementary Fig. 9 | User interface of spelling application. The participant was instructed to spell the sentence prompt (gray text) by timing his brain-clicks to the appropriate highlighted row or column during the switch-scanning cycle. The switch scanning process started by sequentially highlighting the three red pre-selection markers on either side of the sentence, where each highlight lasted 1 sec. This was to allow our participant a brief preparation period in case he wanted to select row 1. Rows 1-8 were then scanned sequentially until the participant brain-clicked on the row containing the appropriate letter or word. Rows 1 and 2 displayed our language model's most likely words and letters based on what the participant had already spelled. Row 3 allowed the participant to add a space, delete a letter or space, or delete an autocompleted word or letter (SPACE, DEL, and A-DEL keys respectively). Rows 4-8 contained all alphabetical letters (and some grammatical symbols) in case the desired letter was not suggested in Row 2. Once a row was selected, the gray pre-selection column on the left was highlighted in yellow for 1 sec to allow the participant a brief preparation period in case he wanted to select column 1. Buttons in the selected row were then sequentially highlighted in yellow for 1 s. At the end of each sequence of column highlights, the BACK key was highlighted to allow the participant to exit the row if it was selected accidentally. Brain-clicking on any button (highlighting it green for 1 s) within any row would reset the switch-scanning process at Row 1. The participant finalized the sentence by selecting the ENTER button in Row 8. The SHIFT (for letter capitalization) and CL buttons (for erasing the spelled words) were functional but not used during the testing sessions. If the participant had not selected any of the eight rows, the switch-scanning process was reset to Row 1.

Supplementary Fig. 10 | Timelines of brain-click use after training. Timeline of training and testing sessions relative to the surgical implantation (grey) where the implantation date is denoted as Day 0. Timeline of brain-click usage with the communication board and spelling interface (blue) relative to the last session of data collection for training the original click detector. Timeline of brain-click usage with the spelling interface (teal) relative to the only session of data collection for retraining a new classification model. The last day of data collection relative to the original and retrained models is denoted by Day 0. Note asterisks (Sessions 18-23 in which the original click detector was used) represent sessions in which brain-click detection performance was suboptimal.

Supplementary Fig. 11 | Performance decline roughly 4 months post-training. For each day a 4-vote threshold was used. (a) Sensitivity of grasp detection for each session. (b) True-positive and false-positive frequencies (TPF and FPF) measured as detections per minute. (c) Average latencies with standard deviation error bars of grasp onset to detection and to click. (d) Correct characters/words per minute (CCPM/CWPM).

Supplementary Fig. 12 | Deviation in movement-aligned high-gamma power. (a) Peak median HG power across all sessions. For each channel, a dot represents the peak HG power of the median HG power trace of the first 30 movement-aligned HG power traces from one spelling session. The dashed vertical line splits the peak HG power values prior to training a new click detector (left of dashed line) and after training a new click detector (right of dashed line). The largest drop in peak median HG power was from channel 112 (outlined in black), the most salient channel for decoding. The green, pink, and blue dots are peak median HG power values from the last day prior to performance drop, first day of decreased performance, and last day with newly trained click detector respectively. The central sulcus is delineated by a thick black line (CS) and widens at the top such that electrodes 111, 119, and 127 are over it. The pre-central sulcus is delineated by a thick green line (Pre-CS). (b) Average spectrogram of the first 30 movement-aligned trials from the last spelling session prior to performance drop. HG spectral power between 110-170 Hz is highlighted. (c) Average spectrogram of the first 30 movement-aligned trials from the first spelling session during performance drop. (d) Average spectrogram of the first 30 movement-aligned trials from the last spelling session after training a new click detector.

Supplementary Fig. 13 | Long-term use of a fixed retrained brain-click detector. (a) Training data was collected on one session where the three sub-bars each represent a separate block of training data collection. (b) The retrained fixed decoder was used by the participant with the complete range of voting thresholds on almost all days after retraining. For each day, each sub-bar represents a separate spelling block of 3 sentences. The horizontal axis spanning both (a) and (b) represents the number of days after the last day of training data collection (day 0).

Supplementary Fig. 14 | Performance evaluation across all voting thresholds. Voting thresholds ranging from 3/7 to 7/7 votes were evaluated on 6 sessions whereas the 2-vote threshold was evaluated only on 4 sessions. Sensitivity of grasp detection (a) true positive frequency (b) false positive frequency (c), and CCPM (d) observed with each voting threshold. Note that all above evaluations were performed using the retrained click detector.

Supplementary Fig. 15 | Switch-scanning spelling performance with a retrained click detector. All performance metrics are shown using the 6-vote threshold. **(a)** Sensitivity of grasp detection for each session. **(b)** True-positive and false-positive rates (TPF and FPF respectively) measured as detections per minute. **(c)** Average latencies with standard deviation error bars of grasp onset to algorithm detection and to on-screen click. **(d)** Correct characters/words per minute (CCPM/CWPM).

Supplementary Fig. 16 | Normalized saliency values. Normalized saliency values of each channel in descending order from a model trained on (a) all channels, (b) all channels excluding chan12, and (c) a 4x3 subset covering cortical hand-knob.

Supplementary Tables

	Blocks	Trials/Set	Number of sets	Post-set "rest"	ISI (seconds)	Total training time (min)
Session 1	1	50	1	No	3.5 - 4.5	3.77
Session 2	2	50	1	No	3.5 - 4.5	7.57
Session 3	1	50	3	Yes	3 - 6	14.09
Session 4	2	30	3	Yes	3 - 6	18.49

Supplementary Table 1 | Experimental parameters for training data collection. For each block during sessions 1 and 2 the participant performed one set of 50 trials whose ISIs were jittered between 3.5-4.5 s. For each block during sessions 3 and 4, we introduced a 90 s rest period after each of three sets of trials during which the participant was asked to remain still and fixate his eyes on a Rest stimulus. On session 4, we reduced the number of trials/set from 50 to 30 due to the participant's difficulty in focusing throughout the duration of the task. The ISI for sessions 3 and 4 was jittered between 3 - 6 s to further reduce anticipatory behavior. The above sessions were used to train the original click detector.

Supplementary Video Descriptions

Supplementary Video 1 | Switch-scan control of a communication board.

Using a switch-scanning paradigm, the participant navigates to one of 32 symbols on the communication board. The participant must time his attempted grasps such that the click occurs when the desired row or column is highlighted in red. The scan rate across rows and columns is 1 switch/1.5 s. Once a row or column is selected, it turns yellow or green respectively.

Supplementary Video 2 | Switch-scan spelling.

Using a switch-scanning paradigm, the participant navigates to the appropriate letter on the static keyboard or suggested letter or word to complete the prompted sentence (gray). The participant must time his attempted grasps such that the click occurs when the desired row or column is highlighted. The scan rate across rows and columns is 1 switch/1 s. After the sentence is complete, the participant clicks ENTER at the bottom of the screen.

References

1. Metzger, S. L. *et al.* Generalizable spelling using a speech neuroprosthesis in an individual with severe limb and vocal paralysis. *Nat Commun* **13**, 6510 (2022).
2. Silversmith, D. B. *et al.* Plug-and-play control of a brain–computer interface through neural map stabilization. *Nat Biotechnol* **39**, 326–335 (2021).
3. Benabid, A. L. *et al.* An exoskeleton controlled by an epidural wireless brain–machine interface in a tetraplegic patient: a proof-of-concept demonstration. *The Lancet Neurology* **18**, 1112–1122 (2019).
4. Ajiboye, A. B. *et al.* Restoration of reaching and grasping movements through brain-controlled muscle stimulation in a person with tetraplegia: a proof-of-concept demonstration. *The Lancet* **389**, 1821–1830 (2017).
5. Vansteensel, M. J. *et al.* Fully Implanted Brain–Computer Interface in a Locked-In Patient with ALS. *N Engl J Med* **375**, 2060–2066 (2016).
6. Collinger, J. L. *et al.* High-performance neuroprosthetic control by an individual with tetraplegia. *The Lancet* **381**, 557–564 (2013).

**Title**

A click-based electrocorticographic brain-computer interface enables long-term high-performance
switch-scan spelling

**Authors**

Daniel N. Candrea¹, Samyak Shah², Shiyu Luo¹, Miguel Angrick², Qinwan Rabbani³, Christopher Coogan²,
Griffin W. Milsap⁴, Kevin C. Nathan², Brock A. Wester⁴, William S. Anderson⁵, Kathryn R. Rosenblatt⁶,
Alpa Uchil², Lora Clawson², Nicholas J. Maragakis², Mariska J. Vansteensel⁷, Francesco V. Tenore⁴, Nicolas
F. Ramsey⁷, Matthew S. Fifer⁴, Nathan E. Crone²

1 Department of Biomedical Engineering, Johns Hopkins University School of Medicine, Baltimore, MD

2 Department of Neurology, Johns Hopkins University School of Medicine, Baltimore, MD

3 Department of Electrical and Computer Engineering, Johns Hopkins University, Baltimore, MD

4 Research and Exploratory Development Department, Johns Hopkins University Applied Physics Laboratory, Laurel, MD

5 Department of Neurosurgery, Johns Hopkins University School of Medicine, Baltimore, MD

6 Department of Anesthesiology and Critical Care Medicine, Johns Hopkins University School of Medicine, Baltimore, MD

7 Department of Neurology and Neurosurgery, UMC Utrecht Brain Center, Utrecht, The Netherlands

**Abstract**

*Background*

Brain-computer interfaces (BCIs) can restore communication in movement- and/or speech-impaired
individuals by enabling neural control of computer typing applications. Single command “click” decoders
provide a basic yet highly functional capability.

*Methods*

We sought to test the performance and long-term stability of click-decoding using a chronically
implanted high density electrocorticographic (ECoG) BCI with coverage of the sensorimotor cortex in a
human clinical trial participant (ClinicalTrials.gov, NCT03567213) with amyotrophic lateral sclerosis
(ALS). We trained the participant’s click decoder using a small amount of training data (<44 minutes
across four days) collected up to 21 days prior to BCI use, and then tested it over a period of 90 days
without any retraining or updating.

*Results*

Using this click decoder to navigate a switch-scanning spelling interface, the study participant was able
to maintain a median spelling rate of  2 characters per min. Though a transient reduction in signal
power modulation interrupted testing with this fixed model, a new click decoder achieved comparable
performance despite being trained with even less data (<15 min, within one day)

*Conclusion*

These results demonstrate that a click decoder can be trained with a small ECoG dataset while retaining
robust performance for extended periods, providing functional text-based communication to BCI users.

**Plain Language Summary**

People living with amyotrophic lateral sclerosis (ALS) struggle to communicate with family and
caregivers due to progressive muscle weakness. This study investigated whether the brain signals of a
participant with ALS could be used to control a spelling application. Specifically, when the participant
attempted to make a fist, a computer algorithm detected increased neural activity from electrodes
implanted on the surface of his brain, and thereby generated a mouse-click. The participant used these
self-generated clicks to select letters or words from a spelling application to type sentences. Our
algorithm was trained using less than one hour's worth of recorded brain signals and then performed
reliably for a period of three months. This approach can potentially be used to restore communication
to other severely paralyzed individuals over an extended period of time and after only a short training
period.

Introduction

Brain-computer interfaces (BCIs) can allow individuals with a variety of motor impairments to control
assistive devices using their neural signals¹⁻¹¹. In particular, implantable BCIs may provide round-the-clock
availability. These capabilities are derived either from single neuron activity recorded by microelectrode
arrays (MEAs), or from neural population activity recorded by macroelectrodes (typically consisting of
electrocorticographic (ECoG) arrays on the cortical surface)¹². Although sophisticated capabilities and
high performance of MEA BCIs have been reported, their use outside of research environments has
been limited due to varying degrees of long-term signal attrition^{13,14} and day-to-day instability in
decoding models trained on single neuron activity, often requiring frequent recalibration¹⁵. On the other
hand, extensive safety and efficacy data from the use of chronic ECoG recordings for epilepsy
management¹⁶ suggests that ECoG implants have the potential to deliver greater long-term signal
stability. However, the utility of ECoG for chronically implanted BCIs has only been tested in a few
participants.

In the first clinical trial of a chronic ECoG BCI¹, a participant with quadriplegia and anarthria due to
amyotrophic lateral sclerosis (ALS) attempted hand movements to generate "brain clicks", in turn
controlling a switch-scanning spelling application. These brain clicks were detected as spectral changes
in ECoG signals recorded from a single pair of electrodes on the surface of hand area of contralateral
motor cortex. Though the participant used these brain clicks to communicate in her daily life for more
than 3 years¹⁷, several months of data collection were necessary for parameter optimization. In a
separate clinical trial^{4,18}, participants with severe upper limb paralysis due to ALS or primary lateral
sclerosis were implanted with an endovascular stent-electrode array and required 1-12 sessions of
training with their brain click BCI before long-term use⁴. However, due to the location of electrodes in
the superior sagittal sinus, the participants triggered brain clicks with attempted foot movements, which
may not be intuitive for computer control. Moreover, device limitations in both clinical trials may have
constrained brain click speed and overall performance of the BCIs. Vansteensel et al. reported 87-91%
click accuracy (comprised of correctly detected and withheld clicks) with a 1 s latency¹ while Mitchell et
al. reported ~82% accuracy with a 0.9 s latency or a 97% accuracy with a 2.5 s latency⁴.

In this study, we tested whether improved click performance could be achieved using high density ECoG
recordings from sensorimotor cortex. We implanted two 8 x 8 ECoG grids (4 mm pitch, PMT Corp.,

Chanhassen, MN) over left hand and face cortical regions in a clinical trial participant with ALS
(ClinicalTrials.gov, NCT03567213). The participant generated clicks using the implanted BCI to spell
sentences at a significantly improved spelling rate compared to prior brain click work using a switch-
scanning paradigm¹. Moreover, the participant achieved high click-detection accuracies with low false-
positive rates and low latencies from attempted movement onset to click. We found that a fixed ECoG-
based click decoder trained on a limited dataset maintained high performance over a period of several
93 months without requiring re-training or daily model adaptation. Finally, offline analysis suggested that
similar performance is achievable with a smaller number of ECoG electrodes over only the cortical hand-
knob region.

[revised manuscript text omitted]

After computing each channel's trial-aligned HG power (-1 s to 2.5 s post-cue), we accounted for the
inter-trial variability due to reaction delay by re-aligning each trial's HG power using a subset of highly
activated channels²². This resulted in generally increased HG power correlations between trials
(Supplementary Figs. 3-5). We visually determined the onset and offset of the re-aligned trial-averaged
HG power from the channels used for re-alignment (Supplementary Fig. 6). The average neural activity
onset and offset were manually estimated from the aligned neural data to be roughly 0.2 s and 1.2 s
post-cue, respectively, with neural activity more clearly differentiating from rest activity starting at 0.3 s
post-cue and ending at 1.1 s post-cue. We consequently assigned grasp labels to ECoG feature vectors
falling between 0.3 s and 1.1 s post-cue for each trial, and rest labels to all other feature vectors. Since
this overall strategy relies only on the visual inspection of neural signals, we believe it to be compatible
with reduced availability of ground truth signals, like movement, as might be the case in locked-in
participants.

**Model architecture and training**

We designed a recurrent neural network in a many-to-one configuration to learn changes in HG power
over sequences of 1 s (Supplementary Fig. 7). Each 128-channel HG power vector was input into a long
short-term memory (LSTM) layer with 25 hidden units for modelling sequential dependencies. From
here, 2 consecutive fully-connected (FC) layers with 10 and 2 hidden units, respectively, determined
probabilities of the rest or grasp class. The former utilized an eLU activation function while the latter
employed softmax to output normalized probability values. In total, the architecture consisted of 17,932
trainable parameters, and was trained on a balanced dataset of rest and attempted grasping sequences
by randomly downsampling from the overrepresented rest class.

[revised manuscript text omitted]

**Performance evaluation**

*Sensitivity and click rates*

Sensitivity was measured as the percentage of correctly detected clicks:

$$323 \quad \text{Sensitivity} = \frac{N_{\text{true clicks}}}{N_{\text{attempted grasps}}} \times 100\%$$

where in one session $N_{\text{true clicks}}$ were the total number of correct clicks and N_{grasps} were the total
number of attempted grasps, and where $N_{\text{true clicks}} \leq N_{\text{attempted grasps}}$. For a detected click to be
correct (i.e., a true positive), it had to have occurred on the user interface (as visual feedback to the
participant) within 1.5 s after the onset of an attempted grasp. Attempted grasps with no clicks
occurring within this time period were considered false negatives. Clicks that occurred outside this time
period were assumed to be unrelated to any attempted grasp and were thus considered false positives.

True positive and false positive frequencies (TPF and FPF respectively) were measured per unit time and
for each session were defined as the following:

$$334 \quad TPF = \frac{N_{TP}}{T} = \frac{N_{\text{true clicks}}}{T} \quad FPF = \frac{N_{FP}}{T}$$

[revised manuscript text omitted]

*Reproducibility of experiments*

Neural data collection and processing as well as decoder performance were reproducible across sessions
as the participant was able to repeatedly demonstrate click control using neural signals from attempted
hand movements to spell sentences. However, as this study reports on the first and only participant in
this trial so far, further work will be necessary to test the reproducibility of these results in other
participants.

**Results**

**Long-term usage with a fixed click detector**

The participant used the fixed click detector to effectively control a switch-scanning application for a
total of 626 min spanning a 90-day period that started on Day 21 after the completion of training data
collection (Fig. 2b). Specifically, we recorded one session with the medical communication board and 17
sessions with the spelling application. We defined Day 0 as the last session of training data collection.
We used a voting threshold of 10/10 votes with the communication board. Using the spelling
application, we initially used a voting threshold of 7/7 votes, but reduced this threshold to 4/7 votes on
Day +81 as the participant reported that he preferred an increased sensitivity despite the resulting
increase in false positive detections. We found that the decoder performance remained robust for 111
421 days.  

**Figure 2 | Long-term use of a fixed click detector.** (a) Training data was collected during 4 sessions that occurred within a
period of 15 days. For each day, each sub-bar represents a separate block of training data collection (6 training blocks total). (b)
Using the fixed decoder, one block of switch-scanning with the communication board was performed +21 days post-training
data collection (purple). From Day +46 to Day +81, the fixed decoder was used for switch-scan spelling with a 7-vote threshold
(blue). From Day +81 to Day +111, the fixed decoder was used for switch-scan spelling with a 4-vote threshold (teal). For each
429 day, each sub-bar represents a separate spelling block of 3-4 sentences. The horizontal axis spanning both (a) and (b)
represents the number of days relative to the last day of training data collection (Day 0).

**Switch-scanning performance**

With the switch-scanning medical communication board, the click-detection model achieved 93%
sensitivity (percentage of detected clicks per attempted grasps) with a median latency of 1.23 s from
movement onset to on-screen click (visual feedback on the user interface) using a 10-vote threshold. No
false positives were detected.

Using the switch-scanning spelling application (from Day +46 to Day +111), the click detector achieved a
median detection sensitivity of 94.9% using a 7-vote threshold, and a sensitivity of 97.8% when using a
4-vote threshold ($P = 0.057$, Wilcoxon Rank-Sum test; Fig. 3a). The median true positive frequency (TPF)
was 10.7 per min using a 7-vote threshold, which improved to 11.6 per min when using a 4-vote
threshold ($P = 0.005$, Wilcoxon Rank-Sum test; Fig. 3b); the median false positive frequency (FPF) was
0.029 per min (1.74 per h) using a 7-vote threshold and 0.101 per min (6.03 per h) when using a 4-vote
threshold ($P = 0.20$, Wilcoxon Rank-Sum test; Fig. 3b).

As expected, we observed a decrease in latency from movement onset to algorithmic detection and on-
screen click when switching from the 7-vote to the 4-vote threshold (Fig. 3c). Using the 7-vote threshold,
the median detection latency was 0.75 s and significantly dropped to 0.48 s using the 4-vote threshold
($P = 0.013$, Wilcoxon Rank-Sum test). Meanwhile the median on-screen click latency was 0.93 s using the
7-vote threshold and dropped to 0.68 s using the 4-vote threshold ($P = 3 \times 10^{-4}$, Wilcoxon Rank-Sum
test). The delay between algorithmic detection and on-screen click was consistently ~200 msec, due to
network and computational overhead.

Consequently, the participant was able to achieve high rates of spelling (Fig. 3d). Specifically, median
spelling rate was 9.1 correct characters per minute (CCPM) using the 7-vote threshold, which
significantly improved to 10.2 CCPM using the 4-vote threshold ($P = 0.031$, Wilcoxon Rank-Sum test).
Similarly, he achieved 1.85 correct words per minute (CWPM) using the 7-vote threshold, which
significantly improved to 2.14 CWPM using the 4-vote threshold ($P = 0.015$, Wilcoxon Rank-Sum test). In

one session, the participant achieved a spelling rate greater than 11 CCPM with the 4-vote threshold,
which to our knowledge is the highest spelling rate achieved using single-command BCI control with a
switch-scanning spelling paradigm.

**Figure 3 | Long-term switch-scanning spelling performance.** Across all subplots, triangular and circular markers represent
metrics using a 7-vote and 4-vote voting threshold respectively. (a) Sensitivity of grasp detection for each session. Dashed line
delineates 100% sensitivity. (b) True-positive and false-positive frequencies (TPF and FPF) measured as detections per minute.
Dashed line delineates 0 FPF. (c) Average latencies with standard deviation error bars of grasp onset to algorithm detection and
to on-screen click. The averages and standard deviations were computed from latency measurements across all spelling blocks
from one session using the same voting threshold. Using 7-vote and 4-vote voting thresholds, on-screen clicks happened an
average of 207 ms and 203 ms respectively after detection. Note that detection latencies were not registered in the first six
sessions. (d) Correct characters and words per minute (CCPM and CWPM).

Decoder retraining due to transient performance drop

On Day +118 (Supplementary Fig. 10 for timeline), the detector sensitivity fell below the pre-set

performance threshold of 80% (Supplementary Fig. 11), which was likely due to a drop in the

movement-aligned HG response across a subset of channels (Supplementary Fig. 12). We found no

hardware or software causes for the observed deviations in HG responses. Moreover, the participant

had no subjective change in strength, no changes on detailed neurological examination or cognitive

testing, and no new findings on brain computerized tomography images.

To ensure that BCI performance was not permanently affected, we retrained and tested a click detector
with the same model architecture using data collected roughly four months after the observed
performance drop (Supplementary Note 5). The new click detection algorithm used a total of 15 min of
training data, which was all collected within one day (Supplementary Fig. 13a); afterward, the model
weights remained fixed again. To determine the optimal voting threshold for continued long-term use,
we additionally evaluated real-time click performance using all voting thresholds from 2/7 to 7/7 votes
with this new click detection algorithm (Supplementary Fig. 14).

The participant used this retrained click detector for a total of 428 min in six sessions spanning a 21-day
period after re-training (Supplementary Fig. 13b). The optimal combination of sensitivity
(Supplementary Note 6) and false detections was achieved using a 6-vote threshold. Using this
threshold, we achieved similar performance metrics to those from the original click detector with a 4-
vote threshold, namely a median detection sensitivity of 94.8%, median TPF and FPF of 11.3 per min and
0.20 per min respectively, and a median CCPM and CWPM of 10.1 and 2.2 respectively (for all
comparisons $P > 0.05$, Wilcoxon Rank-Sum test) (Supplementary Fig. 15). Expectedly the median on-
screen click latency was 0.86 s, roughly 200 ms higher compared to the previous 4-vote threshold, due
to the two extra votes required for generating a click ($P = 10^{-3}$, Wilcoxon Rank-Sum test).

**Electrode contributions to grasp classification**

To assess which channels produced the most important HG features for classification of attempted
grasp, we generated a saliency map across all channels used to train our original model (Fig. 4a). As
expected, channels covering cortical face region were generally not salient for grasp classification. The
channel producing the most salient HG features was located in the upper-limb area of somatosensory
cortex (channel 112, Supplementary Fig. 16), with a saliency value 55% and 88% higher than the next
two most salient channels respectively (Supplementary Fig. 16a). Indeed, prior to the observed
performance drop, this channel had a relatively amplified spectral response compared to other channels
during attempted grasp. We then computed the corresponding offline classification accuracy of our
original model architecture for comparison to a model architecture without channel 112 and an
architecture using channels only over cortical hand-knob (see *Methods: Channel contributions and*
*offline classification comparisons*); the mean accuracy from repeated 10-fold cross-validation (CV) was
92.9% (Fig. 4b).

To ensure that real-time classification accuracy was not entirely driven by channel 112, we evaluated a
model trained on HG features from all other channels offline. As expected, this model relied strongly on
channels covering the cortical hand-knob region (Fig. 4c), and notably was not as dependent on a single
channel; the saliency of the most important channel was only 23% and 60% larger than the next two
most salient channels, respectively (Supplementary Fig. 16b). The offline mean classification accuracy
from repeated 10-fold CV was 91.7% (Fig. 4d), which was not significantly lower compared to the mean
accuracy using all channels ($P = 0.139$, Wilcoxon Rank-Sum test with 3-way Bonferroni-Holm correction,
Fig. 4g).

As channels covering the cortical hand-knob region made relatively larger contributions to decoding
results, we investigated the classification accuracy of a model trained on HG features from a subset of
electrodes covering only this region (Fig. 4e). Saliency values followed a flatter distribution; the saliency
of the most important channel was only 21% and 44% larger than the next two most salient channels

respectively (Supplementary Fig. 16c). Though the offline mean classification accuracy from repeated
 10-fold CV remained high at 90.4% (Fig. 4f), it was statistically lower compared to the mean accuracy
 using all channels ($P = 0.015$, Wilcoxon Rank-Sum test with 3-way Bonferroni-Holm correction, Fig. 4g).
 This suggests that a model trained on HG features from only the cortical hand-knob could still produce
 effective click detection, but parameters used for data labeling, model training, and post-processing may
 need to be more thoroughly explored to optimize click performance.

**Figure 4 | Channel importance for grasp classification.** Saliency maps for the model used in real-time, a model using HG
 features from all channels except from channel 112, and a model using HG features only from channels covering cortical hand-
 knob are shown in (a), (c) and (e) respectively. Electrodes overlaid with larger circles represent greater importance for grasp
 classification. White and transparent circles represent electrodes which were not used for model training. Mean confusion
 matrices from repeated 10-fold CV using models trained on HG features from all channels, all channels except for channel 112,
 and channels covering only the cortical hand-knob are shown in (b), (d), and (f) respectively. For all confusion matrices, the
 539 percent value in each element of the matrix represents how many times the validation features across all repetitions of all
 540 validation folds were predicted correctly or incorrectly. The mean classification accuracy was computed from averaging the
 541 values on the diagonal of the confusion matrix. (g) Box and whisker plot showing the offline classification accuracies from 10
 cross-validated testing folds using models with the above-mentioned channel subsets. Specifically, for one model configuration,
 each dot represents the average accuracy of the same validation fold across 20 repetitions of 10-fold CV (see *Methods: Channel*
 *contributions*). Offline classification accuracies from CV-models trained on all features from all channels were statistically higher
 than CV-models trained on features from channels only over cortical hand-knob ($*P = 0.015$, Wilcoxon Rank-Sum test with 3-
 way Bonferroni-Holm correction). Offline classification accuracies from CV-models trained on features from all channels except
 for channel 112 were not statistically different from those trained on features from all channels or features only from channels
 only over cortical hand-knob.

 **Discussion**

In this study we show that a clinical trial participant with ALS was able to use a fixed decoder trained on
 a limited multichannel electrocorticographic (ECoG) dataset to generate stable real-time clicks over a
 period of three months. Specifically, the participant used his click detector to select the appropriate
 letters and words to form sentences using a switch-scanning spelling application. Our detector's high
 sensitivity (97.8%), low false positive frequency (0.101 per min) and minimal latency between onset of
 attempted grasp and click (0.48 s) allowed him to quickly and reliably spell sentences over a several
 558 months without retraining the model.

A significant barrier to the use of BCI systems by clinical populations outside of the laboratory is that
users must often undergo an extensive period of training for optimizing fixed decoders¹, or daily model
retraining or updating³. For example, reliable switch-scan spelling was demonstrated for up to 36
563 months using a fixed decoder but required several months of data collection to optimize parameters for
inhibiting unintentional brain clicks^{1,17}. However, our click detector's long-term performance with a
relatively small training dataset suggests a potentially reduced need for model optimization using ECoG
signals with higher spatial density (for example, 12 electrodes with 4 mm pitch covering the cortical
hand-knob region in this study compared to 4 electrodes with 10 mm pitch in the aforementioned one).
Similarly, an endovascular electrode stent-array was recently used to train an attempted movement
detector⁴. Though this is an extremely promising BCI technology for click decoding, the anatomical
constraints on the number and proximity of electrodes in the stent-array to motor cortex may make it
difficult to scale up from simple brain-clicks to more complex BCI commands^{3,36,37}. The device we used in
this study may have included more electrodes over upper-limb cortex than was necessary for click
detection, but it allowed us to explore the upper bounds of click performance that might be expected
for a device with these capabilities in a participant with ALS.

Our model detected intended clicks with high sensitivity and low false positive rates. The high sensitivity
was likely attributable to the high contrast between HG power during movement vs. rest, or baseline
conditions, which had previously enabled real-time grasp detection^{3,38,39} but may have not been as
robustly detected by the Activa PC+S device⁴⁰ in previous work¹. The voting window provided a simple
yet effective heuristic strategy for inhibiting false detections; post-hoc analysis of real-time performance
revealed that false detections particularly increased when less than three votes were required for
producing a click (Supplementary Fig. 13). We initially chose a conservative voting threshold of 100%
(7/7 votes), but later adjusted it to 57% (4/7 votes), as the participant reported that he preferred an
increased sensitivity and reduced click latency despite a slight increase in false detections. This
experience supports the utility of allowing users to fine-tune algorithmic parameters that can affect BCI
performance and the user experience and that may vary significantly among users and among different
applications that use a click-detector.

Using a switch-scanning spelling application, the participant achieved high spelling rates by timing his
clicks to select the appropriate row or column. Our results improve upon the previous work by
Vansteensel et al. (2016) in which a participant with ALS was implanted with four contacts over hand
motor cortex and achieved a spelling rate of 1.8 CPM and a latency of 1 s per click. These results may
have been limited by lower sensitivity for high frequency activity, a single bipolar channel, and a 5 Hz
transmission rate of power values (related to energy consumption of wireless signal transmission, see
Vansteensel et al., 2016). In fact, our spelling rates were comparable to those from other clinical
populations who have used switch scanning keyboards without a BCI, including people living with ALS⁴¹
or other causes of motor impairments⁴². It is worth noting that although the integration of eye-tracking
with click decoding may enable even faster user interface navigation and spelling rates^{4,43}, it may also
cause eyestrain during long periods of use⁴⁴ and worsen as residual eye movements deteriorate in late-
stage ALS⁴⁵⁻⁴⁷.

To explore whether more limited electrode coverage of sensorimotor cortex would be sufficient for
comparable click performance, we conducted a channel-wise saliency analysis. Despite the substantially
higher saliency of one channel in post-central gyrus adjacent to the cortical hand knob, many of our
highly salient channels were located over the pre-central gyrus at the cortical hand knob^{11,48}, and a
virtual grid confined to this area had only a slight reduction in grasp classification accuracy (90.4%, vs.
92.9% in the all-channel model). As suggested by our high offline accuracy, a click detector of
comparable performance might be effective using this smaller cortical coverage while leaving open the
possibility of training models for multiclass or cursor-based control.

After nearly four months without re-training or updating our model, we observed a drop in BCI
performance caused by a modest decrease in the modulation of upper-limb HG power in several
electrodes over hand area of sensorimotor cortex. This decrease was especially pronounced in the most
salient channel used to train the original detector, so it was not unexpected that BCI performance was
affected. There were no accompanying new neurological symptoms or changes in cognitive testing, nor
any evidence of adverse events or device malfunction. Variations in signal amplitude and spectral energy
similar to those we observed in our participant have been reported in ECoG signals recorded for several
618 years by the Neuropace (TM) RNS system⁴⁹. However, the RNS system typically stores samples of ECoG
from only 4 bipolar channels (8 contacts) in each patient, and is indicated for patients with epilepsy, not
patients with ALS, for which there is scarce data on long-term ECoG. We are aware of only one such
study¹⁷, but this study did not report signal characteristics on the granular timescale necessary for
comparison to our results. Regardless of the cause, our click detector's small amount of training data did
not include the signal regime we observed during the performance drop. Nevertheless, we successfully
tested another click detector, which was retrained with even less data using a similar workflow, and
achieved equally robust performance in subsequent testing sessions, suggesting that long-term
discernability of HG activity was not affected. In the future, it may be possible to achieve both high
performance and longevity by updating our model periodically, for example once every few months,
simulating a periodic in-lab or outpatient checkup.

Our study adds to the expanding literature on ECoG as an effective recording modality for long-term BCI
use. Importantly, due to the participant's residual upper limb movement, we were able to assess click
performance using his "ground-truth" movement attempts. However, more work is needed to
determine how a rapidly trained fixed click detector can provide long-term efficacy for the population of
individuals suffering from more severe movement impairments. Robust click-detection capability
complements recent major advancements in real-time spelling^{2,50} and speech decoding^{51,52} and provides
a more application-agnostic capability for navigating menus and applications. Optimal spelling
performance, however, was likely not realized as the linguistic statistics of our Harvard sentence
prompts were not sufficiently representative of the word sequences on which our language model was
trained. Therefore, we expect that spelling rates could be substantially improved during free-form
spelling and even more so with a language model tuned to the linguistic preferences of the participant.
Further, there is likely a user-specific regularity of model updates that would optimize the balance
between independent long-term BCI use and technician intervention, which is especially relevant during
home-use. Finally, we expect that click detectors, in addition to their utility as a communication tool,
may be critical for accessibility software beyond spelling interfaces or communication boards such as
web-browsers, internet of things (IoT), and multimedia platforms, and thus merit further investigation.

Data availability

Source data for Figs. 2- 4 can be accessed in Supplementary Data 1 (accessible upon manuscript acceptance). Beginning immediately after publication, individual participant data (neural, and behavioral) and study protocol will be available from the corresponding author upon reasonable request.

Code availability

The analytical code for regenerating Figs.2-4 is available at [Zenodo link (accessible upon manuscript acceptance)]. Code used for offline model development and post-hoc analysis is freely available at [Zenodo link (accessible upon manuscript acceptance)]. Model training and offline analysis were done using Python (version 3.9.13). The recurrent neural network was built using Keras with a TensorFlow backend (version 2.8.0). Real time decoding was done in Python (version 3.10.12) using the ezmsg²⁵ messaging architecture (version 3.0.0).

Author contributions

[revised manuscript text omitted]

- 22. Williams, A. H. *et al.* Discovering Precise Temporal Patterns in Large-Scale Neural Recordings
through Robust and Interpretable Time Warping. *Neuron* **105**, 246-259.e8 (2020).
- 23. Kingma, D. P. & Ba, J. Adam: A Method for Stochastic Optimization. (2014)
doi:10.48550/ARXIV.1412.6980.
- 24. He, K., Zhang, X., Ren, S. & Sun, J. Delving Deep into Rectifiers: Surpassing Human-Level
Performance on ImageNet Classification. (2015) doi:10.48550/ARXIV.1502.01852.
- 25. Peranich, P., Milsap, G. & peranpl1. ezmsg. (2022).
- 26. Rezeika, A. *et al.* Brain–Computer Interface Spellers: A Review. *Brain Sciences* **8**, 57 (2018).
- 27. Koester, H. H. & Simpson, R. C. Effectiveness and usability of Scanning Wizard software: a tool
for enhancing switch scanning. *Disability and Rehabilitation: Assistive Technology* **14**, 161–171 (2019).
- 28. Koester, H. H., Simpson, R. C., & ATP. Method for enhancing text entry rate with single-switch
scanning. *J Rehabil Res Dev* **51**, 995–1012 (2014).
- 29. Mankowski, R., Simpson, R. C. & Koester, H. H. Validating a model of row–column scanning.
*Disability and Rehabilitation: Assistive Technology* **8**, 321–329 (2013).
- 30. Mackenzie, I. S. & Felzer, T. SAK: Scanning ambiguous keyboard for efficient one-key text entry.
*ACM Trans. Comput.-Hum. Interact.* **17**, 1–39 (2010).
- 31. Bhattacharya, S., Samanta, D. & Basu, A. Performance Models for Automatic Evaluation of
Virtual Scanning Keyboards. *IEEE Trans. Neural Syst. Rehabil. Eng.* **16**, 510–519 (2008).
- 32. Angelo, J. Comparison of Three Computer Scanning Modes as an Interface Method for Persons
With Cerebral Palsy. *The American Journal of Occupational Therapy* **46**, 217–222 (1992).
- 33. Sanh, V., Debut, L., Chaumond, J. & Wolf, T. DistilBERT, a distilled version of BERT: smaller,
faster, cheaper and lighter. Preprint at <http://arxiv.org/abs/1910.01108> (2020).
- 34. Rothausler, E. H. IEEE Recommended Practice for Speech Quality Measurements. *IEEE Trans.*
*Audio Electroacoust.* **17**, 225–246 (1969).

- 35. Sundararajan, M., Taly, A. & Yan, Q. Axiomatic Attribution for Deep Networks. *Proceedings of*
*the 34th International Conference on Machine Learning, PMLR 70*, 3319–3328.
- 36. Branco, M. P. *et al.* Decoding hand gestures from primary somatosensory cortex using high-
density ECoG. *Neuroimage 147*, 130–142 (2017).
- 37. Wang, W. *et al.* An Electrocorticographic Brain Interface in an Individual with Tetraplegia. *PLoS*
*ONE 8*, e55344 (2013).
- 38. Yanagisawa, T. *et al.* Electrocorticographic control of a prosthetic arm in paralyzed patients.
*Annals of Neurology 71*, 353–361 (2012).
- 39. Yanagisawa, T. *et al.* Real-time control of a prosthetic hand using human electrocorticography
signals. *Journal of Neurosurgery 1715–1722* (2011) doi:10.3171/2011.1.JNS101421.
- 40. Swann, N. C. *et al.* Chronic multisite brain recordings from a totally implantable bidirectional
neural interface: experience in 5 patients with Parkinson’s disease. *Journal of Neurosurgery 128*, 605–
616 (2018).
- 41. Lancioni, G. E. *et al.* A man with amyotrophic lateral sclerosis uses a mouth pressure
microswitch to operate a text messaging system with a word prediction function. *Developmental*
*Neurorehabilitation 16*, 315–320 (2013).
- 42. Koester, H. H. & Arthanat, S. Text entry rate of access interfaces used by people with physical
disabilities: A systematic review. *Assistive Technology 30*, 151–163 (2018).
- 43. Oxley, T. Long-Term Safety of a Fully Implanted Endovascular Brain-Computer Interface for
Severe Paralysis. *Archives of Physical Medicine and Rehabilitation 103*, e53 (2022).
- 44. Käthner, I., Kübler, A. & Halder, S. Comparison of eye tracking, electrooculography and an
auditory brain-computer interface for binary communication: a case study with a participant in the
locked-in state. *J NeuroEngineering Rehabil 12*, 76 (2015).

- 45. Sharma, R. Oculomotor Dysfunction in Amyotrophic Lateral Sclerosis: A Comprehensive Review.
*Arch Neurol* **68**, 857 (2011).
- 46. Kang, B.-H., Kim, J.-I., Lim, Y.-M. & Kim, K.-K. Abnormal Oculomotor Functions in Amyotrophic
Lateral Sclerosis. *J Clin Neurol* **14**, 464 (2018).
- 47. Farr, E., Altonji, K. & Harvey, R. L. LOCKED-IN Syndrome: Practical Rehabilitation Management.
*PM&R* **13**, 1418–1428 (2021).
- 48. Pistohl, T., Schulze-Bonhage, A., Aertsen, A., Mehring, C. & Ball, T. Decoding natural grasp types
from human ECoG. *NeuroImage* **59**, 248–260 (2012).
- 49. Sun, F. T., Arcot Desai, S., Tcheng, T. K. & Morrell, M. J. Changes in the electrocorticogram after
implantation of intracranial electrodes in humans: The implant effect. *Clinical Neurophysiology* **129**,
676–686 (2018).
- 50. Willett, F. R., Avansino, D. T., Hochberg, L. R., Henderson, J. M. & Shenoy, K. V. High-
performance brain-to-text communication via handwriting. *Nature* **593**, 249–254 (2021).
- 51. Moses, D. A. *et al.* Neuroprosthesis for Decoding Speech in a Paralyzed Person with Anarthria. *N*
*Engl J Med* **385**, 217–227 (2021).
- 52. Willett, F. R. *et al.* *A high-performance speech neuroprosthesis.*
<http://biorxiv.org/lookup/doi/10.1101/2023.01.21.524489> (2023) doi:10.1101/2023.01.21.524489.

Reviewer #1 (Remarks to the Author):

I thank the authors for their great contribution to the BCI literature on ECoG-based selection decoding in an individual with ALS. The novelty of this proposal is high, and demonstrates improvements from previous efforts which by using a more extensive cortical grid and achieving higher communication rates.

This paper describes the implant of two ECoG grids in an individual with ALS, who, over the course of about a year trains and eventually uses the device to operate a switch-scanning interface. The authors describe the longitudinal performance of the system that used two different scanning interfaces, as well as the longitudinal performance, and the adaptations made over the course of the study. I think it is a instructive piece of work and sets the stage for future exciting results out of this group.

I have a few major concerns which I would like addressed, in addition to multiple other comments I have included in a revised manuscript and supplementary materials.

1) Better description is needed of the participant in terms of established clinical measures, such as ALSFRS-R scores.

Thank you for this suggestion. We have appended the ALSFRS-R scores to the second paragraph of the Participant section and to the Supplementary Information (modifications in green). We have copied the summary of the results below:

“The clinical care team at Johns Hopkins Hospital obtained the assessment for the ALSFRS-R measure one day before we started collection of training data. In total, the study participant in the clinical trial scored 26 out of 48 points.”

Which cognitive and neurological tests, mentioned first in the results section on decoder retraining, were performed?

Thank you for this question. We have added a description of the cognitive testing that the participant underwent in paragraph 3 of Participant section (modifications in green).

“The participant was screened with cognitive testing prior to his enrollment in the study, and no evidence for dementia was found. During monthly safety assessments, the participant underwent a brief cognitive testing battery. This has not revealed any significant decline in cognitive function since study enrollment.”

2) The manuscript contains some blocks of text which this reviewer finds unnecessary for the understanding of the manuscript, or which better belong in other locations.

a) Most of the last paragraph of the introduction belongs in the results and methods.

Thank you for this suggestion. We have rewritten the Introduction, including the last paragraph, that we have copied below for the reviewer’s convenience (relevant text is green in the revised manuscript):

“The previous studies described above showed that click detectors can be used with a variety of BCI applications and can contribute significantly to a user’s repertoire of communication modalities. Despite these promising results, the potential performance limits of such click detectors have remained relatively underexplored. In particular, chronic high-performance use without model retraining is a critical factor for enabling independent home-use, as BCI users should have round-the-clock access to a functioning click detector that requires minimal caregiver involvement. By leveraging the stability of ECoG signals, we were able to train a model on a limited dataset and test it for a period of three months without retraining or daily model adaptation. Specifically, we demonstrated an improved click detector with a substantially increased spelling rate using a switch-scanning paradigm.”

b) Most of the first paragraph of the methods, along with Supplementary notes 1 and 4 can be excluded.

Thank you for this suggestion. However, by request of the editor, we must keep content that relates to the clinical trial. Since we reference Supplementary Notes 1 and 4 in paragraph 1 of the Methods, it follows that we must not delete those either.

c) Related to b), the participant paragraph of the methods tries to make statements related to the whole CortiCom clinical trial, which I do not think are necessary for this paper.

Thank you for this suggestion. However, by request of the editor, we indeed must include statements on how the results of the study contribute to the overall trial. However, we have moved this statement from the Participant section to the Clinical trial section of the Methods (relevant is green in revised manuscript):

“The secondary outcomes of the CortiCom trial are reported here only partially, as click detection is only one of a variety of BCI control strategies explored by the trial; specifically, our success rate and latency are reported in terms of click detection accuracy and time from attempted movement onset to click.”

d) others noted in attached manuscript.

3) Cue-aligned vs trial-aligned HG power. The description of this epoch shifting/time warping was not well understood by me. Also, the need for performing this was not well described. Did decoder performance suffer if this shifting of training epochs was not performed? Better description needs to be given of how shifting/warping occurred, what extent of shifting/warping was performed, and how this affected classification performance. Other than the technical demonstration that shifting trials to match the peak of each other increases the correlation between them, I don’t think Supplementary figures 4 & 5 are useful.

We thank the reviewer for these critiques and suggestions. We have rewritten the relevant paragraph, which is now in the Label assignment section, to clarify our method for assigning labels and the rationale and methods for shift-warping. We have copied that paragraph below for the reviewer’s convenience (relevant text is green in revised manuscript):

“We assigned rest and grasp labels to each sample in our training dataset by the following steps. First, for each channel we concatenated segments of HG power across all trials, where

each trial segment ranged from -1 s to 2.5 s relative to the beginning of the visual “Go” cue (Supplementary Fig. 3, Cue-aligned). To account for the inter-trial variability of the participant’s reaction delay to the visual “Go” cue, we temporally re-aligned the HG power across all trial segments using a shift warping model³⁷ (Supplementary Fig. 3, Re-aligned). This model was trained on only a subset of highly modulated channels (determined qualitatively; Supplementary Fig. 3 caption) to decrease the potential influence of artificial patterns from low-modulation channels when re-aligning trial segments. Note that for each trial, the resulting temporal re-alignment was applied similarly across all 128 channels. This re-alignment resulted in generally increased HG power correlations between trials (Supplementary Figs. 4, 5). We then computed the trial-averaged HG power traces using the re-aligned trial segments of only these highly modulated channels and visually determined the onset and offset of the average modulation relative to the beginning of the visual “Go” cue (Supplementary Fig. 6). This onset and offset time were estimated to be 0.3 s and 1.1 s, respectively, relative to onset of the “Go” cue. We consequently assigned grasp labels to ECoG feature vectors falling between and including $0.3\text{ s} + t_{\text{shift}}$ and $1.1\text{ s} + t_{\text{shift}}$ relative to the “Go” cue for each trial. Note that it was necessary to include the term t_{shift} to the bounds where grasp labels were assigned to account for the shift that was applied to each trial. Rest labels were applied to feature vectors at all other time points. We adopted this labeling strategy because it relied only on the visual inspection of neural signals, simulating the lack of ground truth for attempted movements that would be expected for BCI users with Locked-in Syndrome (LIS).”

We also apologize for the confusion between “trial-aligned” and “cue-aligned.” The term “trial-aligned” was left in the main text by mistake and has now been removed completely.

3.1) My understanding is that video was not used to capture movement onset and offset required to perform shifting. Why?

Thank you for this question. We did not use the attempted movement onset and offset from video recordings to perform movement-alignment of the power trial rasters. This was because we aimed to simulate an inevitable locked-in state of the participant where attempted movement would not be observable. We clarified this at the end of the Label assignment section:

“We adopted this labeling strategy because it relied only on the visual inspection of neural signals, simulating the lack of ground truth for attempted movements that would be expected for BCI users with Locked-in Syndrome (LIS).”

4) Some discussion is warranted for why no attempt was made at performing imagination of hand grasp to determine similarities and differences with actual hand grasp in terms of cortical representation and performance.

We appreciate the reviewer’s concern. This study reports one of the first results from our clinical trial, which is exploring multiple strategies for optimizing functionality of ECoG-based BCI over extended use periods, leveraging the stability of ECoG signals. Taken within the context of this clinical trial, we aimed to allow the participant to use the BCI as soon as possible via a control strategy that we hypothesized would be robust over an extended use period. As the participant’s condition progresses, we may consider investigating control strategies based on imagined movements; we have now added that possibility and others at the end of paragraph 7 of the Discussion (relevant text in green):

“There were several limitations to this study. Though the participant retained the ability to perform partial grasping movements, additional work is needed to determine how performance of a rapidly trained click detector would generalize to individuals with more severe movement impairments. Click detection using signals from less affected regions of the cortex and control strategies that are not based on attempted movement, but rather on imagined movements or responses to sensory input could serve as alternative control strategies.”

Reviewer # 1 Continued: Responses to comments from manuscript PDF

Commented [A1]: This belongs in methods and results.

Thank you for your suggestion regarding moving the information in the final paragraph of the Introduction to Methods and Results:

Previously the final paragraph began with: “We implanted two 8 x 8 ECoG grids (4 mm pitch, PMT Corp., Chanhassen, MN) over left hand and face cortical regions in a clinical trial participant with ALS ...”

We have re-written our Introduction and have copied the final paragraph below for the reviewer’s convenience (relevant text is green in the revised manuscript):

“The previous studies described above showed that click detectors can be used with a variety of BCI applications and can contribute significantly to a user’s repertoire of communication modalities. Despite these promising results, the potential performance limits of such click detectors have remained relatively underexplored. In particular, chronic high-performance use without model retraining is a critical factor for enabling independent home-use, as BCI users should have round-the-clock access to a functioning click detector that requires minimal caregiver involvement. By leveraging the stability of ECoG signals, we were able to train a model on a limited dataset and test it for a period of three months without retraining or daily model adaptation. Specifically, we demonstrated an improved click detector with a substantially increased spelling rate using a switch-scanning paradigm.

Commented [A2]: I don’t see why this is needed. ECoG is a familiar technology to readers and it is clear from the nature of the study

Thank you for this suggestion. However, by request of the editor, we must keep the content related to the clinical trial, which is in the Clinical trial section. Since we refer to Supplementary Notes 1 and 4 in this section, it follows that we must not delete those either.

Commented [A3]: Is it necessary to make statements related to the CortiCom trial? It seems unnecessary and premature.

Thank you for this suggestion. However, by request of the editor, we indeed must include statements on how the results of the study contribute to the overall trial. However, we have moved this statement from the Participant section to the Clinical trial section of the Methods (relevant text is green in revised manuscript):

“The secondary outcomes of the CortiCom trial are reported here only partially, as click detection is only one of a variety of BCI control strategies explored by the trial; specifically, our success rate and latency are reported in terms of click detection accuracy and time from attempted movement onset to click.”

Commented [A4]: What is not clear in this description is that the participant had robust performance of a hand grasp, which is clear only in the supplementary videos. Was imagined movement attempted as well?

Thank you for these questions. We have included a statement about the participant’s residual upper-limb movements in paragraph 2 of the Participant section (modifications in green):

*“The participant was a right-handed man who was 61 years old at the time of implant in July 2022 and diagnosed with ALS roughly 8 years prior. Due to bulbar dysfunction, the participant had severe dysphagia and progressive dysarthria. This was accompanied by progressive dyspnea. The participant could still produce overt speech, but slowly and with limited intelligibility. He had experienced progressive weakness in his upper limbs such that he ~~is~~*was* incapable of performing activities of daily living without assistance;. He *could* partially close his fingers in an attempted grasp gesture, but he had insufficient strength to hold a cup with one hand. His lower limbs ~~are less affected~~ had good strength and allowed him to ambulate, albeit with intermittent imbalance due to impaired arm swing...”*

We did not collect any data on imagined movements. Briefly, this study shows one of the first results from our clinical trial, which is largely aimed at optimizing functionality of ECoG-based BCI over extended use periods by leveraging the stability of ECoG signals. Taken within the context of this clinical trial, we aimed to allow the participant to use the BCI as soon as possible via a control strategy that we hypothesized would be robust over an extended use period. As the participant’s condition progresses, we may consider investigating control strategies based on imagined movements and we have now added this possibility and others at the end of paragraph 7 of the Discussion. We have copied this below for the reviewer’s convenience (relevant text in green):

“There were several limitations to this study. Though the participant retained the ability to perform partial grasping movements, additional work is needed to determine how performance of a rapidly trained click detector would generalize to individuals with more severe movement impairments. Click detection using signals from less affected regions of the cortex and control strategies that are not based on attempted movement, but rather on imagined movements or responses to sensory input could serve as alternative control strategies.”

Was a cognitive battery or screen run with this participant?

Thank you for this question. We have added a description of the cognitive testing that the participant underwent in paragraph 3 of Participant section and have copied it below for the reviewer's convenience (relevant text is green in revised manuscript).

"The participant was screened with cognitive testing prior to his enrollment in the study, and no evidence for dementia was found. During monthly safety assessments, the participant underwent a brief cognitive testing battery. This has not revealed any significant decline in cognitive function since study enrollment."

Commented [A5]: Do we have the ALSFRS-R scores in bulbar, fine motor, gross motor, and respiratory subdomains?

Thank you for this suggestion. We have included the ALSFRS-R scores in paragraph 2 of the Participant section and the Supplementary Information (relevant text in green). We have copied the summary of the results below:

"The clinical care team at Johns Hopkins Hospital obtained the assessment for the ALSFRS-R measure one day before we started collection of training data. In total, the participant in the clinical trial scored 26 out of 48 points."

Commented [A6]: Where was reference located? Was any re-referencing performed?

Thank you for this question. In paragraph 1 of the Neural implant section, we have specified that the reference wires were located subdurally, lying on back of the ECoG grids and that during all recordings, signals were referenced to the same reference wire. However, due to the small diameter of the wire (0.07 mm), we were not able to localize it in post-operative imaging. No other re-referencing was performed. We have copied the relevant text from this paragraph below for the reviewer's convenience (modifications in green).

"The device included two subdural reference wires, the tips of which were ~~exposed~~ not insulated to match the recording surface area of the ECoG electrodes. Due to the small diameter of the wires (0.07 mm), it was not possible to localize them on a post-surgical CT scan.

...During all recordings, signals were referenced to the same reference wire and no other referencing was performed."

Commented [A7]: A model of the participant's brain?

Thank you for this suggestion. We have specified in in the caption of Fig. 1b that the depicted image is a virtual reconstruction of the participant's brain (modifications in green).

"(b) Position of both 64-electrode grids overlaid on the left cortical surface of a virtual reconstruction of the participant's brain."

Commented [A8]: define

Thank you for this suggestion. We have defined this in the caption of Fig. 1g (modifications in green):

“A Fast Fourier Transform filter was used to compute the spectral power of the 256 ms buffer, from which the high gamma (HG, 110-170 Hz) log-power (110-170 Hz) was placed into a 1 s running buffer (10 feature vectors).”

Commented [A9]: I count $50 + 2*50 + 3*50 + 6*30 = 480$ training sets rather than 260

Thank you for pointing out this error. We have corrected this number to 480 in paragraph 2 of the Training Task section (modifications in green):

“In total, almost 44 min of data (~~260~~480 trials) ~~was~~were collected for model training.”

Commented [A10]: Since the title of the paper includes “long-term”, a description of how study sessions beyond the training stage were performed should be given. Was there a prescribed schedule for testing? Was all testing done in the laboratory setting?

Thank you for these questions. We would like to clarify that the description in the Data collection section refers to data collected during training and during testing with online BCI use, and that all data were collected in the laboratory (modifications in green).

“All data collection and testing were performed in the laboratory. Neural signals were recorded by the Neuroport system at a sampling rate of 1 kHz. BCI2000 was used to present stimuli during training blocks and to store the data from training and online BCI use with the click detector for offline analysis³⁵.”

Further, we have clarified in the Participant section that the weekly testing schedule was based on experimental progress (relevant text is green in the revised manuscript):

“The experimental team was scheduled to meet with the participant three times each week for training data collection or BCI use. Experimental planning occurred weekly and was informed by task-specific progress.”

Commented [A11]: This occurred for training data only? HG-power alignment did not occur online, correct? Was the benefit of HG-power alignment established?

Thank you for these questions. We have rewritten paragraph 2 in Feature extraction and label assignment to clarify how the rest and grasp labels were assigned on a per-trial basis using trial-averaged HG-power, which itself was computed from shift-aligned power trial rasters. We have copied that paragraph below for the reviewer’s convenience (relevant text is green in revised manuscript):

“We assigned rest and grasp labels to each sample in our training dataset by the following steps. First, for each channel we concatenated segments of HG power across all trials,

where each trial segment ranged from -1 s to 2.5 s relative to the beginning of the visual “Go” cue (Supplementary Fig. 3, Cue-aligned)...”

The alignment of power-trial rasters only occurred offline, as this was used for labeling training data. No such alignment occurred online. Though we are not aware of previous studies that have aligned HG power-trial rasters for the purposes of model training, a recent MEA-based BCI study has used this technique to discover trial-averaged spatiotemporal neural patterns from attempted writing [1 (ref. 4 in revised manuscript)]. Though the signals in [1 (ref. 4 in revised manuscript)] were derived from single-neuron activity rather than population HG activity, the group that developed this time warping technique [2 (ref. 37 in revised manuscript)] noted that it “can be flexibly applied to any multi-dimensional time series, including spike trains, fMRI data, or LFP traces.” Therefore, though we are not aware of ECoG-based studies that have previously employed this method, we believe it is valid for aligning HG power trial rasters.

*[1] Willett, F. R., Avansino, D. T., Hochberg, L. R., Henderson, J. M. & Shenoy, K. V. High-performance brain-to-text communication via handwriting. *Nature* **593**, 249–254 (2021).*

*[2] Williams, A. H. et al. Discovering Precise Temporal Patterns in Large-Scale Neural Recordings through Robust and Interpretable Time Warping. *Neuron* **105**, 246-259.e8 (2020).*

Commented [A12]: Looking into reference 22, what method of re-alignment was used? Shifting or linear warping? Are these methods appropriate for ECoG data?

Both methods of realignment will certainly increase correlation because you are matching peak power to peak power.

Why not use the video of hand movement to perform reaction-time realignment?

Thank you for this question. Please note that in the revised manuscript, this reference number is now 37. We have clarified in the Label assignment that only shift alignment was used (relevant text in green):

*“To account for the inter-trial variability of the participant’s reaction delay to the visual “Go” cue, we temporally re-aligned the HG power across all trial segments using a **shift warping mode**³⁷”*

Further, in the last sentence of this section, we have clarified the reason for not using the video of the hand movement to align the power trial rasters (relevant text is green in revised manuscript):

“We adopted this labeling strategy because it relied only on the visual inspection of neural signals, simulating the lack of ground truth for attempted movements that would be expected for BCI users with Locked-in Syndrome (LIS).”

We believe that this re-alignment method is appropriate for ECoG data (please see response to A11).

Commented [A13]: This distinction seems somewhat arbitrary. Don’t you know when the go vs rest conditions were presented? Why do visual inspection of the realigned averages?

Thank you for the question. The “Go” cue was visually presented to the participant very briefly (for 100 ms), and due to the participant’s reaction delay, the presentation of the “Go” cue did not occur synchronously with the participant’s attempted grasping movement. We visually inspected the realigned trial averages to simulate reduced availability of ground truth signals (please refer to our response in A12). We realized that the participant’s residual movement abilities would likely deteriorate due to ALS, and thus, we developed a method of labeling that does not rely on observing his residual movements.

Commented [A14]: Is the direction to the user given somewhere in the video? It is unclear what selections they are told to make.

We apologize for this confusion. The participant was verbally instructed to navigate and click on arbitrary keys on the medical communication board. We have added the verbal cue at the bottom of the video, which also accurately aligns with the onset of the spoken cue by the experimenter. The participant navigated to all three verbally cued keys correctly in this clip. However, we would like to emphasize that:

“Whether the participant clicked the correct or incorrect key had no bearing on sensitivity, TPF, or FPF as these metrics depended only on whether a click truly occurred following an attempted grasp.”

as we mentioned in Sensitivity and click rates section.

Commented [A15]: Should be $N_{\text{attempted grasps}}$?

Thank you for pointing out this error. We have corrected this term in the Sensitivity and click rates section (modifications in green).

“...where in one session $N_{\text{correct clicks}}$ were the total number of correct clicks and $N_{\text{attempted grasps}}$ were the total number of attempted grasps, and where $N_{\text{correct clicks}} \leq N_{\text{attempted grasps}}$.”

Commented [A16]: What if a letter was omitted? Wouldn’t this make all subsequent correct letters in the wrong position?

Thank you for this question. We have clarified in the Spelling rates section that the participant was instructed to correct any errors (modifications in green):

“Spelling rates were measured by correct characters per minute (CCPM) and correct words per minute (CWPM). Spelled characters and words were correct if they exactly matched their positions in the prompted sentence. For example, if the participant spelled a sentence with 30 characters (5 words) with 1 character typo, only 29 characters (4 words) contributed to the CCPM (CWPM). *The participant was instructed to correct any mistakes before proceeding to type the rest of the sentence.*”

Commented [A17]: Recommend integrating this with the “Model Architecture” subheading above since it pertains to how the classifier was created.

Thank you for this suggestion. We have moved the Cross-validation section under Model architecture and training section.

Commented [A18]: Not clear why you excluded this channel until you dig into the results.

We apologize for this confusion. We amended the respective sentence in paragraph 1 of Channel contributions and offline classification comparisons to describe why we performed a saliency analysis and computed cross-validated classification accuracy with all channels aside from Channel 112 (modifications in green):

“We repeated this process using HG features from all channels except ~~one (channel 112)~~ channel 112, which was located over sensory cortex and showed a relatively high activation compared to other channels during attempted movement. We then repeated this process using HG features from a subset of 12 electrodes over cortical hand-knob...”

Commented [A19]: To results/discussion

Thank you for this suggestion. However, per the “Style and formatting checklist” required by the journal to which we are submitting, we are required to have the Statistics and Reproducibility section as part of the Methods.

Commented [A20]: What does this mean exactly? Achieved performance over a certain level for 111 days?

Thank you for this suggestion. We have clarified this in the last sentence of Long-term usage with a fixed click detector (modifications in green):

“We found that the decoder performance remained robust ~~over a period of~~ 111 days.”

Later, a threshold of 80% is mentioned. This should be put in the here or in the methods.

Thank you for this suggestion. However, we have removed the mention of this pre-set threshold, as this does not hold substantial meaning – for simple click detection, an 80% performance threshold is still poor. For example, there is one day (Day +132, Supplementary Fig. 14) on which the sensitivity passes this 80% threshold but is still substantially lower compared to sensitivities from sessions prior to the drop in detector performance. We have copied the revised sentence in paragraph 1 of Click detector retraining due to transient performance drop for the reviewer’s convenience (modifications in green):

“On Day +118 (Supplementary Fig. 13 for timeline), the detector sensitivity ~~markedly decreased~~ fell below the pre-set performance threshold of 80% (Supplementary Fig. 14), which was likely due to a decrease ~~drop~~ in the movement-aligned event-related

synchronization (ERS) of the HG response across a subset of channels (Supplementary Fig. 15)."

Commented [A21]: The obvious piece missing from decreasing the number of required votes is the tradeoff with specificity. How can you define specificity in this setting to make the story more complete?

Specificity is $TNR = 1 - FPR$. You can calculate FPR using your methodology as the number of clicks that occurred outside of the correct "go" window.

In fact, in the discussion, you mention the calculation of FPR, but I don't believe the results are presented.

Thank you for this suggestion. We use the term "False positive frequency" precisely to account for the tradeoff of in a potential increase in false positives with the increase in sensitivity. We observed an increase in sensitivity from 94.9% to 97.8% when switching from a 7-vote threshold to a 4-vote threshold (modifications in green):

*"...the click detector achieved a median detection sensitivity of 94.9% using a 7-vote threshold, and a **significantly increased** sensitivity of 97.8% when using a 4-vote threshold ($W = -1.898$, $P = 0.057$, two-sided Wilcoxon Rank-Sum test; Fig. 4a)"*

However, we did not observe a corresponding increase in false positive frequency:

"...the median false positive frequency (FPF) was 0.029 per min (1.74 per h) using a 7-vote threshold and 0.101 per min (6.03 per h) when using a 4-vote threshold ($W = -1.280$, $P = 0.20$, two-sided Wilcoxon Rank-Sum test; Fig. 4b)."

Specificity (and for the same reasons described below, FPR as well) was not possible to calculate for our task. Briefly, specificity is defined as:

$$TNR = \frac{TN}{FP + TN} = 1 - FPR = 1 - \frac{FP}{FP + TN}$$

where FP and TN are the number of false positives and number of true negatives, respectively. We were able to count the number of FPs using the metric we defined in the Sensitivity and click rates section (i.e., the number of clicks that occurred outside the 1.5 s period after attempted grasp onset). However, the concept of a true negative detection (a correctly omitted click) does not exist in the continuous time domain because there do not exist discrete countable instances of correctly omitted clicks.

We used the term "false positive rate" in the discussion accidentally, and we have now replaced it with "false positive frequency" (defined in the Sensitivity and click rates section) in the first sentence of paragraph 3 (modifications in green):

*"Our model detected intended clicks with high sensitivity and low **false positive frequency (FPF) false-positive-rates.**"*

Commented [A22]: The long plots without gridlines makes it hard to see any trends that may occur. Suggest adding horizontal grid lines to all sub-plots. Suggest replacing plot b with Specificity. For c:, you have have a single y axis called “latencies”, with a key for “click” and “decision” For d: you can have a singly axis called “correct selection rate”, with a key for “words” and “characters”

Thank you for these suggestions. We have incorporated gridlines into all the subfigures to make trends more visible, modified the axis labels and included legends in Fig. 4c and 4d (formally Fig. 3c and 3d). However, we did not replace TPF and FPF in Fig. 4b with specificity because of our reasons described in A21. Modifications to the figure caption are in green.

“Figure 4 | Long-term switch-scanning spelling performance. Across all subplots, triangular and circular markers represent metrics using a 7-vote and 4-vote voting threshold, respectively. (a) Sensitivity of *grasp click* detection for each session. *Dashed line delineates 100% sensitivity.* (b) True-positive and false-positive frequencies (TPF and FPF) measured as detections per minute. *Dashed line delineates 0 FPF.* (c) Average latencies with standard deviation error bars of grasp onset to algorithm detection and to on-screen click. The averages and standard deviations were computed from latency measurements across all spelling blocks from one session using the same voting threshold. Using 7-vote and 4-vote voting thresholds, on-screen clicks happened an average of 207 ms and 203 ms, respectively after detection. Note that detection latencies were not registered in the first six sessions. (d) Correct characters and words per minute (CCPM and CWPM).”

Commented [A23]: First mention of these Thank you. We have described the cognitive and neurological tests in comment A4 and have described these in paragraph 3 of Participant section.

Commented [A24]: Why this four month window? Was a root-cause analysis performed over this period?

Thank you for this question. During this period, we investigated the possible causes for the observed deviations in HG response. However, we found no reason for these signal changes. We have added a sentence to the caption of Supplementary Fig. 13 (formally Supplementary Fig. 10) to clarify this (relevant text is green in revised manuscript):

“Between Days 217 and 309 post-surgical implantation we investigated possible causes for the signal deviation.”

Commented [A25]: Which day? 309?

Thank you for this question. Yes, the new training data was collected on Day 309 post-surgical implantation. We have clarified this in paragraph 2 of Click detector retraining due to transient performance drop (relevant text in green):

“The new click detection algorithm used a total of 15 min of training data, which was all collected within one day, six days before BCI use (Day 309 post-surgical implantation, Supplementary Fig. 17a)...”

Commented [A26]: Suggest replacing real-time with online decoding

Thank you for this suggestion. We have replaced “real-time” with “online” in the main text and Supplementary Information (modifications in green).

Commented [A27]: Again-related to the question of specificity.

Please refer to our response to A21.

Commented [A28]: Aka high specificity. Why not use this terminology?

Please refer to our response to A21.

Commented [A29]: Why is this reference used? Are you referencing the results of this study, or just the device, which was shared across refs 40 and 1?

Thank you for this question. In rewriting the Discussion, we have eliminated that sentence and the reference entirely.

Commented [A30]: Repetitive of results?

Thank you for this question. In rewriting the Discussion, we have eliminated this paragraph because it was indeed repetitive of the results.

Commented [A31]: Why was an imagined hand grasp not attempted?

We appreciate the reviewer's question. Please see our response to A4.

Commented [A32]: Not needed?

Thank you for the suggestion. However, by request of the editor, we must keep all information regarding the clinical trial.

Commented [A33]: This can be included in main text body, in the participant section.

Thank you for the suggestion. We have included this in the Participant section of the main text (modifications in green):

"All results reported here were based on data from the first ~~and only~~ participant ~~to date~~ in the CortiCom trial. The participant gave written consent after being informed of the nature of the research and implant related risks. ~~The experimental team was scheduled to meet with the participant three times each week for training data collection or BCI use. Experimental planning occurred weekly and was informed by task-specific progress.~~ To date this participant has had no serious or device-related adverse events, and thus the primary outcome of the CortiCom trial has been successful. ~~The secondary outcomes of the CortiCom trial are reported, in part, here; specifically, our success rate and latency are reported in terms of click detection accuracy and time from attempted movement onset to click.~~ The participant has consented to continue the study. At this time the device has been implanted for more than a year and continues to be used for research purposes."

Commented [A34]: Not needed?

Thank you for the suggestion. However, by request of the editor, we must keep all information regarding the clinical trial.

Commented [A35]: Which day in timeline of supp figure 10?

We apologize for the confusion. All three blocks were collected on Day 309, post-surgical implantation. We have included a reference to Supplementary Fig. 17 (formally Supplementary Fig. 14) in Supplementary Note 5 and copied it below for the reviewer's convenience (modifications in green):

"All three blocks were collected on the same day (Day 309 post-surgical implantation, Supplementary Fig. 17a)."

Commented [A36]: This window is slightly different than the 0.3-1.1 used prior. Reason for this?

Thank you for this question. The trial-averaged channel traces of the training data for the new fixed model were similar to those in Supplementary Fig. 6, and the onset and offset of the HG activity occurred roughly after the same amount of time. However, we decided to change the bounds of our grasp labels to 0.4 s and 1.2 s (from 0.3 s to 1.1 s) to avoid misclassification of grasp samples which did not contain a large deviation of HG power in the leading edge of the historical time window compared to the baseline period occurring shortly before. We plan to investigate the optimal bounds for assigning grasp labels in the future.

Commented [A37]: Should note that the relative Go and ISI times depicted are not to scale.

Thank you for the suggestion. We have included this in the caption of Supplementary Fig. 1 (modifications in green):

“One trial consisted of one 100 ms “Go” stimulus ($t_{Go} = 0.1$ s, not to scale for the purpose of clear depiction) followed by an interstimulus interval (ISI) during which a ~~centered~~ white crosshair ~~appeared~~remained in the center of the monitor for the duration of the ISI.”

Commented [A38]: How were these channels defined? How was the per-trial shift calculated in each channel?

Thank you for this question. We have rewritten paragraph 2 in Features extraction and label assignment and have clarified that the channels were chosen qualitatively. We’d also like to clarify that the shift that occurred for each trial was the same for all power trial rasters of all channels (relevant text in green).

“To account for the inter-trial variability of the participant’s reaction delay to the visual “Go” cue, we temporally re-aligned the HG power across all trial segments using a shift warping model³⁷ (Supplementary Fig. 3, Re-aligned). This model was trained on only a subset of highly modulated channels (determined qualitatively; Supplementary Fig. 3 caption) to decrease the potential influence of artificial patterns from low-modulation channels when re-aligning trial segments. Note that for each trial, the resulting temporal re-alignment was applied similarly across all 128 channels.”

Commented [A39]: You do not show the power spectrograms for the Speech Grid. Based on the negligible correlation increase from peak shifting, I imagine the spectral responsiveness in this grid was not as pronounced. For completeness, you might want to add to supplementary figure 2. Or for succinctness, you may want to remove b) from this figure.

Thank you for this suggestion. We have removed Supplementary Fig. 4b and have copied the new Supplementary Fig. 4 below for the reviewer’s convenience (caption modifications in green):

“Supplementary Fig. 4 | Correlation increase of HG power in re-aligned signals. The change in correlation between the re-aligned and cue-aligned HG power trials for each electrode of the upper limb grid. Inter-trial HG power correlations—Electrodes in the upper limb grid (a) generally increased for all electrodes their inter-trial HG power correlation, whereas those in the speech grid (b) generally stayed the same. Electrodes in the speech grid (not shown) had negligible changes in inter-trial HG power correlation. The central sulcus (CS) is delineated by a thick black line and widens at the top such that electrodes channels 111, 119, and 127 are over it. The pre-central sulcus (Pre-CS) is delineated by a thick green line.”

Commented [A40]: Here you are showing that a 1-s moving average is used to calculate 100 ms feature vectors. There are 8 vectors corresponding to the Go condition ending at 0.3 – 1.1 s.

What do the vertical bars at 0.2 s and 1.2 s show? I suggest removing all 4 vertical bars.

Thank you for these suggestions. We would like to clarify that the 1 s window is not used to compute a moving average. Rather this window represents 1 s of time history from all the channels for training. We have clarified this in the caption of Supplementary Fig. 6 (modifications in green):

“Supplementary Fig. 6 | Labeling trial-averaged re-aligned HG power. Trial-averaged re-aligned HG power is shown for each of the channels that were used to compute the per-trial shift. These averaged traces were used to determine the bounds for grasp labels on a per-trial basis. For each trial, each sample 100-ms feature vector between 0.3 s and 1.1 s post-cue was labeled grasp while all other time points were labeled rest. For the training label at each time point, the corresponding training data is the 1 s sequence (previous 10 samples) of historical time features up until and including the time point of the that training label. For example, the training data for the grasp label at 1.0 s (dark green outline) is the 1 s sequence of historical time features from 0.1 to 1.0 s (large box with dark green outline). Though only the HG power traces for the above subset of channels were used to inform the assignment of grasp training labels, features from all channels were used as training data.”

We eliminated the vertical bars at 0.2 s and 1.2 s as they were not functionally used for applying training labels. However, we have kept the vertical bars at 0.3 s and 1.1 s to show the bounds by which grasp labels were assigned on a per-trial basis.

Commented [A41]: Not sure what this means. 9 channels were used to compute the per-trial shift?

Thank you for this question. Yes, these 9 channels were used to compute the per-trial shift.

Commented [A42]: See comments about suggested changes to figure 3

Thank you for this suggestion. We have incorporated into all the subfigures gridlines to make trends more visible, modified the axis labels and included legends. However, we did not replace TPF and FPF shown Supplementary Fig. 14b (formally Supplementary Fig. 11b) with specificity because for our reasons described in A21. Modifications to the figure caption are in green.

“Supplementary Fig. 14 | Performance decline roughly 4 months post-training. For each day a 4-vote threshold was used. **(a)** Sensitivity of grasp detection for each session. **(b)** True-positive and false-positive frequencies (TPF and FPF) measured as detections per minute. **(c)** Average latencies with standard deviation error bars of grasp onset to *algorithm* detection and to *on-screen* click. **(d)** Correct characters/words per minute (CCPM/CCWP).”

Commented [A43]: See comments about suggested changes to figure 3

Thank you for this suggestion. We have incorporated into all the subfigures gridlines to make trends more visible, modified the axis labels and included legends. However, we did not replace TPF and FPF shown Supplementary Fig. 18b (formally Supplementary Fig. 15b) with specificity because for our reason described in A21. Modifications to the figure caption are in green.

“Supplementary Fig. 19 | Switch-scanning spelling performance with a retrained fixed click detector. All performance metrics are shown using the 6-vote threshold. (a) Sensitivity of grasp detection for each session. (b) True-positive and false-positive rates (TPF and FPF, respectively) measured as detections per minute. (c) Average latencies with standard deviation error bars of grasp onset to algorithm detection and to on-screen click. (d) Correct characters/words per minute (CCPM/CWPM).”

Commented [A44]: This figure would be more useful if paired with the grid location over annotated brain regions. Right now it can be replaced with the statement “A 4x3 subset of electrodes covering cortical hand knob [achieved similar performance in offline testing]”

Thank you for this suggestion. However, Supplementary Fig. 19c (formally, Supplementary Fig. 16c) is simply a normalized representation of saliency values depicted by circular markers in Fig. 5e (formally Fig. 4e). The benefit to showing these values in a descending bar graph is to more clearly depict the narrower range in saliency values using only channels over cortical hand region compared to the

broader range of saliency values when using all electrodes or only omitting channel 112 (Supplementary Figs. 19a and 19b, respectively).

Reviewer #2 (Remarks to the Author):

Dear Nathan Crone and colleagues,

With great interest I read your manuscript entitled "A click-based electrocorticographic brain-computer interface enables long-term high-performance switch-scan spelling". This research demonstrates the effective and long-term use of a brain-controlled communication aid that allowed a person with amyotrophic lateral sclerosis (ALS) to select letters or icons on a computer screen. Novel features of this research include training a classifier on a limited amount of training data (44 minutes of data were acquired across 4 different days) and testing it across a period of 3 months (18 different test days) after without retraining the classifier. The use of chronically implanted ECoG is also relatively new in the field of brain-computer interfacing (BCI).

The achieved classification speed and performance outperform those of traditional P300 spellers and are comparable to those of other BCI spelling applications. For example, Sutter (1992) achieved a performance of 10-12 bits (full words or characters) per minute using a visual ECoG-driven brain-computer interface. Later on, this same approach based on code-modulated visual evoked potentials (cVEP) was effectively demonstrated in a group of people with ALS using EEG, also yielding average speeds of 10 characters per minute (Verbaarschot et al., 2021).

Demonstrations of effective BCI performance within the target population are rare. Even though the current manuscript reports results of one participant only, this relatively long term study is a relevant contribution to this field.

Major remarks:

1) The advantage of using invasive neuroimaging methods such as ECoG are not immediately clear to me. Comparable results have been achieved using non-invasive methods such as EEG. Could you elaborate what benefits the use of ECoG provide over non-invasive methods?

We thank the reviewer for this question. We are aware of the levels of BCI control that have been accomplished with EEG-based systems and agree that this technology should be developed to its fullest extent, as not every person with communication impairment will want, or will be able to receive, an implanted system. Yet, we would like to emphasize that implanted BCIs based on ECoG electrodes have important advantages for daily use by people with a neurological need for BCI. First, implanted ECoG-based BCIs do not require daily application of external sensors and offer the potential for use with only minimal caregiver involvement [1 (ref. 1 in revised manuscript)]. Second, ECoG-based BCIs can potentially offer stable functionality for up to at least 3 years [2 (ref. 18 in revised manuscript)] (7 years in Vansteensel et al., under review). Third, implanted electrodes are potentially available 24/7 to provide BCI functionality whenever the user wishes to use the system. These factors make ECoG-based BCIs particularly attractive as a communication technology for people with severe motor and communication impairment. The referenced unpublished papers are available upon reviewer's request.

We would also like to emphasize that this study shows some of the first results from a larger clinical trial, which is largely aimed at optimizing functionality of ECoG-based BCI over extended use periods by

leveraging the stability of ECoG signals. Thus, we aim to build on this work to provide more continual BCI use in the participant's home and work toward minimal caregiver/technician interference.

Taken within the context of this clinical trial, the participant's ECoG implant is intended to allow control strategies beyond single-command click-decoding. Indeed, as previous work by us [3-6] and others [7, 8] has shown, multi-command control and cursor movements are control strategies which can both be leveraged by ECoG BCIs. Due to its higher spatial resolution and signal quality over EEG, we believe that ECoG has greater potential for exploring these control strategies. Nonetheless, it is important to show that basic single-command decoding can be used to reliably maintain communication over extended periods of time, especially as the participant's condition progresses.

[1] Vansteensel, M. J. et al. Fully Implanted Brain–Computer Interface in a Locked-In Patient with ALS. *N Engl J Med* **375**, 2060–2066 (2016).

[2] Pels, E. G. M. et al. Stability of a chronic implanted brain-computer interface in late-stage amyotrophic lateral sclerosis. *Clinical Neurophysiology* **130**, 1798–1803 (2019).

[3] Hotson, G. et al. Individual finger control of a modular prosthetic limb using high-density electrocorticography in a human subject. *Journal of Neural Engineering* **13**, 026017 (2016).

[4] Fifer, M. S. et al. Simultaneous Neural Control of Simple Reaching and Grasping With the Modular Prosthetic Limb Using Intracranial EEG. *IEEE Transactions on Neural Systems and Rehabilitation Engineering* **22**, 695–705 (2014).

[5] Bleichner, M. G. et al. Give me a sign: decoding four complex hand gestures based on high-density ECoG. *Brain Struct Funct* **221**, 203–216 (2016).

[6] Thomas, T. M. et al. Decoding native cortical representations for flexion and extension at upper limb joints using electrocorticography. *IEEE Trans Neural Syst Rehabil Eng* (2019) doi:10.1109/TNSRE.2019.2891362.

[7] Silversmith, D. B. et al. Plug-and-play control of a brain–computer interface through neural map stabilization. *Nat Biotechnol* **39**, 326–335 (2021).

[8] Degenhart, A. D. et al. Remapping cortical modulation for electrocorticographic brain–computer interfaces: a somatotopy-based approach in individuals with upper-limb paralysis. *J. Neural Eng.* **15**, 026021 (2018).

Moreover, could you compare the results of this study to other commonly available communication aids that a person with ALS may use, such as eye-tracking or applications that are directly controlled by residual movement?

Thank you for this question. We would like to first note that control strategies based on eye movements (such as eye-tracking or eye-gaze + attention) for people living with ALS can be expected to worsen if they survive into later stages of the disease, as eye control can deteriorate. However, we have included a study in paragraph 5 of the Discussion which reports typing speeds of 17 characters per minute using eye-tracking alone (modifications in blue):

“...while typing speeds of 17 characters per min using eye-tracking alone have been reported⁵⁸. However, control strategies based on eye movements may cause eyestrain during long periods of use²⁷⁻²⁹ and worsen as residual eye movements can deteriorate in late-stage ALS³⁰⁻³³.”

In paragraph 5 of the Discussion, we also report that the spelling rates in this study were comparable to those from previous studies in which participants leveraged residual movements. We have clarified (in blue) that residual movements were leveraged in those previous studies:

*“...our spelling rates were comparable to those from other clinical populations who have used switch scanning keyboards **by leveraging residual movements (and without a BCI)**, including people living with ALS⁴⁹ or other causes of motor impairments⁵⁰.”*

Further, as is suggested by the reviewer in Major remark 5, non-invasive BCIs leveraging code-modulated visually evoked potentials (cVEPs) are also estimated to produce a character per minute (CPM) rate of 10-12 when used by ALS participants [2, (ref. 54 in revised manuscript)]. We have acknowledged this mode of communication in paragraph 4 of the Discussion (relevant text is blue in revised manuscript):

“...non-invasive BCIs using visually evoked potentials are estimated to produce comparable spelling rates⁵² and can potentially be trained with little or no neural data⁵³.”

*[1] Vansteensel, M. J. et al. Fully Implanted Brain–Computer Interface in a Locked-In Patient with ALS. *N Engl J Med* **375**, 2060–2066 (2016).*

*[2] Verbaarschot, C. et al. A visual brain-computer interface as communication aid for patients with amyotrophic lateral sclerosis. *Clinical Neurophysiology* **132**, 2404–2415 (2021).*

*[ref. 60 in revised manuscript] Pasqualotto, E. et al. Usability and Workload of Access Technology for People With Severe Motor Impairment: A Comparison of Brain-Computer Interfacing and Eye Tracking. *Neurorehabil Neural Repair* **29**, 950–957 (2015).*

Does an invasive BCI have a clear advantage over these less invasive and less expensive methods? Since your approach currently relies on residual movement, the direct benefit is not clear to me.

We will refer the reviewer to the answer in the first part of Major remark 1, where we have expanded on the motivation for using invasive BCI. However, we would like to clarify that our BCI system relies on the modulation of the neural signals when the participant attempts to make a grasp, and not the residual grasping movement itself. This is an important distinction because we expect that we will be able to continue decoding modulated activity from attempted grasps even as the participant loses mobility due to disease progression. The feasibility of this approach has been demonstrated in Vansteensel et al. 2016 [1 (ref. 1 in revised manuscript)] and Pels et al. 2019 [2 (ref. 18 in revised manuscript)] where a participant with ALS was able to use a brain-click BCI for more than 3 years. We refer to these studies in the rewritten Introduction:

“...a participant with ALS attempted hand movements to generate brain clicks, in turn controlling a switch-scanning spelling application. These brain clicks were detected from a single pair of electrodes on the surface of the hand area of the contralateral motor cortex¹. Though the participant used these brain clicks to communicate in her daily life for more than 3 years¹⁸.”

[1] Vansteensel, M. J. et al. Fully Implanted Brain–Computer Interface in a Locked-In Patient with ALS. *N Engl J Med* **375**, 2060–2066 (2016).

[2] Pels, E. G. M. et al. Stability of a chronic implanted brain-computer interface in late-stage amyotrophic lateral sclerosis. *Clinical Neurophysiology* **130**, 1798–1803 (2019).

2) I miss information on the opinion of the user in this manuscript. What did the participant think of the spelling application? Did they ever use it autonomously? Would they want to use it in their daily life? Did they enjoy using the application? Or was it frustrating?

We appreciate the reviewer’s concern for the participant’s experience in using the spelling application. The participant used the spelling application under the supervision from the study team in the laboratory. We have included metrics of the participant’s experience with the spelling application administered via the NASA Task Load Index (NASA-TLX), a commonly used set of workload scores describing the mental, physical, and temporal demand of a task as well as the performance, effort, and frustration. We have added a description these metrics in an additional Methods section, copied below (relevant text is blue in revised manuscript):

“Cognitive workload

In order to evaluate the participant’s experience using the switch-scanning spelling application, we asked the participant to complete the NASA task load index (NASA-TLX) questionnaire^{44,45} using the NASA-TLX iOS application, a commonly used set of questions to evaluate a participant’s mental, physical, and temporal demand of a task as well as the perceived performance, effort, and frustration of a task. These categories were each scored from 0-100 where lower and higher scores corresponded, respectively to less and more of each of the six above-mentioned characteristics.”

We report these metrics in the Click detector retraining due to transient performance drop off section, copied below (relevant text is blue in revised manuscript):

“We additionally evaluated the participant’s subjective cognitive workload of switch-scanning spelling with the click detector using the NASA-TLX iOS application. Across the six sessions using the retrained click detector the participant reported scores of 7.5 ± 2.7 (mean \pm standard deviation) for mental demand, 8.3 ± 2.6 for physical demand, 5.8 ± 3.7 for temporal demand, 6.7 ± 5.2 for performance, 6.7 ± 5.2 for effort, and 6.7 ± 2.6 for frustration. These low scores indicate that the participant did not have difficulty in controlling the switch-scanning spelling application via click-detection.”

3) It is not clear to me why a relatively complex decoding strategy was used for this application. You are trying to decode grasp intentions vs. rest, which induces relatively large changes in high gamma power activity. It seems to me that a simple linear decoder may also do this job. What were the reasons for using a non-linear neural network approach instead?

The primary objective of this study was to demonstrate that a decoder trained on a small amount of data could maintain high performance for an extended BCI use period. While the choice of the decoder is an important factor for achieving this goal, we believe that it is primarily the long-term stability of the modulation in ECoG signals which makes this possible. We agree with the reviewer that for the simple task of detecting clicks from attempted grasping movements, a simple linear classifier might have performed just as well. Nevertheless, our motivations for training an LSTM were two-fold:

1) This study is part of a larger clinical trial in which we do not plan to limit upper-limb classification for use only in click-detections. To this end, we aimed to build the model training pipeline such that in the future we could train more complex models for tasks in which the temporal domain would significantly contribute to decoder performance.

2) We aimed to allow the participant to use a high-performing BCI as soon as possible, and to this end we anticipated that a non-linear classifier would achieve higher performance than a linear model due to the advantage of recognizing temporal patterns in neural activity.

We have added these reasons as the first paragraph of Model architecture and training and have copied them below for the reviewer's convenience (relevant text is blue in revised manuscript):

"We used a recurrent neural network (RNN) for classifying rest vs. grasp. As this study is part of a larger clinical trial, we aimed to build the model training pipeline such that in the future we could train more complex models for tasks in which the temporal domain would significantly contribute to decoder performance. Additionally, we aimed to allow the participant to use a high-performing BCI as soon as possible, and to this end we anticipated that a non-linear classifier would achieve higher performance than a linear model due to the advantage of recognizing temporal patterns in neural activity."

Moreover, specific design choices for the network are not well argued (number of layers, number of units per layer). Could you elaborate your design choices in the manuscript?

We prioritized the participant using the BCI as soon as possible and we used the neural network hyperparameters that we initially chose when testing the model offline using cross-validation. We did not optimize for these hyperparameters as we believe that such optimizations would not have significantly improved the model performance for the simple task of detecting clicks. Further, given that the model with this set of hyperparameters achieved high cross-validation accuracy offline and high performance in real-time, we continued using these hyperparameters throughout the rest of the experiment.

4) ALS affects motor control and is reflected by changes in motor areas in the brain. Why do you choose to use a movement for BCI control and record from motor areas in the brain as both these things are likely affected as ALS progresses? Would another, less affected brain area not be a better recording site? And would a task that the participant can always perform with ease (response to sensory input, mental calculation, imagined sensations) not be a better and more pleasant task for them?

We were encouraged to further investigate the decoding potential from motor cortex due to the success of a previous clinical trial (Vansteensel et al., 2016) in which the authors showed that a participant with ALS could control a switch-scanning spelling application via attempted hand movements [1, (ref. 1 in revised manuscript)]. The authors in this study used signals from the hand region of motor cortex and demonstrated that this control strategy was stable for at least three years [2, (ref. 18 in revised manuscript)]. We cite these studies as motivation for our work in paragraph 3 of the Introduction:

“...a participant with ALS attempted hand movements to generate brain clicks, in turn controlling a switch-scanning spelling application. These brain clicks were detected from a single pair of electrodes on the surface of the hand area of the contralateral motor cortex¹. Though the participant used these brain clicks to communicate in her daily life for more than 3 years¹⁸...”

We were similarly encouraged that another clinical trial working with ALS participants has more recently leveraged signals from the motor cortex via the superior sagittal sinus to control a variety of click-based applications [3, 4 (refs. 3, 19, respectively in revised manuscript)]. We have similarly referred to these studies in paragraph 2 of our Introduction:

“...participants with ALS (or primary lateral sclerosis) were implanted with an endovascular stent-electrode array for detecting brain. Brain clicks were generated by attempted foot movements and were used to select a particular icon or letter on a computer screen after navigating to it via eye-tracking (ET)¹⁸. As a result, participants were able to achieve high spelling rates and required 1-12 sessions of training with their brain click BCI before long-term use.”

Therefore, we believe that investigating the degree to which these signals can be leveraged for use of participants with ALS constitutes a significant contribution to the BCI field.

Further, we are not aware of any studies in which signals from cortical regions outside of the sensorimotor cortex were used for successful BCI control with ALS participants. Therefore, we consider that implants in ALS participants over such cortical regions remain largely exploratory. Since our clinical trial is largely aimed at optimizing functionality of ECoG-based BCI, we thus chose to implant our ECoG grids over cortical regions (sensorimotor cortex) which prior work has shown favorable to this aim. However, it is possible that with future clinical trial participants, we may implant electrodes in some of these exploratory regions, on the condition that we have already confidently established a baseline BCI functionality with sensorimotor cortical coverage. In this case, it may be worth investigating alternative control strategies for generating clicks as the reviewer suggests. We have included this possibility at the bottom of paragraph 7 of the Discussion:

“Click-detection using signals from less affected regions of the cortex and control strategies that are not based on attempted movement, but rather on imagined movements or responses to sensory input could serve as alternative control strategies.”

[1] Vansteensel, M. J. et al. Fully Implanted Brain–Computer Interface in a Locked-In Patient with ALS. *N Engl J Med* **375**, 2060–2066 (2016).

[2] Pels, E. G. M. et al. Stability of a chronic implanted brain-computer interface in late-stage amyotrophic lateral sclerosis. *Clinical Neurophysiology* **130**, 1798–1803 (2019).

[3] Mitchell, P. et al. Assessment of Safety of a Fully Implanted Endovascular Brain-Computer Interface for Severe Paralysis in 4 Patients: The Stentrode With Thought-Controlled Digital Switch (SWITCH) Study. *JAMA Neurol* **80**, 270 (2023).

[4] Oxley, T. J. et al. Motor neuroprosthesis implanted with neurointerventional surgery improves capacity for activities of daily living tasks in severe paralysis: first in-human experience. *J NeuroIntervent Surg* **13**, 102–108 (2021).

5) The main novelty of this paper is the fact that the classifier is trained on limited test data, and used across a large period of time without any retraining. However, non-invasive methods such as the c-VEP based BCI speller (Verbaarschot et al., 2021) do not require any training data. What are the benefits of your approach to such spellers?

Thank you for this question. The studies regarding code-modulated visually evoked potentials (cVEPs) to which the reviewer references us use non-invasive scalp EEG and deserve acknowledgment as a potential avenue for developing spelling applications. We have now referenced both the Verbaarschot et al., 2021 and the Thielen et al., 2021 studies [1, 2 (refs. 54, 55 in revised manuscript)] in paragraph 4 of the Discussion:

“...non-invasive BCIs using visually evoked potentials are estimated to produce comparable spelling rates⁵² and can potentially be trained with little or no neural data⁵³”

One of our motivating factors for using an implanted ECoG device is to reduce the amount of time a research technician must spend with the BCI user for not only updating or recalibrating the decoder but also for setting up the appropriate hardware and software components. Though creating a spelling application without requiring any training data (Thielen et al, 2021) is an important step toward minimizing this interaction time, frequent application and maintenance of external EEG sensors by a caregiver or technician would still be necessary. On the other hand, ECoG electrodes do not necessitate such maintenance, and in our study, the only required connection was the attachment of a recording headstage (with a micro-HDMI cable) to the surgically implanted pedestal. Notably, the communication strategy demonstrated in this paper did not require daily training and only required initial collection of training data and then daily calibration that required 1 minute. In the future we envision that this connection will be maintained at least over multiple days (perhaps longer) without requiring such daily calibration. However, for our ECoG-based BCI system to function using signals from our chosen site of implantation (see response to Major remark 4 for reference), it was necessary to train our decoder with prior data; we will continue to explore strategies for minimizing this amount of training data.

Further, we would like to briefly acknowledge that the c-VEP approach ultimately relies on reliable eye gaze to focus on the character that the user wishes to type. Though eye movements are initially preserved in ALS, they can deteriorate as the disease progresses as we mention in the bottom of paragraph 5 of the Discussion:

“However, control strategies based on eye movements may cause eyestrain during long periods of use^{27–29} and worsen as residual eye movements can deteriorate in late-stage ALS^{30–33}”

Though the residual upper-limb movements of the participant in our study will also deteriorate, we expect that modulation of signals in the upper-limb motor cortex from such attempted movements would still be observed and usable for BCI on a long timescale [3,4 (refs. 1,18 in revised manuscript)]. Note that even though Pels et al., 2021 [4] observed a steady decrease in high frequency band power, this decrease occurred over a time scale of roughly three years and modulation of these signals remained usable for BCI control during that time period. We have acknowledged this in paragraph 2 of the Introduction as motivation for our study:

“...a participant with ALS attempted hand movements to generate brain clicks, in turn controlling a switch-scanning spelling application. These brain clicks were detected from a single pair of electrodes on the surface of the hand area of the contralateral motor cortex¹. Though the participant used these brain clicks to communicate in her daily life for more than 3 years¹⁸...”

[1] Verbaarschot, C. et al. A visual brain-computer interface as communication aid for patients with amyotrophic lateral sclerosis. *Clinical Neurophysiology* **132**, 2404–2415 (2021).

[2] Thielen, J., Marsman, P., Farquhar, J. & Desain, P. From full calibration to zero training for a code-modulated visual evoked potentials brain computer interface. *J. Neural Eng.* (2021) doi:10.1088/1741-2552/abecef

[3] Vansteensel, M. J. et al. Fully Implanted Brain–Computer Interface in a Locked-In Patient with ALS. *N Engl J Med* **375**, 2060–2066 (2016).

[4] Pels, E. G. M. et al. Stability of a chronic implanted brain-computer interface in late-stage amyotrophic lateral sclerosis. *Clinical Neurophysiology* **130**, 1798–1803 (2019).

Minor remarks:

(1) Line 36: please clarify what happened when you re-trained the classifier. You collected new training data on that day and tested in the following 21 days? Or did you train and test on the same day?

Thank you for this suggestion. We have clarified in the Abstract and in paragraph 2 of Click detector retraining due to transient performance drop that the new training data was collected six days before BCI use. We have copied the modified text below in both locations for the reviewer’s convenience (relevant modifications in blue):

Abstract

"Though a transient reduction in signal power modulation interrupted testing with this fixed model, a new click decoder can achieve comparable performance despite being trained with even less data collected six days before BCI use (< 15 min, within one day)."

Click detector retraining due to transient performance drop

"The new click detection algorithm used a total of 15 min of training data, which was all collected within one day, six days before BCI use (Day 309 post-surgical implantation, Supplementary Fig. 17a)"

(2) Line 50: Add that the training data was recorded on different days than you tested. For example: "Our algorithm was trained using less than one hour's worth of brain signals recorded several days prior to testing the algorithm. The algorithm then performed reliably for a period of three months without any retraining."

Thank you for this suggestion. We have included it in the Plain Language Summary and copied it below for the reviewer's convenience (modifications in blue):

"Our algorithm was trained using less than one hour's worth of recorded brain signals recorded several days prior to testing the algorithm. The algorithm then performed reliably for a period of three months without any retraining."

(3) Line 62-63: you comment on the use of micro-electrodes on the long term, but micro-electrode arrays have been shown to record reliably for up to 7 years in a row (Hughes et al., 2020)! In addition, deep-brain stimulation electrodes have been shown to reliably perform for extended periods of time too.

We appreciate the reviewer referring us to the Hughes et al. (2020) study. However, even in that study, we found that the authors report "the number of electrodes with high-amplitude recordings decreased significantly over time on both platinum-motor and SIROF-sensory electrodes" and indeed have gone on to show that "all signal quality metrics decreased over time" as they originally stated in their preprint. We have, however, incorporated the Hughes et al. (2020) [ref. 14 in revised manuscript] study into our Introduction as a motivating factor for ECoG implantation.

"Although sophisticated capabilities of MEA-based BCIs have been reported, signal attrition¹²⁻¹⁴ may affect long-term performance..."

We also thank the reviewer for encouraging us to consider acknowledging electrodes for deep-brain stimulation (DBS). However, we believe that the comparison to such electrodes is out of the scope of our current study. Though these electrodes (and perhaps other electrodes for purposes of which we may not be aware) may perform reliably for their use cases over extended periods of time, we are not aware of any study in which long-term functionality of such electrodes was analyzed within the context of brain-computer interface research.

(4) Line 67: please define what you mean by "chronic". How long does the implant need to exist for it to become "chronic"?

We thank the reviewer for this question. We have clarified this term to mean greater than 30 days in paragraph 1 of the Introduction (modifications in blue):

“However, the utility of ECoG for chronically (> 30 days) implanted BCIs has only been tested in a few participants.”

(5) Line 148-149: what task did the participant perform to assess the relevant cortical areas for implantation?

Thank you for this question. In paragraph 2 of the Neural implant section we have added the tasks that the participant was instructed to perform during fMRI, we describe the use of somatosensory evoked potentials to identify the central sulcus, and the use of vibrotactile stimulation of individual fingers for high gamma responses (modifications in blue):

“The locations of targeted cortical representations were estimated prior to implantation using anatomical landmarks from a pre-operative structural MRI, functional MRI (sequential attempted finger tapping, tongue movement, and humming), ~~and~~ intraoperative somatosensory evoked potentials, and intraoperative high gamma responses to vibrotactile stimulation of the individual fingers.”

(6) Figure 1: can you increase the size of subfigure h? It is a bit small to see clearly

Thank you for the suggestion. We have increased the size of the left and right panels of subfigure h (top: before, bottom: after):

We have also added a sentence in the caption of subfigure h to clarify the small text in the panels: “The example sentence shown is “the birch canoe slid on the smooth planks”

(7) Line 188: the ISI is chosen between a lower an upper bound. What were these bounds?

We apologize for this confusion. We have clarified in paragraph 2 of Training task that the reader may refer to Supplementary Table 1 for the ISI for each training block (modifications in blue):

“The length of each ISI was randomly chosen to vary uniformly between a lower and upper bound (Supplementary Table 1) to reduce anticipatory behavior.”

We have also copied Supplementary Table 1 below for the reviewer’s convenience (modifications in blue):

	Blocks	Trials/set	Number of sets	Post-set “rest”	ISI (seconds)	Total training time (min)
Session 1	1	50	1	No	3.5 - 4.5	3.77
Session 2	2	50	1	No	3.5 - 4.5	7.57
Session 3	1	50	3	Yes	3 - 6	14.09
Session 4	2	30	3	Yes	3 - 6	18.49

Supplementary Table 1 | Experimental parameters for training data collection. For each block during sessions 1 and 2 the participant performed one set of 50 trials whose ISIs were jittered between 3.5 - 4.5 s. For each block during sessions 3 and 4, we introduced a 90 s rest period after each of three sets of trials during which the participant was asked to remain still and fixate his eyes on a Rest stimulus. On session 4, we reduced the number of trials ~~per~~ set from 50 to 30 due to the participant’s difficulty in focusing throughout the duration of the task. The ISI for sessions 3 and 4 was jittered between 3 - 6 s to further reduce anticipatory behavior. The above sessions were used to train the original click detector.

(8) Line 236: what does "randomly downsample" mean? How can the rest class be overrepresented if you have balanced rest and grasp classes?

We apologize for this confusion. We have included the following explanation in a new section called Equal class sizes for training:

“Since 800 ms of data per-trial were labeled as grasp (see Label assignment), while the remainder of the time in the trial ($t_{\text{remainder}} = t_{\text{min ISI}} + t_{\text{Go}} - 800 \text{ ms} \geq 2,300 \text{ ms}$) was labeled as rest, the rest class was overrepresented and therefore randomly downsampled such that the decoder would be trained on a balanced dataset of rest and attempted grasping sequences. Note that $t_{\text{remainder}}$ is at least 2,300 ms because the minimum ISI was 3 s, while the duration of the visual cue “Go” was 0.1 s.”

We have also clarified in the first sentence of the Model architecture paragraph that each 1 s sequence for the RNN was labeled according to only the sequence’s leading edge (modifications in blue).

“We designed an RNN ~~recurrent neural network~~ in a many-to-one configuration to learn the temporal dynamics ~~changes~~ in HG power over sequences of 1 s (Supplementary Fig. 7) with each sequence associated with only the label at the leading edge of the sequence.”

(9) Line 283: why a lock-out period of 1 second? Why not longer or shorter?

We chose a lock-out period of 1 second with the only constraint being that a click detected at the lowest possible voting threshold would not produce a second click from the same movement. Thus, we chose 1 second because none of the attempted grasps produced modulation lasting longer than 1 second (indeed modulation seemed to last only 700-800 ms). The participant did not report any discomfort with this lock-out period and therefore we did not perform a deeper analysis into a more optimal lock-out

period. However, we will investigate this parameter in our future work.

(10) Line 347: please include your reasoning for choosing specific voting thresholds. This information is provided in Supplementary Note 6. I would include this in the main paper.

Thank you for this suggestion. We did not include this in the main text of the manuscript because we did not take the systematic approach described in Supp Note 6 when training our original fixed model. After the participant informed us that he would prefer increased sensitivity (reducing the voting threshold down from 7 votes), we initially aimed to take the approach we described in Supplementary Note 6. However, we discovered a mistake in our code for online click detection in which alternative voting thresholds that were set to lower than 7 votes were automatically set to 4 votes. Fortunately, the participant preferred using the click detector with the 4-vote threshold compared to that with the 7-vote threshold. We corrected this mistake and repeated the analysis we originally intended with the new fixed detector, as described in Click detector retraining due to transient performance drop.

(11) Lines 413-420: this seems more Methods than Results. It feels like this is a repetition of information that I already read. Maybe only mention this in the Methods?

Thank you for this suggestion. We've integrated most of the sentences in this paragraph to appropriate Methods sections, copied below for the reviewer's convenience (modifications in blue):

"Training task

Training data was collected across four sessions (six training blocks in total) spanning ~~15~~16 days (Fig. 2a). We defined Day 0 as the last session of training data collection."

"Spelling application

...However, after several sessions of spelling and feedback from the participant, on Day +81 we reduced the voting threshold requirement to a 4-vote threshold (any 4/7 classifications within the ~~running~~ voting window needed to be grasp to initiate a click). This is because the participant reported that he preferred increased sensitivity despite a possible increase in false positive detections. We again enforced a lock-out period of 1 s."

"Online switch-scanning

Using the communication board, the participant was instructed to navigate to and select one of the keys verbally cued by the experimenter. If the participant selected the incorrect row, the cued key was changed to be in that row. Once a key was selected, the switch-scanning cycle would start anew (Supplementary ~~MovieVideo~~ 1, Fig. 3a, Supplementary Fig. 8). We recorded one session with the communication board on Day +21 after the completion of training data collection (Fig. 2b)...

...We recorded blocks with the switch-scanning spelling application across 17 sessions."

(12) Figure 2: please increase the font size on the x and y axes.

Thank you for this suggestion. We have increased the font size in Fig. 2 (top: before, bottom: after):

(13) Figure 4: please increase the font size of b, d, and f.

Thank you for this suggestion. We have increased the font size in Fig. 5 (formally Fig. 4) b, d, and f (top: before, bottom: after):

(14) Please include Supplementary figures 8 and 9 in the main text. These depict your main experimental task and belong in your main text.

We appreciate the reviewer’s suggestion and agree that the depictions of the main experimental task belong in the main text. We have added this figure (Fig. 3) after the Spelling application section in our revised manuscript. We have made all the necessary changes in referencing the figures which follow (i.e., Figs. 3,4 are now Figs. 4,5).

“Figure 3 | Switch-scanning applications. The participant was instructed to select an experimenter-cued graphical button (a) or to spell the sentence prompt (pale gray text) (b) by timing his clicks to the appropriate highlighted row or column during the switch-scanning cycle. For a detailed description of (a) and (b), refer to Supplementary Figs. 8 and 9, respectively.”

However, due to the lengthy but thorough descriptions for each application, we have decided to leave these figures as Supplementary Figs. 8 and 9 as well.

References:

Sutter, E. E. (1992). The brain response interface: communication through visually-induced electrical brain responses. *Journal of Microcomputer Applications*, 15(1), 31-45.

Verbaarschot, C., Tump, D., Lutu, A., Borhanazad, M., Thielen, J., van den Broek, P., ... & Desain, P. (2021). A visual brain-computer interface as communication aid for patients with amyotrophic lateral sclerosis. *Clinical Neurophysiology*, 132(10), 2404-2415.

Hughes, C. L., Flesher, S. N., Weiss, J. M., Downey, J. E., Collinger, J. L., & Gaunt, R. A. (2020). Neural stimulation and recording performance in human somatosensory cortex over 1500 days. medRxiv, 2020-01.

Reviewer #2 Continued: Responses to comments from manuscript PDF:

Lines 34-36 (Abstract/Results):

10-12 characters per minute using ECoG and c-VEP (Martinez-Cagigal et al., 2021). This method does not need any training data, or can even work with less than 1 minute of training data. A speller allows you to say anything you want. This click-decoder is limited to the available options. Does the sensorimotor cortex activity remain stable as ALS progresses? Visual cortex maybe less affected...

Thank you for pointing out this study to us. We have acknowledged this work in the Discussion (modifications in blue):

"... non-invasive BCIs using visually evoked potentials are estimated to produce comparable spelling rates⁵² and can potentially be trained with little or no neural data⁵³."

Please refer to our response to Major remark 5 for details. Further, ALS is a progressive disease primarily of motor neurons. Despite this, we are encouraged by recent clinical trials that demonstrated stable decoding of signals from motor cortex for years (Please refer to our response to Major remark 4 for details). However, it remains to be seen how the disease progression will affect signals from the sensorimotor cortex of our participant in the future.

Finally, though visual cortex may be less affected by disease progression, eye-movements can deteriorate over time. These eye-movements would be necessary for controlling any decoder based on visually evoked potentials.

Please refer to our response to Major remark 5 for details.

Line 36 (Abstract/Results):

But also tested on that same day?

Thank you for this question. We clarified in our revised manuscript that the new fixed model was tested six days after data collection. Please refer to our response to Minor remark 1 for details.

Line 47 (Plain Language Summary):

I would not call this a "spelling" application. It is more like a menu/button selection.

Thank you for this suggestion. However, the sole purpose of the application the participant used for spelling was primarily spelling. This spelling application could not be used for any other purpose.

Line 49 (Plain Language Summary):

It is a mouse-click task!

Yes, the output of our click detector is a click, akin to a mouse click by an able-bodied participant. We used the term “mouse-click” in the Plain Language Summary as we believe it can more clearly communicate the message of this paper to an audience not familiar with click-BCIs or BCIs in general. We chose this term over “brain-click” or “detected click”, etc.

Line 51 (Plain Language Summary):

add: recorded across several days prior to testing the algorithm.

Thank you for this suggestion. We have added this into our Plain Language Summary. Please refer to our response to Minor remark 2 for details.

Line 53 (Plain Language Summary):

Isn't it odd to decide to use a motor command for paralyzed people? Why not a selective brain response to tactile or visual stimuli?

Thank you for this question. We chose to use an attempted movement due to the success from previous clinical trials, in which attempted movements were also used to control a click. Please refer to our response to Major remark 4 for details.

Line 59 (Introduction):

Did the user use the application by him/herself for an extended period of time? Is this application what the user wants?

Thank you for these questions. The participant used the fixed click detector for three months without retraining or updating, albeit under supervision from the study team in the laboratory.

Click detection with switch scanning is one of several BCI control strategies that we are exploring in the CortiCom clinical trial. It represents the most basic and potentially one of the most stable and reliable means of communication. We chose to explore this particular strategy based on experience from previous BCI studies, but our participant did appreciate the potential utility of this approach.

Line 67 (Introduction):

Long-term?? ECoG electrodes are typically implanted less than 2 weeks! Whereas microelectrodes have been shown to be functional for over 7 years!! And what about Deep Brain stimulation devices? Those remain functional for years too!

Thank you for this suggestion. However, we found that long-term MEA signal quality metrics actually decreased in the paper referred to us by the reviewer. Additionally, we believe that comparison to DBS electrodes is out of scope of the current study as those electrodes serve primarily a fundamentally different purpose (stimulation) than the recording electrodes used for brain-computer interface. Please refer to the response to Minor remark 3 for details.

Line 68 (Introduction):

Define "chronic"

Thank you for this suggestion. We have defined "chronic" as greater than 30 days (> 30 days). Please refer to the response for Minor remark 4 for details.

Line 149 (Neural implant):

What kind of task was used to assess these locations?

Thank you for this suggestion. We have added the tasks that the participant was instructed to perform during the fMRI. Please refer to our response to Minor remark 5 for details.

Line 153 (Figure 1):

These images are a bit too small to see accurately

Thank you for the suggestion. We have increased the size of the left and right panels of subfigure h. Please refer to our response to Minor remark 6 for details.

Line 162 (Figure 1 Caption):

Why an RNN? Is that necessary? Seems like a linear decoder may also suffice

Thank you for this question. We appreciate the reviewer's concern that the choice of the decoding model should be appropriately chosen for the corresponding experimental task. We retrospectively agree with the reviewer that for the simple task of detecting clicks from attempted grasping movements, a simple linear classifier would have likely performed just as well. Please refer to our response to Major remark 3 for details.

Line 163 (Figure 1 Caption):

Why 7?

A voting length of 7 votes corresponds to 700 ms, roughly the time duration of the participant's movements. Though we labeled 800 ms worth of training data per-trial as movement (see Model architecture and training), we heuristically estimated that 8 votes generated too low a sensitivity.

Line 182 (Training task):

So actual movement is performed. Would it also work with imagined movement? Or at least no overt movement result?

Thank you for this question. We have not thoroughly investigated the potential of imagined movements for click detection. However, it may be worth investigating in the future alternative control strategies such as imagined movements. Please refer to our response to Major remark 4 for details.

Line 188 (Training task):

What were these bounds?

We apologize for the confusion. We have clarified in paragraph 2 of the Training task section in the revised manuscript that the reader may refer to Supplementary Table 1 for the ISI of each training block. Briefly, the bounds were either 3-6 s or 3.5-4.5 s post-visual cue. Please refer to our response to Minor remark 7 for details.

Line 190 (Training task):
were

Thank you. We have fixed this grammatical error in the revised manuscript.

Line 231 (Model architecture and training):
A "long" short-term memory? That seems confusing wording

We apologize for the confusion. An LSTM is an acronym for "long short-term memory," and is a commonly used type of RNN.

Line 232 (Model architecture and training):
Why these specific parameters?

We aimed to allow the participant to use the BCI as soon as possible, and so we used the neural network hyperparameters that we initially chose when testing the model offline using cross-validation. We did not optimize for these hyperparameters as we believe that such optimizations would not have significantly improved the model performance for the simple task of detecting clicks. Further, given that the model with this set of hyperparameters achieved high cross-validation accuracy offline and high performance in real-time, we continued using these hyperparameters throughout the rest of the experiment.

Line 236 (Model architecture and training):
What does randomly downsample mean? How is the rest class overrepresented if you have a balanced dataset of rest and attempted grasp sequences?

We apologize for the confusion. The rest class was overrepresented in terms of samples in the training data. This is because the shortest time between trials was 3 s ISI. Per-trial, there were also another 100 ms for the visual cue. Together, these make up 3,100 ms per-trial. Then, 800 ms of that per-trial time was labeled as grasp, which left at least 2,300 ms worth of samples being labeled as rest. We have added further clarification in a section called Equal class sizes for training. Please refer to our response to Minor remark 8 for details.

Line 282 (Medical communication board):
Why 10 votes?

Thank you for this question. We initially visually estimated that the duration of the participant's attempted movements was around 1 s. This translated into 10 votes, each representing 100 ms increments of data.

Line 284 (Medical communication board):
Why 1 second? Why not shorter or longer?

We chose a lock-out period of 1 second with the only constraint being that a click detected at the lowest possible voting threshold would not produce a second click from the same movement. Thus, we chose 1 second because none of the attempted grasps produced modulation lasting longer than 1 second (indeed modulation seemed to last only 700-800 ms). The participant did not report any discomfort with this lock-out period and therefore we did not perform a deeper analysis into a more optimal lock-out period. However, we will investigate this parameter in our future work.

Line 329 (Sensitivity and click rates):
So a false negative is when the amount of detected grasps in a row does not meet the 10 or 7 voting requirement?

Thank you for this question. More specifically, a false negative occurs when the voting threshold of the most recent set of votes (7 votes or 10 votes, for example) is not met within 1.5 s of the onset of attempted grasp. However, if this number is met after the 1.5 s window, it is considered a false positive.

Line 351 (Click latencies):
So 10 vote was 1sec? Why this long latency? Did it not work when it was shorter?

Thank you for this question. We initially chose a conservative value for our voting threshold such as to avoid false positive detections. Heuristically, we lowered this value in later experiments.

Line 418 (Long-term usage with a fixed click detector):
This is Methods, not Results. Some things are repeated, I don't think that is necessary here.

Thank you for this suggestion. We have integrated most of the sentences from this paragraph into the appropriate Methods sections. Please refer to our response to Minor comment 11 for details.

Line 421 (Long-term usage with a fixed click detector):
consecutive days?

Thank you for this suggestion. We have clarified this in the last sentence of Long-term usage with a fixed click detector (modifications in blue):

*"We found that the decoder performance remained robust **over a period of 111 days.**"*

Line 423 (Figure 2):

Please slightly increase the font size

Thank you for this suggestion. We have increased the font size. Please refer to our response to Minor comment 12 for details.

Line 462 (Switch-scanning performance):

It is pretty good indeed

Thank you.

Line 484 (Switch-scanning performance):

So this was new training data? Collected that day?

Thank you for this question. We have clarified in the Abstract and Methods that the data for the new fixed click detector was collected six days before use. Please refer to the response to Minor remark 1 for details.

Line 533 (Figure 4):

Please increase the font size on these b, d and f plots.

Thank you for this suggestion. We have increased the font size. Please refer to our response to Minor remark 13 for details.

Line 588 (Discussion):

I miss the opinion of the participant about the application in this paper.

We appreciate the reviewer's concern for the participant's experience. We have included metrics of the participant's experience with the spelling application administered via the NASA Task Load Index (NASA-TLX). Please refer to our response to Major remark 2 for details.

Line 600 (Discussion):

How does this BCI performance compare to other communication aids? So eyetracking or joystick only? A brain surgery is quite something to do! Here you only need a yes or no, so one output signal. This may be more easily achieved by EMG or eyetracking. Way less invasive.

Thank you for this question. We have added in the Discussion that " non-invasive BCIs using visually evoked potentials are estimated to produce comparable spelling rates⁵² and can potentially be trained with little or no neural data⁵³."

We would also like to emphasize that eye-tracking is often not a feasible solution for individuals with ALS due to the possible deterioration of their eye-movements. In addition, because the participant will

eventually lose motor capabilities, decoding EMG signals or using assistive devices, like joysticks, which require motor input may not be feasible for use.

Please refer to our response to Major remark 1 for details.

Reviewer #3 (Remarks to the Author):

This study demonstrates an ECoG-based single click decoder that has long-term stability for over three months, trained on very small amount of neural data. The authors address a fundamental requirement of stable decoder performance over long periods without recalibration for practical BCI applications, leveraging the well-known stability of ECoG signals. The manuscript is clearly presented. I have some comments, questions and suggestions regarding framing of the problem in the context of current click BCIs, exploration of some results and limitations of this approach.

Click BCI is not a new concept, as authors have discuss, however, I feel that the novelty of this particular work could be fleshed out more in comparison to the merits of ECoG as a stable neural signal. Authors point out in Introduction (line 62) that MEA BCIs can have sophisticated capabilities but use outside research environment is limited. However, this is true of all types of current BCIs. Also, long-term safety and efficacy of MEAs in humans have been demonstrated consistently for two decades, and is not just the characteristic of ECoG (line 65). New research in MEA BCIs show that frequent decoder recalibration might not be required with advance online unsupervised training methods that run in the background during online BCI use (e.g., CORP) so this might not be a limiting factor. Regarding the amount of training data required, it is mentioned that stent-electrode based BCI studies also required 1 or more sessions of data (line 77) which indicates that training on small amounts of data from single session has been shown before with ECoG-like signals. In view of these arguments, I think the introduction should be framed to justify strengths of this work. Single command BCIs are common with EEG and have shown decent performance in various studies using different paradigms, hence this should also be discussed.

Thank you for these critiques and suggestions. We have rewritten the Introduction to present a stronger case for ECoG-based click-detectors by highlighting their signal stability while also better addressing the shortcomings of previous ECoG-based click detectors.

We agree with the reviewer that use outside of research environments is limited for all BCIs and that the safety and efficacy of human MEA implants has also been extensively investigated. Therefore, we have removed our comments regarding MEA out-of-lab use and comments regarding better safety and efficacy pertaining to ECoG implants. Further, we also agree with the reviewer that CORP is a promising new methodology for unsupervised training of MEA-based decoders using corrected text outputs and we are excited to see how this research advances. We have included this in paragraph 1 of the Introduction.

“Nevertheless, there have been promising advances in online recalibration by correcting text outputs using language models¹⁶.”

Additionally, we agree with the reviewer that EEG-based BCIs are also effective for single-command detection and have been used in a variety of paradigms. We have also commented on this in paragraph 1 of the Introduction.

“On the other hand, EEG-based BCIs can be effective for single-command decoding, which has been used in a variety of paradigms¹⁷.”

The stability of the decoder shown in the results is remarkable. It could be useful to show a plot with the accuracy of each real-time session (computed offline) using the most updated decoder trained on all available prior sessions data to assess the maximum performance (upper bound on the accuracy) the decoder could have had in comparison to the fixed decoder currently used.

We thank the reviewer for appreciating the remarkably stable performance of the decoder we report in this manuscript. We have computed the sensitivity of each session, using a model trained on data from spelling blocks on all previous sessions. We have explained this procedure in a new Methods section called Model updates using previous spelling blocks, which we have copied below.

“Model updates using previous spelling blocks

To assess whether the spelling task itself could function as a modality by which to collect further training data, we trained “updated” classification models with data from spelling blocks of preceding sessions. We then simulated performance of these models on spelling blocks from the subsequent sessions. For example, the simulated performance of a click detector trained on data from all spelling blocks recorded up until and including day d was evaluated on all spelling blocks from day $d+1$. We used largely the same procedure to train each updated model as we did for the original fixed model, with only two differences. First, we determined the onset and offset times of the re-aligned trial averaged HG power traces relative to the start of attempted movement rather than a “Go” cue, which was not present during the spelling blocks. Second, since two attempted grasps could have occurred within a very short duration of each other (e.g., clicking into a row followed by clicking into the first column), we excluded from training all attempted grasps which occurred less than 3 s (the minimum jittered ISI, Supplementary Table 1) after a preceding attempted grasp. As described in *Sensitivity and click rates*, sensitivity using the original fixed detector was computed by determining the number of click detections (occurring as visual feedback to the participant) that occurred within 1.5 s after the onset of an attempted grasp. Since our offline analysis simulated algorithmic detection (and not on-screen clicks), we therefore shifted all click detections by 200 ms to account for the consistent delay between algorithmic detection and on-screen click mentioned above. TPF and FPF were computed as described above. We again only used data from online spelling sessions during which the click detector operated with the 4-vote voting threshold. The seed update model was trained on the third block of the online spelling session on Day +81 (Fig. 2b).”

We have also added the results of this simulation analysis in paragraph 2 of a new results section labeled Simulation performance and in a new Supplementary Fig. 12, both copied below.

“We compared the simulated performance metrics (sensitivity, TPF and FPF) of the original fixed click detector to simulated metrics from click detectors with updated models trained on data from all preceding spelling blocks (Supplementary Fig. 12). The simulated median detection sensitivity of these updated click detectors was 99.0%, which was higher than the 97.3% simulated sensitivity of the original fixed detector ($W = -3.098$, $P = 0.002$, two-sided Wilcoxon Rank-Sum test). Correspondingly, the simulated median TPF of the updated click detectors was 11.711 per min, slightly higher than the 11.574 per min simulated TPF of the original fixed detector ($W = 2.468$, $P = 0.014$, two-sided Wilcoxon Rank-Sum test). However, the simulated FPF of the updated click detectors was 0.641 per min (38.46 per h), higher

than the 0.157 per min (9.42 per h) simulated FPF of the original fixed detector ($W = 2.941$, $P = 0.003$, two-sided Wilcoxon Rank-Sum test)."

Supplementary Fig. 12 | Click detector performance using updated models. For all metrics, only spelling sessions in which the click detector operated with a 4-vote threshold are used. Dark and pale markers represent simulated metrics using the updated and original fixed click detectors, respectively. (a) Simulated sensitivity of click detection for each session. (b) Simulated true-positive and false-positive frequencies (TPF and FPF) measured as detections per minute.

Finally, we discuss these results in paragraph 4 of the Discussion, also copied below for the reviewer's convenience (relevant text in purple):

"The robust changes in HG modulation likely contributed to the high simulated performance of the click detectors that were trained on subsets of the original training data. Nonetheless, the results of our simulated model updates suggest that periodically updating fixed models with recent training data may enable higher sensitivity. The concurrent increase in FPF was likely due to training on false positive-inducing features that occurred independently of attempted grasping and were consequently labeled as rest. Though recent advances in unsupervised label correction have been primarily used for online retraining of speech models¹⁶, it may be possible to apply analogous methods to click detector outputs for relabeling such false positive-inducing features."

In figure 3, please also show the accuracy of the system along with sensitivity as well as the confusion matrix of classification (like fig 4) for three groups: (1) cross-validated results on training sessions, (2) 7-vote real-time sessions and (3) 4-vote real-time sessions for complete understanding of the decoder performance.

Thank you for your suggestions. We would first like to clarify that the confusion matrices shown in Fig. 5 (previously Fig. 4) represent cross-validated accuracy of models trained on the 9 held-out folds of the training data, as the reviewer suggests in (1). Specifically, all the original training data for the fixed model was split into 10 folds used for repeated cross-validation, resulting in the confusion matrix shown in Fig. 5b with an accuracy of 92.9% ("the mean accuracy from repeated 10-fold cross-validation (CV) was 92.9% (Fig. 4b)"). We have clarified this approach in paragraph 2 of Channel contributions and offline classification comparisons and have copied it below for the reviewer's convenience (modifications in purple):

To inform whether models trained with HG features from these smaller subsets of channels could retain robust click performance, we computed offline classification accuracies using 10-fold cross-validation (see Cross-validation) of the training data. We repeated cross-validation (see above) such that for each of the 10 validation folds a set of 20 accuracy values was produced. We then took the average of these 20 values to obtain a final accuracy for each fold. For each subset of channels, a confusion matrix and accuracy value were generated using the true and predicted labels across all validation folds and all repetitions. We compared these results to those generated by using features from all channels.

However, we have also added a paragraph in the Channel contributions and offline classification comparisons section describing the computation of the confusion matrix and accuracy from the 7-vote online sessions and the 4-vote online sessions, copied below.

“Finally, we computed the confusion matrix and classification accuracy value across all spelling blocks in which the click detector operated with a 7-vote threshold and with a 4-vote threshold using the original fixed model (trained on features from all channels). As described in Model updates using previous spelling blocks true grasp labels were assigned to each trial within the bounds of the onset and offset of the re-aligned trial averaged HG power traces relative to the attempted movement start. Again, data corresponding to attempted grasps which occurred less than 3 s after a preceding attempted grasp were excluded from labeling. All other samples were labeled as rest. An equal amount of rest and grasp samples were used for computing the confusion matrix and corresponding accuracy.”

We have also added these results to paragraph 2 in Switch-scanning performance and in Supplementary Fig. 10, copied below for the reviewer’s convenience (modifications in purple).

“Using the switch-scanning spelling application (from Day +46 to Day +111), the click detector achieved a median detection sensitivity of 94.9% using a 7-vote threshold, and a significantly increased sensitivity of 97.8% when using a 4-vote threshold ($W = -1.898$, $P = 0.057$, two-sided Wilcoxon Rank-Sum test; Fig. 4a). Offline classification accuracies across all spelling blocks where a 7-vote and 4-vote threshold were used was 90.8% and 93.6%, respectively (Supplementary Fig. 10).”

Supplementary Fig. 10 | Offline confusion matrices of spelling blocks. The confusion matrices on a sample-by-sample basis of all spelling blocks where the original fixed click detector operated with a 7-vote (a) and 4-vote (b) threshold.

Looking retrospectively, what is the minimum amount of data required to get the highest real-time accuracy? In other words, what would be the decoder accuracy as a function of the amount of training data.

Thank you for this suggestion. We performed simulations where we explored the click detector's performance using subsets of the original training data. We have described this in detail in the Performance as a function of training trials section in the Methods, copied below:

"Performance as a function of training trials

We investigated the relationship between the number of trials used for training the classification model and the resulting simulated performance of a click detector (sensitivity and FPF). This was done to determine whether similar performance to online spelling could have been achieved using a click detector model trained on fewer trials. To do this, we trained classification models with various numbers of training trials and tested them offline on data collected from online spelling sessions. Using the training procedure described above, we trained six models, each trained with an additional block's worth of data (Supplementary Table 1) compared to the preceding model. As such, six models were trained on data containing either 50, 100, 150, 300, 390, or 480 trials (3.77, 7.56, 11.34, 25.43, 34.68, and 43.92 min, respectively). Note that the click detector model used for online spelling was trained on the same 480 trials, the entirety of the original training dataset. The models were tested on data from each online spelling block during which the click detector operated with a 4-vote voting threshold. Models were not tested on spelling blocks in which a 7-vote threshold was used because for a majority of these sessions (Days 46-56), there was no audio-synchronization cue to align the neural data recorded by the Neuroport system (and the resulting click-detections from simulation analysis) with the recorded video frames. As such, it was not possible to accurately determine when the simulated click detections would have occurred relative to the onset of attempted grasp. Sensitivity and FPF were computed as described in Sensitivity and click rates. Then, for each specific number of training trials, we computed the across-session median sensitivity and FPF."

We have also added the results of these simulations in the paragraph 1 of the Simulation performance section and supplementary Fig. 11, both copied below:

"The six click detectors trained on 50, 100, 150, 300, 390, and 480 trials achieved simulated median sensitivities of 84.3%, 91.4%, 93.5%, 87.16%, 95.8%, and 95.8% and FPFs of 0.154, 0.221, 0.321, 0.177, 0.195, and 0.096 per min respectively, (Supplementary Fig. 11). The simulated median sensitivity and FPF of any of the click detectors trained on 50, 100, 150, 300, or 390 trials were not significantly different from the simulated median sensitivity and FPF of the click detector trained on all the original training data (480 trials) (for all comparisons $P > 0.05$, two-sided Wilcoxon Rank-Sum test)."

Supplementary Fig. 11 | Click detector performance as a function of training data. Simulated sensitivity (a) and FPF (b) are shown in relation to the number of trials used for training the corresponding click detector. For a specific number of training trials, the simulated sensitivity and FPF of the click detector are shown across all sessions in which the click detector operated with a 4-vote threshold (pale markers). The simulated median sensitivity and FPF values are shown as bold markers.

Finally, we have acknowledged in paragraph 4 of the Discussion that it may be possible to train a model with even fewer trials due to the robust changes in HG modulation, while simultaneously maintaining a high performance (relevant text in purple):

“The robust changes in HG modulation likely contributed to the high simulated performance of the click detectors that were trained on subsets of the original training data. Nonetheless, the results of our simulated model updates suggest that periodically updating fixed models with recent training data may enable higher sensitivity. The concurrent increase in FPF was likely due to training on false positive-inducing features that occurred independently of attempted grasping and were consequently labeled as rest. Though recent advances in unsupervised label correction have been primarily used for online retraining of speech models¹⁶, it may be possible to apply analogous methods to click detector outputs for relabeling such false positive-inducing features.”

It’s not clear to me whether the correct characters per min and correct words per min reported in fig 3 are calculated for selecting single characters (number of actual clicks) or on the output of autocomplete, which would require fewer clicks but inflate the CCPM/CWPM.

Thank you for raising this concern. We have modified the last sentence of Spelling rates to clarify that all analyses of spelling performance were based on spelling done with autocompletion options (modifications in purple):

“Note that all spelling was performed with assistance of autocompletion options from the language model and subsequently all analyses of spelling performance were based on this assisted spelling.”

Supplementary fig 2 shows very strong event-related desynchronization (ERD) in lower frequencies around beta band uniformly in majority of channels. Could this be used as more robust feature for click detection to supplement high gamma power? The robustness of ERD in EEG literature for movement

detection is well known, which can also be consistently observed here. Could that be leveraged for better click classification and improved stability? It would be interesting to see if ERD also reduced after day 118 when HG was reduced. Why were the features restricted to HG?

Thank you for this suggestion. We did not include features from lower frequency bands due to event-related synchronization (ERS), which occurred immediately after ERD. Additionally, this modulation occurred on a longer timescale than the HG activity which would have made labeling the data on a sample-by-sample basis difficult. We have included this explanation in the Feature extraction section and have copied it below for the reviewer's convenience (modifications in purple):

“...We summed the spectral power in the frequency band between 110 and 170 Hz to compute ~~on~~the high-gamma (HG) power. We chose this frequency band due to the rapid timescale of HG modulation during attempted grasping and chose to exclude features from lower frequency bands due to event-related synchronization (ERS) occurring immediately following event-related desynchronization (ERD) (Supplementary Fig. 2). This pattern of low frequency ERD quickly followed by ERS occurred on a longer timescale than the HG activity and would have made it difficult to clearly define the onset and offset times of the trial-averaged neural activity for assigning rest and grasp labels for model training (see Label Assignment). We chose this lower bound of the HG frequency band because post-movement low frequency activity sometimes extended to 100 Hz in several channels (Supplementary Fig. 2). For each 100 ms increment, this resulted in a 128-channel feature vector that was used in subsequent model training.”

We also investigated whether low-frequency ERD (10-30 Hz) displayed a concurrent deviation in movement-aligned power as seen for HG features in Supplementary Fig. 15a (formally Supplementary Fig. 12a). We now report these results in paragraph 1 of Click detector retraining due to transient performance drop and refer to Supplementary Fig. 16. We have copied this text and figure below for the reviewer's convenience (relevant text is green in revised manuscript):

“Conversely, we found an increase in the magnitude of low frequency power (10-30 Hz) (Supplementary Fig. 16).”

Supplementary Fig. 16 below:

“Supplementary Fig. 16 | Increase in magnitude of movement-aligned low frequency power. (a) For each channel, one dot represents the minimum value of the median low frequency power trace from the first 30 movement-aligned trials. The horizontal axis represents spelling sessions. The two dashed vertical lines for each channel split these values into three regions: left) values prior to performance drop while using the original fixed click detector; middle) values during the performance drop while using the original fixed click detector; right) values after retraining a new fixed click detector. The largest increase in the magnitude of minimum low frequency values (representing an increase in event-related desynchronization) was observed in channel 108, the second-most salient channel of the original click detection model (outlined in black). The green, pink, and blue dots are minimum low frequency power values from the last session prior to performance drop, the fifth session with the observed decreased performance, and the first session using the newly trained click detector, respectively. The central sulcus is delineated by a thick black line (CS) and widens at the top such that channels 111, 119, and 127 are over it. The pre-central sulcus is delineated by a thick green line (Pre-CS). (b)-(d) show the trial-averaged spectral power from channel 108, with the low frequency band (10-30 Hz) highlighted only in the region of event-related desynchronization. Trial-averaged spectrograms of the first 30 movement-aligned trials from the last spelling session prior to performance drop (b), the first spelling session with the observed decreased performance (c), and the first spelling session after training a new click detector (d).”

We have now acknowledged the potential benefit of low frequency ERD to the robustness of a click detector in paragraph 7 of the Discussion:

“...we did not explore the utility of low frequency power suppression, which can be extremely stable^{60,61} and may improve the robustness of a click detector.”

Again, in supplementary figure 2, some ERD can be observed before go cue, is this due to anticipatory behaviour?

Thank you for this observation. This pre-cue low-frequency activation is likely due to anticipatory activity. Though we jittered our interstimulus interval (ISI) between 3.5 and 4.5 s, this may have not been a wide enough time range, and the participant may have been able to predict with a limited accuracy when the next cue would occur. We have added the following sentence in the caption of Supplementary Figure 2:

“Additionally, low frequency activation is noticeable in some channels before cue onset, which was likely due to anticipatory activity.”

Why was RNN chosen for binary classification (instead of simply flattening the sequential features which can be sufficient for simple binary classification)? Did you try a simpler classifier e.g., logistic regression, SVM or LDA? These can be faster to train and require less data.

Thank you for this question. We would like to emphasize that that the primary objective of this study was to demonstrate that a decoder trained on a small amount of data would maintain high performance for an extended BCI use period. While the choice of the decoder is an important factor for achieving this goal, we believe that it is primarily the long-term stability of the modulation in ECoG signals which makes this possible. In hindsight, we agree the reviewer's suggestion that a straightforward linear classifier, such as an SVM, might have delivered comparable performance. However, our motivations for training an LSTM were two-fold:

1) This study is part of a larger clinical trial where we aim to expand classification of attempted upper-limb movements beyond click-detections. As such, we created a model training pipeline that would allow us to train more complex models in the future, particularly for tasks where the temporal aspect of the neural signals could significantly influence a decoder's performance.

2) We aimed for our participant to quickly benefit from a high-performing BCI. In order to achieve this, we hypothesized that a non-linear classifier would outperform a linear model. This is because the non-linear classifier has the advantage of recognizing temporal patterns in neural activity, thereby potentially enhancing overall performance.

We have added these reasons as the first paragraph of Model architecture and training, which we have copied below for the reviewer's convenience:

“We used a recurrent neural network (RNN) for classifying rest vs. grasp. As this study is part of a larger clinical trial, we aimed to build the model training pipeline such that in the future we could train more complex models for tasks in which the temporal domain would significantly contribute to decoder performance. Additionally, we aimed to allow the participant to use a high-performing BCI as soon as possible, and to this end we anticipated

that a non-linear classifier would achieve higher performance than a linear model due to the advantage of recognizing temporal patterns in neural activity.”

It would be helpful to readers to put your CCPM and CWPM in context of other similar BCI studies in the discussion. It will be useful to also compare the performance with similar EEG based systems. What advantage would this BCI system provide over a binary classification EEG BCI?

We thank the reviewer for this question and have acknowledged that EEG-based BCIs have achieved comparable performance. We have echoed this point in paragraph 5 of the Discussion, copied below for the reviewer’s convenience:

“...non-invasive BCIs using visually evoked potentials are estimated to produce comparable spelling rates⁵² and can potentially be trained with little or no neural data⁵³”

However, we would like to emphasize that implantable BCIs have several important advantages over non-invasive recording modalities. Firstly, they eliminate the need for daily application of external sensors and potentially require minimal caregiver involvement for independent home use [1 (ref. 1 in revised manuscript)]. Secondly, ECoG-based BCIs ensure stable functionality for a duration for up to at least 3 years [2 (ref. 18 in revised manuscript)] (7 years in Vansteensel et al., under review). Lastly, implanted electrodes are potentially always accessible to users, providing BCI functionality round-the-clock, allowing users to use the system at their convenience, including both day and night. The referenced unpublished papers are available upon the reviewer’s request.

*[1] Vansteensel, M. J. et al. Fully Implanted Brain–Computer Interface in a Locked-In Patient with ALS. N Engl J Med **375**, 2060–2066 (2016).*

*[2] Pels, E. G. M. et al. Stability of a chronic implanted brain-computer interface in late-stage amyotrophic lateral sclerosis. Clinical Neurophysiology **130**, 1798–1803 (2019).*

How does it compare with P300 speller in terms of speed and accuracy?

Thank you for this question. We have now commented on the performance of P300 spellers for participants with ALS in paragraph 5 of the Discussion (relevant modifications in purple):

“BCIs based on P300 spellers have achieved a wide range of accuracies (65-100%) and have achieved spelling rates of 1.2 - 6 CCPM for people with ALS⁵⁴⁻⁵⁷, while typing speeds of 17 characters per min using eye-tracking alone have been reported⁵⁸. However, control strategies based on eye movements may cause eyestrain during long periods of use²⁷⁻²⁹ and worsen as residual eye movements can deteriorate in late-stage ALS³⁰⁻³³”

Also, how does this BCI performance compare with other technologies/aids that you participant uses for communication in his daily life?

Thank you for this question. Comparison to other technologies/aids that the participant uses is difficult to assess because we do not have any subjective metrics (such as NASA-TLX) from the participant of any other device that he uses. Any other assistive devices that the participant uses are used at his home, and we do not formally log of how often he uses them. These two factors make it difficult to compare the use of our BCI with the participant's other assistive devices. However, we will consider formally logging his at-home use with other assistive devices. In addition, we now report how we assessed the participant's cognitive workload in the Methods (copied below):

“Cognitive workload

In order to evaluate the participant's experience using the switch-scanning spelling application, we asked the participant to complete the NASA task load index (NASA-TLX) questionnaire^{43,44} using the NASA-TLX iOS application, a commonly used set of questions to evaluate a participant's mental, physical, and temporal demand of a task as well as the perceived performance, effort, and frustration of a task. These categories are each scored from 0-100 where lower and higher scores correspond, respectively to less and more of each of the six above-mentioned characteristics.”

And in the Results:

“We additionally evaluated the participant's subjective cognitive workload of switch-scanning spelling with the click detector using the NASA-TLX iOS application. Across the six sessions using the retrained click detector the participant reported scores of 7.5 ± 2.7 (mean \pm standard deviation) for mental demand, 8.3 ± 2.6 for physical demand, 5.8 ± 3.7 for temporal demand, 6.7 ± 5.2 for performance, 6.7 ± 5.2 for effort, and 6.7 ± 2.6 for frustration. These low scores indicate that the participant did not have difficulty in controlling the switch-scanning spelling application via click-detection.”

While single click decoding can be useful and robust, this is still not practical and very slow for use with the speed of ~ 2.2 words/min speed. Please discuss limitations of this approach. It seems to me that the bottleneck here is the switch scanning application used which traverses each row and column and pauses for 1s at each switch. Would a faster switch scanning application design (something similar to Dasher) improve the speed? Please discuss how can overall speed of the whole BCI system be improved?

Thank you for this question and we agree with the reviewer that the main limitation is speed. As the reviewer mentions, 2.2 words/min is slow. There are several ways in which we can improve the word typing speed.

First, the preparatory rows and columns can be eliminated (see description of these features below, copied from Supplementary Fig. 9 caption).

“The switch scanning process started by sequentially highlighting the three red pre-selection markers on either side of the sentence, where each highlight lasted 1 s. This was to allow the participant a brief preparation period in case he wanted to select row 1.”

“Once a row was selected, the gray pre-selection column on the left was highlighted in yellow for 1 s to allow the participant a brief preparation period in case he wanted to select column 1.”

We have added expanded on this in Supplementary Note 8 (copied below):

“We included pre-selection rows and columns to allow the participant a brief preparation period before clicking into the first row or first column, respectively (Supplementary Fig. 9). However, the total time the switch-scanner spent in these preparatory regions was 4 s per scanning cycle, during which the application could not receive clicks. For a future study participant, we may slowly start shortening these preparation periods and possibly eliminate them completely as the participant becomes more comfortable with timing his or her attempted grasps to highlighted rows or columns. Considering that we designed the application such that the language model’s most probable words and letters appear in the top two “clickable” rows, reducing the total scan time by 4 s per desired button may significantly increase the spelling rate. It could similarly be beneficial to rearrange the keyboard such that the most used letters take the fewest scans to reach (i.e., placement in the top left corner of the keyboard).”

Secondly, we expect that the sentence samples from the Harvard sentence corpus, which the participant was asked to complete were not representative of the statistics on which the language model was trained. The Harvard sentences corpus is meant to contain a phoneme distribution which is representative of those used in the English language. However, due to achieving this phoneme distribution with a limited number of sentence samples, the resulting sentences themselves are not actually representative of language used in real-life communication. We expect that the language model could be substantially more effective during free-form spelling, and ideally tuned to the participant’s lexicon and linguistic patterns. We have commented on this point in the paragraph 7 of the modified Discussion:

“Spelling rates were likely hindered by preparatory periods (Supplementary Note 8), and by the divergent linguistic statistics between the Harvard sentence prompts and the language model used for letter and word-autocompletion. We expect that spelling rates would improve with free-form spelling and a language model fine-tuned to the BCI user’s preferences.”

What computational overheads cause latency of 200ms for producing the click after its detection and can this be reduced?

Thank you for this question. This latency was caused by the time it took to transmit and present the detection on the front-end of the application. Cumulatively, this was likely a combination of the BCI2000 streaming data packet size (100 ms), the delay in the front-end application for state change registration and the delay in animation rendering. We believe this latency can be reduced by 1) lowering the streaming packet size, which will depend on the speed of our per-packet processing, and 2) by reducing our messaging time between our front-end application and BCI2000.

And finally, would this decoder with HG features work robustly in cases where there is no residual arm/hand movement in a participant to perform actual hand movement for initiating clicks?

Thank you for this question. We expect that modulation of signals in the upper-limb motor cortex from such attempted movements would still be observed and usable for BCI [1,2 (refs. 1,18 in revised manuscript)], as we acknowledge in paragraph 3 of the Introduction:

“...the participant used these brain clicks to communicate in her daily life for more than 3 years¹⁸...”

Note that even though Pels et al., 2021 [2 (ref. 18 in revised manuscript)] observed a steady decrease in high frequency band power, this decrease happened over a time scale of roughly three years and modulation of these signals remained usable for BCI control during that time period.

We plan to continuously monitor high frequency (110-170 Hz) signal modulation during attempted grasps as our participant’s condition progresses. Additionally, we will similarly monitor the stability of lower frequency bands. If needed, it may also be possible to train our decoder on features from a more stable frequency band if the modulation of the current frequency band becomes unusable for click detection.

[1] Vansteensel, M. J. et al. Fully Implanted Brain–Computer Interface in a Locked-In Patient with ALS. *N Engl J Med* **375**, 2060–2066 (2016).

[2] Pels, E. G. M. et al. Stability of a chronic implanted brain-computer interface in late-stage amyotrophic lateral sclerosis. *Clinical Neurophysiology* **130**, 1798–1803 (2019).

Minor: Which channel activity is shown in supplementary fig 12 b,c,d

We apologize for this confusion. We have clarified in the caption the channel which is shown:

“(b)-(d) show the trial-averaged spectral power from channel 112, with the HG frequency band (110-170 Hz) highlighted.”

Reviewer #4 (Remarks to the Author):

Summary

In this well written and interesting manuscript, the authors describe a brain-computer interface (BCI) that can translate cortical signals of a person with (incomplete) hand paralysis as they attempt to grasp their right hand into clicks, which are subsequently used to drive a switch-scan spelling interface to communicate in full sentences.

Importantly, they also show that the interface can achieve stable performance for months without recalibration.

The methods appear sound; however, some comments should be addressed during revision:

Comments

The sudden performance drop is perplexing. Do saliencies appear consistent before / after the performance drop?

Thank you for this question. We have computed the saliency map for attempted grasps using features from spelling blocks during the drop in performance. We describe the process of computing this saliency map in Supplementary Note 7 (relevant text is orange in revised Supplementary Information):

“We computed the importance of HG features from each channel when generating grasp during the time period when we observed a drop in click detector performance. We used largely the same procedure to label each sample as we describe in Feature extraction and label assignment with only two differences. First, we determined the onset and offset time of the re-aligned trial averaged HG power traces relative to the start of attempted movement rather than to a “Go” cue, which was not present during the spelling blocks. Second, since two attempted grasps could have occurred within a very short duration of each other (e.g., clicking into a row followed by clicking into the first column), we excluded from training all attempted grasps which occurred less than 3 s (the minimum jittered ISI, Supplementary Table 1) after a preceding attempted grasp. We then computed the integrated gradients (as described in Channel contributions and offline classification comparisons) from the original fixed click detector with respect to the input features from each sample labeled as grasp. This generated an attribution map for each sample⁷. The final saliency map was computed by averaging the attribution maps across all samples and normalizing the resulting mean values between 0 and 1.”

We have also added a paragraph to the Channel contributions to grasp classification section in the main text and copied it below for the reviewer’s convenience (relevant text is orange in revised manuscript):

“Finally, we assessed which channels produced the most important HG features for attempted grasping during period with the performance drop (Supplementary Note 7). Despite lower performance (Supplementary Fig. 14), relative channel contributions remained largely the same (Supplementary Fig. 21) with the saliency value from channel 112 being 30% and 42% larger than the next two most salient channels (channels 108 and 118), respectively. Indeed, despite the drop in the movement-aligned HG responses across many

channels, these relative saliency values suggest that some structure of channel importance was conserved.”

Finally, we have added Supplementary Fig. 21, which displays the saliency map:

“Supplementary Fig. 21 | Normalized saliency values during performance drop. Saliency maps for assessing which channels produced the most important HG features for attempted grasping during the period with the performance drop. Channels overlaid with larger and more opaque circles represent greater importance of that channel’s HG features during attempted grasp. The channels with the three highest saliency values are marked.”

Did the authors try other types of BCI control, such as cursor control or having, for example, 3 different hand target? Given the performance, it seems likely that 128 channels of ECoG could facilitate control over a richer output space to get faster communication rates. Some explanation for why the authors settled on a single type of output could be helpful.

Thank you for this question, and we agree with the reviewer that with the dense ECoG coverage over upper-limb cortex control strategies such as cursor control should definitely be investigated. Indeed, previous BCI studies have demonstrated multi-command and cursor movements are control strategies which can both be leveraged by the BCI user. Moreover, the current study was conducted as part of a clinical trial in which multiple different strategies for BCI communication, including speech BCI, are being explored. Here, we aimed to provide the participant with high-performance BCI functionality as soon as possible, given the progressive nature of his ALS. We also anticipated that switch-scanning, due to its simplicity, would serve as a reliable method of communication in case more sophisticated BCI communication strategies were less performant at any given time. Thus, we thought it was important to show that single-command decoding for communication can be achieved quickly and could be reliably maintained over time.

In Figure 1, can the authors add some visual indication that the patient is attempting to grasp their hand? It would help convey the system use a little more clearly.

Thank you for this suggestion. We would like to point out that we Supplementary Movies 1 and 2 were included, which both display the participant performing attempted grasps to generate clicks using two applications. However, we have added images of the participant's hand to Fig. 1g to indicate when the participant was attempting to grasp vs. when the participant was relaxing his hand (top: before, bottom: after).

We have also added a sentence to the Fig. 1g caption to describe this:

“Transparent images of the hand are shown to indicate the attempted grasp before a click and a relaxed configuration otherwise.”

Can the authors comment on why they chose 110-170 Hz as high-gamma range? It seems that the range that groups use varies in the literature.

Thank you for this question. Yes, we have commented on this in the paragraph 1 of Feature extraction and label assignment:

“We chose this lower bound of the HG frequency band because post-movement low frequency activity sometimes extended upward of 100 Hz in several channels (Supplementary Fig. 2).”

Briefly, we wanted to ensure that our frequency range for high-gamma was not contaminated by post-movement lower-frequency rebound.

The Discussion spends a lot of time simply restating content from the Results. It is preferred for Discussions to instead frame and interpret the results, not simply repeat them. The Discussion could be heavily trimmed and condensed.

Thank you for this suggestion. We have substantially trimmed the Discussion without neglecting issues that we thought were important for contextualizing our results relative to previous studies or highlighting their strengths and limitations, including room for future improvements. We have attempted to eliminate duplications between the Results and Discussion sections in several places (please refer to orange text in the Discussion of the revised manuscript).

Reviewers' comments:

Reviewer #1 (Remarks to the Author):

Thanks to the authors for addressing all my questions. They have been answered to my satisfaction. I have two points of clarification remaining, though these are considered minor.

Original comment

2) The manuscript contains some blocks of text which this reviewer finds unnecessary for the understanding of the manuscript, or which better belong in other locations.

a) Most of the last paragraph of the introduction belongs in the results and methods.

Response:

Thank you for this suggestion. We have rewritten the Introduction, including the last paragraph, that we have copied below for the reviewer's convenience (relevant text is green in the revised manuscript):

“The previous studies described above showed that click detectors can be used with a variety of BCI applications and can contribute significantly to a user's repertoire of communication modalities. Despite these promising results, the potential performance limits of such click detectors have remained relatively underexplored. In particular, chronic high-performance use without model retraining is a critical factor for enabling independent home-use, as BCI users should have round-the-clock access to a functioning click detector that requires

minimal caregiver involvement. By leveraging the stability of ECoG signals, we were able to train a model on a limited dataset and test it for a period of three months without retraining or daily model adaptation. Specifically, we demonstrated an improved click detector with a substantially increased spelling rate using a switch-scanning paradigm.”

Change is acceptable except the last sentence should clarify what the improvement was (accuracy and speed?) and what it was compared to. Previous ECoG switch-scanning BCIs? References?

Original Comment

[A19]: To results/discussion

Response

Thank you for this suggestion. However, per the “Style and formatting checklist” required by the journal to which we are submitting, we are required to have the Statistics and Reproducibility section as part of the Methods.

I agree that the methods for determining reproducibility should be placed in the methods, as you have done with the Statistical analysis paragraph. The reproducibility paragraph, however, should describe the method used to determine reproducibility, (performance across sessions), not the results of experiment reproducibility.

Reviewer #2 (Remarks to the Author):

The authors have adequately addressed my concerns and made the desired changes. I thank the authors for their clear and concise responses and I have no further objections to the publication of this paper.

Reviewer #3 (Remarks to the Author):

I would like to thank authors for answering my queries and their hard work for revising the manuscript based on some suggestions. I am satisfied with the revision. I have a few minor follow-up questions:

Line 69 needs a reference: “However, the utility of ECoG for chronically (> 30 days) implanted BCIs has only been tested in a few participants.”

Studies mentioned in lines 73, 75 etc. require references. It is not clear how many different studies are discussed in these couple of sentences.

In the last sentence of the introduction (line 103), authors state that this study demonstrates an improved click detector with increased spelling rate. In the previous two paragraphs in the introduction, several different metrics of spelling/click performance of various studies have been reported. It would be very useful if you can bring your results together and compare them with these previous studies directly (you can choose one or more metric/s such as spelling rate, click detection latency, detection accuracy) in one place in the discussion section to support the statement in line 103. Currently in the discussion, the results are only compared with Vansteensel 2016, and it is somewhat difficult to compare your results with the different metrics reported for the studies in the introduction.

In line 64, I am not sure what methodology for recalibration has been used in reference 16, but I don't think it is simply correcting text outputs using LM for recalibration, rather I think it is recalibrating the decoder by retraining it continuously in the background after every trial to keep its accuracy steady over longer timeframes. Please check this in the cited reference and rephrase the statement accordingly.

In reference to my comment about other assistive devices used by the participant, if the participant is still enrolled in the trial, I would suggest that the authors ask the participant for his informal subjective view on how he finds using the spelling/click BCI and how it compares to the methods he uses for communication in his daily life. Please also ask the participant what kind of assistive devices he currently uses and report this in the manuscript even if you don't have access to formal comparison. It is quite important for the field to understand BCI users' perspectives on the BCIs they are testing (especially for the implanted BCIs) and other assistive technologies available to them.

Line 263: Thanks for pointing out that HG features were chosen as their modulation happens on a faster timescale as compared to ERD/S. However, I am not sure why the occurrence of ERS post-ERD would be a problem. I am not convinced with the presence of ERS as the reason for excluding low-frequency activity. In theory, one could still only use ERD occurring prior to the movement onset to detect click without introducing latencies and completely disregard ERS arising afterwards. Also looking at supplementary figure 2, it appears that HG and peak ERD (dark blue) occur for the same duration (timescale).

Thanks for clarifying that the character/word rates were computed after autocomplete assist. The true positive frequency of this decoder is 10 per min with 7-vote but the correct character rate is only 9.1 (that too after autocomplete assist) and similarly for 4-vote strategy, TPF (11) is higher than

CCPM (10.2). Is this because the participant clicked on wrong characters more often? If so, what might be the reason for this (e.g., layout/UX of the switch scanning app)? If available, please report wrong character selection rate per minute (WCPM) so the CCPM and speller performance can be understood in this context in results paragraph - line 604.

With the latency of 0.6s per click + 1s cool off period (total latency 1.6s per click), I think the maximum capacity of this BCI can be 37 clicks per min. It would be good to mention this and discuss why the TPF is only 11 (despite very low FPF), which could be because of the wait times for the speller to slowly iterate through rows/cols to reach the desired character before it could be selected via BCI click. Hence, in theory, with faster UX, the character selection frequency can increase.

In the figures, statistically significant difference between different elements could be using * notation or mention this in the caption of the figure for clarity if applicable.

Line 642 “Conversely, we found an increase in the magnitude of low frequency power”, should this be ... conversely, we found increased desynchronization in low frequencies during this transient period? Looking at supplementary figure 16, it appears that ERD has increased during this transient period.

Reviewer #4 (Remarks to the Author):

the authors have done a great job to address my comments. Eddie Chang

Reviewer #1 (Remarks to the Author):

Thanks to the authors for addressing all my questions. They have been answered to my satisfaction. I have two points of clarification remaining, though these are considered minor.

Original comment

- 2) The manuscript contains some blocks of text which this reviewer finds unnecessary for the understanding of the manuscript, or which better belong in other locations.
a) Most of the last paragraph of the introduction belongs in the results and methods.

Response:

Thank you for this suggestion. We have rewritten the Introduction, including the last paragraph, that we have copied below for the reviewer's convenience (relevant text is green in the revised manuscript):
"The previous studies described above showed that click detectors can be used with a variety of BCI applications and can contribute significantly to a user's repertoire of communication modalities. Despite these promising results, the potential performance limits of such click detectors have remained relatively underexplored. In particular, chronic high-performance use without model retraining is a critical factor for enabling independent home-use, as BCI users should have round-the-clock access to a functioning click detector that requires minimal caregiver involvement. By leveraging the stability of ECoG signals, we were able to train a model on a limited dataset and test it for a period of three months without retraining or daily model adaptation. Specifically, we demonstrated an improved click detector with a substantially increased spelling rate using a switch scanning paradigm."

Change is acceptable except the last sentence should clarify what the improvement was (accuracy and speed?) and what it was compared to. Previous ECoG switch scanning BCIs? References?

Thank you for this suggestion. We have clarified in the last sentence of the Introduction that the improvements were primarily compared to prior switch scanning work by Vansteensel et al., (2016). We have copied this sentence for the reviewer's convenience (modified text is green in revised manuscript):

"Specifically, we demonstrated a switch scanning BCI with a substantially improved spelling rate compared to prior switch scanning BCI work¹."

In this sentence we mentioned that the primary improvement was the increased spelling rate, which we now directly compare in paragraph 5 of the Discussion. However, we refrained from comparing more specific metrics such as accuracy and speed in the Introduction because these comparisons are either not possible or require more nuance and explanation more appropriate for the Discussion. For example, Vansteensel et al., (2016) were not able to measure latency from movement attempts to click detection due to the locked-in state of their participant. Because their decoder required five consecutive 200 ms epochs with neural features exceeding a pre-determined threshold, their latency could not be shorter than 1 s (revised text is green below):

"Our results improve upon the previous switch scanning performance reported by Vansteensel et al., (2016)¹. In that study in which a participant with ALS ~~that~~ was implanted with four contacts over hand motor cortex and achieved a spelling rate of 1.8 CPM-letters per minute ~~and~~ with a latency of at least 1 s per click (compared to a spelling rate of 10.2 CPM and 0.68 s latency reported in the present work). The authors were not able to measure the latency from movement attempts to click detection due to the locked-in state of their

participant. Because their click detector required five consecutive 200 ms epochs with neural features exceeding a pre-determined threshold, their latency could not have been shorter than 1 s.”

Due to the locked in state of their participant, the authors were also not able to directly measure the sensitivity of click detection as a percentage of attempted movements. Instead, they estimated the overall accuracy (87-91%) from the number of correct clicks and the number of times clicks were correctly withheld on a per-trial basis during switch scanning (where one trial pertains to one individual row or column scan). In contrast, because our participant had residual movement that we could observe, we were able to directly measure the sensitivity, as well as the false positive frequency of the click detector, eliminating any effects from inattention or inadvertent movement attempts.

Finally, in the same paragraph (paragraph 5 of the Discussion), we have now made a more detailed comparison of our spelling rates to those reported by Oxley et al., (2020) and of our latency measurements to those reported by the same group in Mitchell et al., (2023):

“Although spelling rates were slightly lower than those observed in Oxley et al., (2020)¹⁹ (14-18 CCPM) the participants in that study used eye-tracking (ET) to first navigate to the appropriate letter before selecting it with a brain click. Additionally, though the same group reported accuracies of 97.4% and ~82% in selecting one of five targets (without ET), these corresponded to relatively longer latencies of 2.5 s and 0.9 s, respectively³.”

References:

1. Vansteensel, M. J. et al. Fully Implanted Brain–Computer Interface in a Locked-In Patient with ALS. *N Engl J Med* **375**, 2060–2066 (2016).

3. Mitchell, P. et al. Assessment of Safety of a Fully Implanted Endovascular Brain–Computer Interface for Severe Paralysis in 4 Patients: The Stentrode With Thought-Controlled Digital Switch (SWITCH) Study. *JAMA Neurol* **80**, 270 (2023).

19. Oxley, T. J. et al. Motor neuroprosthesis implanted with neurointerventional surgery improves capacity for activities of daily living tasks in severe paralysis: first in-human experience. *J NeuroIntervent Surg* neurintsurg-2020-016862 (2020) doi:10.1136/neurintsurg-2020-016862.

Original Comment

[A19]: To results/discussion

Response

Thank you for this suggestion. However, per the “Style and formatting checklist” required by the journal to which we are submitting, we are required to have the Statistics and Reproducibility section as part of the Methods.

I agree that the methods for determining reproducibility should be placed in the methods, as you have done with the Statistical analysis paragraph. The reproducibility paragraph, however, should describe the method used to determine reproducibility, (performance across sessions), not the results of experiment reproducibility.

Thank you for your suggestion. We have now tried to follow the “Style and formatting checklist” more precisely. We have copied the instructions for the Statistics and Reproducibility below for the reviewer’s reference:

“The Methods should include a separate section titled “Statistics and Reproducibility” with general information on how the statistical analyses of the data were conducted, and general information on the reproducibility of experiments, including the sample sizes and number of replicates and how replicates were defined.”

We have rewritten this paragraph and copied it below (relevant changes are green in the revised manuscript):

Neural data collection, ~~and~~ processing ~~and as well as~~ performance of the click detector were reproducible ~~across sessions~~ as the participant was able to repeatedly demonstrate click control ~~with stable performance across sessions on different days using neural signals from attempted hand movements to spell sentences~~. Samples of sensitivity, TPF, FPF, latencies, CCPM, WCPM, and CWPM measurements were created with data from online BCI use with voting thresholds ranging from 2 to 7 votes. Only samples from the 4-vote, 6-vote, and 7-vote conditions were statistically compared. Samples of simulated sensitivity, TPF, and FPF were the same size as those created with data from online BCI use with a 4-vote threshold. The sizes of the samples described above were not predefined as they depended on the number of spelling blocks collected with a specific voting threshold, which was adjusted based on participant feedback. We defined the sample size for comparing offline classification accuracies using various channel combinations (All channels, No Channel 112, Hand knob) as the number of folds used for cross-validation. ~~However, a~~ As this study reports on the first participant in this trial so far, further work will be necessary to test the reproducibility of these results in other participants.

Reviewer #2 (Remarks to the Author):

The authors have adequately addressed my concerns and made the desired changes. I thank the authors for their clear and concise responses and I have no further objections to the publication of this paper.

We thank Reviewer #2 for the feedback.

Reviewer #3 (Remarks to the Author):

I would like to thank authors for answering my queries and their hard work for revising the manuscript based on some suggestions. I am satisfied with the revision. I have a few minor follow-up questions:

1) Line 69 needs a reference: “However, the utility of ECoG for chronically (> 30 days) implanted BCIs has only been tested in a few participants.”

Thank you for this suggestion. We have added the appropriate references to this sentence (superscript citations numbers are purple in revised manuscript).

However, the utility of ECoG for chronically (> 30 days) implanted BCIs has only been tested in a few participants^{1,3,19}.”

References:

1. Vansteensel, M. J. et al. Fully Implanted Brain–Computer Interface in a Locked-In Patient with ALS. *N Engl J Med* **375**, 2060–2066 (2016).

3. Mitchell, P. et al. Assessment of Safety of a Fully Implanted Endovascular Brain–Computer Interface for Severe Paralysis in 4 Patients: The Stentrode With Thought-Controlled Digital Switch (SWITCH) Study. *JAMA Neurol* **80**, 270 (2023).

19. Oxley, T. J. et al. Motor neuroprosthesis implanted with neurointerventional surgery improves capacity for activities of daily living tasks in severe paralysis: first in-human experience. *J NeuroIntervent Surg* neurintsurg-2020-016862 (2020) doi:10.1136/neurintsurg-2020-016862.

2) Studies mentioned in lines 73, 75 etc. require references. It is not clear how many different studies are discussed in these couple of sentences.

Thank you. We have now referenced three studies in the first two sentences of paragraph 2 of the Introduction (Lines 72-76) and have copied these sentences below (superscript citations numbers are purple in revised manuscript):

“Recent studies have demonstrated ECoG-based BCI control for participants with amyotrophic lateral sclerosis (ALS) by detecting a “brain click,”^{1,3,19} an event-related change in spectral signals due to a distinct action, such as attempting a hand movement. In a recent clinical trial, participants with ALS (or primary lateral sclerosis) were implanted with an endovascular stent-electrode array for detecting such brain clicks^{3,19}.”

Note that we go on to summarize these studies in the following paragraphs.

References:

1. Vansteensel, M. J. et al. Fully Implanted Brain–Computer Interface in a Locked-In Patient with ALS. *N Engl J Med* **375**, 2060–2066 (2016).

3. Mitchell, P. et al. Assessment of Safety of a Fully Implanted Endovascular Brain-Computer Interface for Severe Paralysis in 4 Patients: The Stentrode With Thought-Controlled Digital Switch (SWITCH) Study. *JAMA Neurol* **80**, 270 (2023).

19. Oxley, T. J. et al. Motor neuroprosthesis implanted with neurointerventional surgery improves capacity for activities of daily living tasks in severe paralysis: first in-human experience. *J NeuroIntervent Surg* neurintsurg-2020-016862 (2020) doi:10.1136/neurintsurg-2020-016862.

3) In the last sentence of the introduction (line 103), authors state that this study demonstrates an improved click detector with increased spelling rate. In the previous two paragraphs in the introduction, several different metrics of spelling/click performance of various studies have been reported. It would be very useful if you can bring your results together and compare them with these previous studies directly (you can choose one or more metric/s such as spelling rate, click detection latency, detection accuracy) in one place in the discussion section to support the statement in line 103. Currently in the discussion, the results are only compared with Vansteensel 2016, and it is somewhat difficult to compare your results with the different metrics reported for the studies in the introduction.

Thank you for this suggestion. In paragraph 5 of the Discussion, we have now added our spelling rate and latency for a direct comparison to these metrics reported in Vansteensel et al., (2016) (ref. 1 in Introduction), but we qualify the latency comparison with the fact that the participant in that study was in a locked-in state. We have copied this below for the reviewer's convenience (relevant text is purple in the revised manuscript):

"Our results improve upon the previous switch scanning performance reported by Vansteensel et al., (2016)¹. In that study in which a participant with ALS ~~that~~ was implanted with four contacts over hand motor cortex and achieved a spelling rate of 1.8 ~~CPM~~ letters per minute ~~and~~ with a latency of at least 1 s per click (compared to a spelling rate of 10.2 CCPM and 0.68 s latency reported in the present work). The authors were not able to measure the latency from movement attempts to click detection due to the locked-in state of their participant. Because their click detector required five consecutive 200 ms epochs with neural features exceeding a pre-determined threshold, their latency could not have been shorter than 1 s."

We have additionally included a brief discussion on the spelling rate from Oxley et al., (2020) (ref. 19 in Introduction) in paragraph 5 of the Discussion (relevant text is purple in the revised manuscript):

"Although spelling rates were slightly lower than those observed in Oxley et al., (2020)¹⁹ (14-18 CCPM) the participants in that study used eye-tracking (ET) to first navigate to the appropriate letter before clicking it."

We believe the spelling rate from Oxley et al. is the most salient metric for comparison to our results, although the participant in that study did not use a switch scanning paradigm to navigate to the appropriate letter. We have also explained this in paragraph 2 of the Introduction:

"These brain clicks were generated by attempted foot movements and were used to select a particular icon or letter on a computer screen after navigating to it via eye-tracking (ET)¹⁹."

Finally, we have now also included a brief discussion on the same group's latency measurements from Mitchell et al., (2023) (ref. 3 in Introduction) immediately following the discussion on Oxley et al., (2023) (relevant text is purple in the revised manuscript):

“Additionally, though the same group reported accuracies of 97.4% and ~82% in selecting one of five targets (without ET), these corresponded to relatively longer latencies of 2.5 s and 0.9 s, respectively³.”

4) In line 64, I am not sure what methodology for recalibration has been used in reference 16, but I don't think it is simply correcting text outputs using LM for recalibration, rather I think it is recalibrating the decoder by retraining it continuously in the background after every trial to keep its accuracy steady over longer timeframes. Please check this in the cited reference and rephrase the statement accordingly.

Thank you for this suggestion. We have rephrased our sentence to specify that the recalibration was done continuously, in the background, and on a per-trial basis (relevant modifications are purple in the revised manuscript):

*“Nevertheless, there have been promising advances in **continual** online recalibration (in the background and on a per-trial basis) after ~~by~~ correcting text outputs using a language model¹⁶.”*

We believe that this description now matches closely to the original description of this technique (Fan et al., 2023), which we have copied below:

*“In this work, we present an alternative approach called **CORP: Continual Online Recalibration with Pseudo-labels**. CORP leverages the structure in language to enable self-recalibration of communication iBCIs without interrupting the user (i.e., plug-and-play). Specifically, CORP uses language models (LMs) to automatically correct communication iBCI text outputs, and continually retrains the decoder using these corrected outputs (“pseudo-labels”).”*

References:

16. Fan, C. et al. Plug-and-Play Stability for Intracortical Brain-Computer Interfaces: A One-Year Demonstration of Seamless Brain-to-Text Communication. in *Advances in Neural Information Processing Systems* vol. 36 (2024).

5) In reference to my comment about other assistive devices used by the participant, if the participant is still enrolled in the trial, I would suggest that the authors ask the participant for his informal subjective view on how he finds using the spelling/click BCI and how it compares to the methods he uses for communication in his daily life. Please also ask the participant what kind of assistive devices he currently uses and report this in the manuscript even if you don't have access to formal comparison. It is quite important for the field to understand BCI users' perspectives on the BCIs they are testing (especially for

the implanted BCIs) and other assistive technologies available to them.

Thank you for this suggestion. We have added the participant's informal subjective view as Supplementary Note 5 and referred to in paragraph 2 of the Participant section (relevant text is purple in main text and in the Supplementary Information):

Supplementary Information:

"During the study described in this writing, it was not necessary for the participant to rely on his BCI, and he mostly did not need to rely on other assistive communication devices at home. Due to his slowly progressing condition, he could still verbally communicate with the experimental team and his family, albeit laboriously and with limited intelligibility. Further, the participant retained eye movements and residual movements in both his upper and lower extremities but could not perform activities of daily living without assistance.

When the participant did use an assistive communication device at home, he used primarily his Tobii Dynavox (Danderyd, Sweden) or cell phone. The participant used the Tobii Dynavox's virtual keyboard (QWERTY layout) for typing and used eye-tracking to select letters and predictive words. Additionally, the participant would also type slowly on his cell phone using primarily his index finger, and he used the text-to-speech feature to synthesize his text to audio. He used these methods infrequently.

During the study, the participant reported that it was still easier to control the Tobii Dynavox's spelling application via eye control than it was to control the BCI-based spelling application. His eye movements were not yet significantly affected by ALS. Nevertheless, eye fatigue would require him to pause the application for a few minutes every 10-15 minutes. Also, jitter in the eye cursor was more pronounced when he was fatigued. Finally, changes in the ambient lighting sometimes required him to adjust his eye movements to correct for unintended cursor deviations."

Participant (paragraph 2):

"The participant was a right-handed man who was 61 years old at the time of implant in July 2022 and diagnosed with ALS roughly 8 years prior. Due to bulbar dysfunction, the participant had severe dysphagia and progressive dysarthria. This was accompanied by progressive dyspnea. The participant could still produce overt speech, but slowly and with limited intelligibility. He did not, however, heavily rely on assistive communication devices (Supplementary Note 5)."

6) Line 263: Thanks for pointing out that HG features were chosen as their modulation happens on a faster timescale as compared to ERD/S. However, I am not sure why the occurrence of ERS post-ERD would be a problem. I am not convinced with the presence of ERS as the reason for excluding low-frequency activity. In theory, one could still only use ERD occurring prior to the movement onset to detect click without introducing latencies and completely disregard ERS arising afterwards. Also looking at supplementary figure 2, it appears that HG and peak ERD (dark blue) occur for the same duration (timescale).

Thank you for these suggestions and observations.

We agree with the reviewer that the low-frequency ERS arising after ERD could potentially be disregarded as it would have likely occurred during the post-click lock-out periods, during which the decoder output was irrelevant. However, we would still need to explore strategies of optimally labeling samples from this period of ERS such as introducing a third class called “ers” to not otherwise increase the variance of the feature space for existing “rest” samples.

Briefly it is worth clarifying that though peak ERD and HG activity seemed to occur during the same time period, low frequency ERD generally occurred on a longer time scale. For example, consider channel 107 in Supplementary Fig. 2 where the post-cue ERD (in frequencies close to 50 Hz) visibly extended past the HG ERS. Additionally, some channels contained increased low-frequency ERD prior to cue (vertical black line, Supplementary Fig. 2), which was likely due to anticipatory activity during training data collection (as mentioned in the caption of Supplementary Fig. 2). This activity could have introduced a large variance in the feature space for samples labeled as rest and thus could have resulted in potentially decreased decoder output. We have added this possibility in the Feature extraction section (relevant text is in purple in the revised manuscript):

“Similarly, some channels contained low-frequency ERD prior to cue, which was likely due to anticipatory activity (Supplementary Fig. 2). This could have caused greater variance in the feature space of samples labeled as rest.”

However, we have acknowledged in paragraph 8 (previously paragraph 7) of the Discussion that low-frequency features were robust and could have been leveraged to improve decoder performance after appropriately addressing the challenges described above. We have copied those sentences below for the reviewer’s convenience (relevant text is in purple in the revised manuscript):

“Additionally, we did not explore the utility of low frequency power suppression, which has been stable and useful in other studies^{60,61}. It is possible that by optimizing our training paradigm to minimize anticipatory low-frequency ERD and by appropriately labeling rebound ERS for model training, we could further improve the robustness of our click detector.”

7) Thanks for clarifying that the character/word rates were computed after autocomplete assist. The true positive frequency of this decoder is 10 per min with 7-vote but the correct character rate is only 9.1 (that too after autocomplete assist) and similarly for 4-vote strategy, TPF (11) is higher than CCPM (10.2). Is this because the participant clicked on wrong characters more often? If so, what might be the reason for this (e.g., layout/UX of the switch scanning app)? If available, please report wrong character selection rate per minute (WCPM) so the CCPM and speller performance can be understood in this context in results paragraph - line 604.

Thank you for this question. Due to the design of the switch scanning paradigm, the participant needed to perform two clicks for each button selection. The first click was for selecting the row, while the second click was for selecting a button within that row. For example, without predictive language modeling and assuming no incorrect clicks, the TPF would have been equal to twice the CCPM. We have clarified this at the bottom of paragraph 4 in Switch scanning performance (relevant text is purple in the revised manuscript):

“Note that the CCPM for both voting conditions was lower than the respective TPFs, most likely because two true positive detections were necessary for one correct button click (be it a letter on the static keyboard or predicted letter or word).”

We have now also reported the wrong character per minute rate (WCPM) in the relevant sections of the Methods and Results (in the revised manuscript, relevant modifications are purple and sometimes also bold for extra visibility):

Spelling rates:

*“Spelling rates were measured **by-in units of** correct characters per minute (CCPM) and correct words per minute (CWPM). Spelled characters and words were correct if they exactly matched their positions in the prompted sentence. For example, if the participant spelled a sentence with 30 characters (5 words) with 1 character typo, only 29 characters (4 words) contributed to the CCPM (CWPM). **The frequency of character typos was measured in units of wrong characters per minute (WCPM).** The participant was instructed to correct any mistakes before proceeding to type the rest of the sentence. Note that all spelling was performed with assistance of autocompletion options from the language model and subsequently all analyses of spelling performance were based on this assisted spelling.”*

Statistical analysis:

*“Spelling blocks with a specific voting threshold were collected on no more than nine sessions. Given this small sample size, we could not assume normality in the distribution of the sample mean of any of the performance metrics (sensitivity, TPF, FPF, latencies, CCPM, **WCPM, and CWPM.**”*

Switch scanning performance (paragraph 4):

*“Consequently, the participant was able to achieve high rates of spelling (**Fig. 4d, e**). Specifically, the median spelling rate was 9.1 correct characters per minute (CCPM) using the 7-vote threshold, which significantly improved to 10.2 CCPM using the 4-vote threshold ($W = -2.163$, $P = 0.031$, two-sided Wilcoxon Rank-Sum test). **The wrong characters per minute (WCPM) rate was low at 0.2 using the 7-vote threshold and remained low at 0.1 after switching to the 4-vote threshold ($W = 1.192$, $P = 0.233$, two-sided Wilcoxon Rank-Sum test).**”*

Click detector retraining due to transient performance drop (paragraph 3):

*“Using this threshold, we achieved similar performance metrics to those from the original click detector with a 4-vote threshold, namely a median detection sensitivity of 94.8%, median TPF and FPF of 11.3 per min and 0.20 per min respectively, and a median CCPM, **WCPM**, and CWPM of 10.1, **0.1** and 2.2, respectively (for all comparisons $P > 0.05$, two-sided Wilcoxon Rank-Sum test) (Supplementary Fig. 19).”*

Additionally, we have also made the appropriate modifications to Fig. 4, and Supplementary Figs. 14 and 19. We have copied below Fig. 4 of the main text for the reviewer’s convenience (relevant modifications are purple in the revised manuscript):

Figure 4 | Long-term switch scanning spelling performance. Across all subplots, triangular and circular markers represent metrics using a 7-vote and 4-vote voting threshold, respectively. (a) Sensitivity of click detection for each session. (b) True positive and false positive frequencies (TPF and FPF) measured as detections per minute. (c) Average latencies with standard deviation error bars of grasp onset to algorithm detection and to on-screen click. The averages and standard deviations were computed from latency measurements across all spelling blocks from one session using the same voting threshold. Using 7-vote and 4-vote voting thresholds, on-screen clicks happened an average of 207 ms and 203 ms, respectively after detection. Note that algorithmic detection latencies were not registered in the first six sessions. (d) Correct and wrong characters and words per minute (CCPM and WCPM-CWPM, respectively). (e) Correct words per minute (CWPM).

8) With the latency of 0.6s per click + 1s cool off period (total latency 1.6s per click), I think the maximum capacity of this BCI can be 37 clicks per min. It would be good to mention this and discuss why the TPF is only 11 (despite very low FPF), which could be because of the wait times for the speller to slowly iterate through rows/cols to reach the desired character before it could be selected via BCI click. Hence, in theory, with faster UX, the character selection frequency can increase.

Thank you for this suggestion. In our previous revisions, we added Supplementary Note 8 (now Supplementary Note 9), where we discussed that the relatively low TPF could have been caused in large part by a total preparatory time of 4 s per scanning cycle. We have now also discussed the theoretical maximum TPF in the absence of any preparatory periods, and we have done this in the context of approaching a theoretical maximum of 35 per min with an optimal user interface. We have copied this paragraph below for the reviewer's convenience (added text is purple):

We included pre-selection rows and columns to allow the participant a brief preparation period before clicking into the first row or first column, respectively (Supplementary Fig. 9). However, the total time the switch scanner spent in these preparatory regions was 4 s per scanning cycle, during which the application could not receive clicks. For a future study participant, we may slowly start shortening these preparation periods and possibly eliminate them completely as the participant becomes more comfortable with timing his or her attempted grasps to highlighted rows or columns. Considering that we designed the application such that the language model's most probable words and letters appear in the top two "clickable" rows, reducing the total scan time by 4 sec per desired button could significantly increase the spelling rate. For example, assuming a TPF of 11 per min where half of the true positives were row clicks (succeeding preparatory periods of 3 s) and half were column clicks (succeeding preparatory periods of 1 s), the total time of the scanner spent in preparatory periods would be $22 \text{ s} \left(5.5 \text{ TP} \times \frac{3 \text{ s}}{\text{TP}} + 5.5 \text{ TP} \times \frac{1 \text{ s}}{\text{TP}} \right)$. Reducing these preparatory times to 0 s would have thus resulted in a TPF of 17.4 per min $\left(\frac{11 \text{ TP}}{60 \text{ s} - 22 \text{ s}} \times \frac{60 \text{ s}}{\text{min}} \right)$. It could similarly be beneficial to rearrange the keyboard such that the most used letters take the fewest scans to reach (i.e., placement in the top left corner of the keyboard). Indeed, given a latency of 0.68 s from attempted grasp onset to click with the lock-out period of 1 s (1.68 s attributed to each click) a theoretical maximum TPF of 35 clicks per min could be approached with the appropriate user interface.

9) In the figures, statistically significant difference between different elements could be using * notation or mention this in the caption of the figure for clarity if applicable.

Thank you for this suggestion. We reviewed the figures in the main text and in the supplementary material and confirmed that the only place where the * was appropriate was in Fig. 5g of the main text:

In this figure, we show in a box and whisker plot that there is a small, but significant difference in the confusion scores using features from all channels vs. features from channels over only the hand knob. We specified that the * represents a P-value of statistical significance in the caption of this figure:

“(* $P = 0.015$, two-sided Wilcoxon Rank-Sum test with 3-way Bonferroni-Holm correction).”

Other figures to which we considered adding a * were Fig. 4, and Supplementary Figs. 11 and 18. However, for Fig. 4, we did not display the comparison between 7-vote and 4-vote metrics using box and whisker plots (or any comparable visualization) and therefore there was nowhere to neatly place the *. In Supplementary Fig. 11, there were no statistically significant differences between simulated sensitivities or FPFs corresponding to any pair of number of training trials. In Supplementary Fig. 18, we did not compare metrics of any two voting thresholds using P-value; rather the information in this figure was used for computing an F_1 -score for selecting the optimal voting threshold.

10) Line 642 “Conversely, we found an increase in the magnitude of low frequency power”, should this be ... conversely, we found increased desynchronization in low frequencies during this transient period? Looking at supplementary figure 16, it appears that ERD has increased during this transient period.

Thank you for this suggestion. We have corrected this as recommended. See below for your convenience (relevant text is purple in the revised manuscript):

“Conversely, we found an increase in the event-related desynchronization (ERD) magnitude of low frequency power (10-30 Hz) (Supplementary Fig. 16).”

Reviewer #4 (Remarks to the Author):

the authors have done a great job to address my comments. Eddie Chang

We thank you for your feedback.

REVIEWERS' COMMENTS:

Reviewer #1 (Remarks to the Author):

Thank you for the revisions. I recommend moving to publication.

Reviewer #3 (Remarks to the Author):

The authors have addressed all my comments and I am satisfied with the revised manuscript.